# `LightFair`: Towards an Efficient Alternative for Fair T2I Diffusion via Debiasing Pre-trained Text Encoders

**Boyu Han**[1,2]      **Qianqian Xu**[1,3*]      **Shilong Bao**[2]      **Zhiyong Yang**[2]
**Kangli Zi**[1]      **Qingming Huang**[2,1*]

[1] State Key Laboratory of AI Safety, Institute of Computing Technology, CAS
[2] School of Computer Science and Tech., University of Chinese Academy of Sciences
[3] Peng Cheng Laboratory

{hanboyu23z,xuqianqian,zikangli}@ict.ac.cn,
{baoshilong,yangzhiyong21,qmhuang}@ucas.ac.cn

## Abstract

This paper explores a novel lightweight approach `LightFair` to achieve fair text-to-image diffusion models (T2I DMs) by addressing the adverse effects of the text encoder. Most existing methods either couple different parts of the diffusion model for full-parameter training or rely on auxiliary networks for correction. They incur heavy training or sampling burden and unsatisfactory performance. Since T2I DMs consist of multiple components, with the text encoder being the most fine-tunable and front-end module, this paper focuses on mitigating bias by fine-tuning text embeddings. To validate feasibility, we observe that the text encoder's neutral embedding output shows substantial skewness across image embeddings of various attributes in the CLIP space. More importantly, the noise prediction network further amplifies this imbalance. To finetune the text embedding, we propose a collaborative distance-constrained debiasing strategy that balances embedding distances to improve fairness without auxiliary references. However, mitigating bias can compromise the original generation quality. To address this, we introduce a two-stage text-guided sampling strategy to limit when the debiased text encoder intervenes. Extensive experiments demonstrate that `LightFair` is effective and efficient. Notably, on Stable Diffusion v1.5, our method achieves SOTA debiasing at just $1/4$ of the training burden, with virtually no increase in sampling burden. The code is available at https://github.com/boyuh/LightFair.

## 1 Introduction

Recently, with the rapid progress of machine learning [108, 5, 101, 104, 103] and computer vision [54, 33, 32, 96, 93], *text-to-image (T2I)* diffusion models [38, 9, 44], such as *Stable Diffusion (SD)* [78], have gained widespread attention. These models effectively combine text-based inputs with image generation, delivering remarkable performance across a broad range of applications [86, 55, 17, 11, 49]. However, research [13, 82, 92] has revealed that these models often produce **biased content** regarding various demographic factors, say gender, race, and age. Such biases pose significant societal risks, particularly when these models are deployed in real-world scenarios [10, 13, 62, 112].

Many efforts have been made to mitigate attribute bias, which can generally be divided into two camps. The first camp [84, 27] involves retraining or fine-tuning diffusion models to adjust the generated distribution. However, most of them rely on a strategy that couples different parts of the diffusion model for full-parameter training. It leads to highly complex gradient chains [84], resulting in a significant computational burden, as shown by the **solid-lined method** in Fig. 1. Moreover, tuning a large number of parameters may lead to excessive debiasing, which lowers generation quality.

---

[*]Corresponding authors.

39th Conference on Neural Information Processing Systems (NeurIPS 2025).

The second camp [24, 15, 45] uses post-processing methods during inference, relying on external reference information or auxiliary networks to address attribute bias. The use of third-party models increases sampling time and reduces generation efficiency, as shown by the **dashed-lined methods** in Fig. 1. Furthermore, hidden biases in these third-party models often prevent these methods from ensuring complete fairness. Hence, a natural question arises: *Can we develop a lightweight alternative to resolve attribute bias effectively?*

In search of an answer, this paper explores a novel approach `LightFair` to achieve lightweight debiasing by refining the pre-trained text encoder. Specifically, we argue that the text embedding in T2I DMs inherently carries biases, which lead to biased generated images. To this end, we empirically demonstrate that an **unfair** T2I DM exhibits a **skewed or imbalanced embedding distribution** between neutral text and various image attributes in the CLIP space, where the extent of bias is reflected in the distances between these embeddings. Most importantly, we reveal that the bias introduced by the text encoder can be **further amplified** during the recurrent denoising prediction steps, underscoring the critical need to directly debias the text encoder.

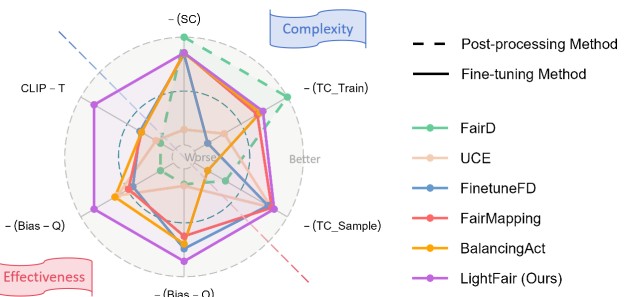

Figure 1: **Overview of the complexity and effectiveness of different methods.** Complexity metrics include spatial complexity during training (SC), time complexity during training (TC_Train), and time complexity during sampling (TC_Sample). Effectiveness metrics include Bias-O and Bias-Q for measuring generative bias, and CLIP-T for evaluating generation quality. Lower-is-better metrics are negated for consistent comparison. See Sec. I.15 for detailed results.

Next, to ensure efficient inference, we perform LoRA [41] fine-tuning on the text encoder to mitigate bias without relying on auxiliary networks. However, achieving fairness requires aligning the attribute distribution of generated images with a fair distribution, but the former is often difficult to obtain. Based on our findings, we demonstrate that equalizing distances within the embedding space can implicitly achieve equalized distributions (Thm. 4.1). Motivated by this, we propose a collaborative distance-constrained debiasing strategy, which comprises two key components. 1) it enforces constraints on the distances between text embeddings and the semantic centers of different attributes, promoting both equalized odds and equalized quality. 2) since we use image embeddings to approximate the semantic centers of various attributes, we introduce an adaptive foreground extraction strategy to minimize the influence of background features. As shown in Fig. 1, our `LightFair` achieves lower time and space complexity, ensuring lightweight debiasing.

Taking a step further, we observe that debiasing may inevitably harm the model's generation quality [84, 51]. To mitigate this impact, it is crucial to find the optimal intervention time for the debiased text encoder. To that end, we conduct a fine-grained analysis of the diffusion model's generation process [72, 60]. The results reveal that low-frequency information (attribute-independent) emerges during early denoising stages, while high-frequency information (**attribute-dependent**) appears in **later** stages (Thm. 4.2). In light of this, we propose a two-stage text-guided sampling strategy, where the debiased text encoder is applied only in the later sampling stages. This approach balances bias mitigation and image quality preservation. Finally, comprehensive empirical studies consistently speak to the efficacy of our proposed method.

Our main contributions are summarized as follows:

- This paper shows the adverse effects of the text encoder on fairness in text-to-image diffusion models. To our knowledge, this issue remains underexplored within the fairness community.

- We propose a lightweight fine-tuning method `LightFair` to achieve fair diffusion models. It employs a collaborative distance-constrained debiasing strategy to maintain both equalized odds and equalized quality. It also incorporates a two-stage text-guided sampling strategy that mitigates its impact on image generation quality.

- Comprehensive empirical results across two versions of SD, four attributes, and diverse prompts demonstrate the effectiveness and lightweight nature of our proposed method in addressing bias.

## 2 Related Work

**Text-to-image Generative Methods.** The fields of machine learning [85, 7, 6, 18, 95, 102] and computer vision [2, 3, 4, 42, 43, 58, 61] have undergone a paradigm shift from understanding [66, 64, 63, 97, 56, 25] to generative models [38, 78, 31, 110]. Among these, T2I generative modeling has emerged as a key research direction. T2I generation methods are mainly divided into three categories based on their probabilistic modeling approach: autoregressive models [76, 106, 109], generative adversarial networks [30, 47, 48, 71], and diffusion models [38, 78, 9, 21]. In recent years, diffusion models have advanced significantly, offering greater stability, scalability, and higher image quality. *Denoising Diffusion Probabilistic Models (DDPM)* [38] generate images unconditionally through a straightforward, iterative denoising process. Stable Diffusion [78], an extension of DDPM, incorporates text guidance to produce high-resolution images. Additionally, diffusion-based architectures have been successfully applied to various tasks, including style-transfer [73, 111, 74, 16], scene generation [11, 98, 87, 100, 28, 53], and image-editing [67, 17, 29, 50, 57], achieving notable results.

**Bias in Diffusion Models and Mitigation Methods.** Diffusion models are highly data-driven and prone to inheriting and amplifying imbalances and biases [10, 13, 62, 88, 112] present in large-scale datasets [81]. [13, 82, 92] observe that, when no attribute prompts are provided, Stable Diffusion exhibits attribute biases along social dimensions such as gender and race. [24] guides fair generation by introducing random attribute text prompts. [15, 45] perform text prompt corrections in the latent space. [84] modifies model parameters through fine-tuning on balanced data. [27] conducts concept editing by updating the model's cross-attention layers. [70, 51, 91, 39] introduce an auxiliary network to help the model eliminate bias. However, these methods often treat the diffusion model as an end-to-end system, overlooking the unique roles of its individual components. Such untargeted fine-tuning may lead to over-debiasing and a decline in generation quality. Moreover, many of these approaches depend on external reference information or auxiliary networks to address attribute bias. The fairness and performance of these third-party models are difficult to guarantee, making it challenging to achieve a truly fair diffusion model. Most importantly, these methods impose significant computational burdens during training or sampling. To address these issues, this paper proposes a debiasing method focused on the text encoder. The method features a lightweight design, eliminates the need for auxiliary networks, and offers a targeted approach to mitigate bias.

## 3 Preliminaries

In this section, we briefly introduce the diffusion model and the fair diffusion model.

**Diffusion Model.** The diffusion model [38] consists of two processes: a forward noising process and a reverse denoising process. In the forward process, samples $\mathbf{x}_0 \sim q(\mathbf{x})$ drawn from a given data distribution are progressively corrupted with Gaussian noise, eventually degrading into pure Gaussian noise over $T$ time steps. It is defined as:

$$q(\mathbf{x}_t|\mathbf{x}_0) = \mathcal{N}(\mathbf{x}_t; \sqrt{\bar{\alpha_t}}\mathbf{x}_0, (1 - \bar{\alpha}_t)\mathbf{I}), \tag{1}$$

where $\overline{\alpha_t} = \prod_{i=1}^{t} \alpha_i$ and $\alpha_t \in (0, 1)$ is a hyperparameter controlling the noise level. In the backward process, a neural network parameterized by $\theta$ predicts the noise added at each time step during the forward process, recovering $\mathbf{x}_T$ back to the original data distribution $\mathbf{x}_0$. The denoising process is modeled as:

$$p_\theta(\mathbf{x}_{t-1}|\mathbf{x}_t) = \mathcal{N}(\mathbf{x}_{t-1}; \mu_\theta(\mathbf{x}_t, t), \sigma_t^2\mathbf{I}), \tag{2}$$

Here, $\mu_\theta(\mathbf{x}_t, t) = \frac{1}{\sqrt{\alpha_t}}(\mathbf{x}_t - \frac{\beta_t}{\sqrt{1-\bar{\alpha}_t}}\epsilon_\theta(\mathbf{x}_t, t))$ is parameterized by the noise prediction network $\epsilon_\theta(\mathbf{x}_t, t)$, $\beta_t = 1 - \alpha_t$, and $\sigma_t^2$ is typically chosen as either $\sigma_t^2 = \beta_t$ or $\sigma_t^2 = \frac{1-\bar{\alpha}_{t-1}}{1-\bar{\alpha}_t}\beta_t$.

**Stable Diffusion.** The Stable Diffusion [78] is a classic text-to-image diffusion model. It additionally provides a prompt $P$ to guide the diffusion model in generating images. Specifically, it employs a noise prediction network (typically a U-Net) in the latent space while utilizing a text encoder (usually CLIP) to encode $P$, thereby providing textual guidance. For latent diffusion models, an image encoder $g^e$ maps the training image $\mathbf{x}_0$ to its latent space representation $\mathbf{z}_0 = g^e(\mathbf{x}_0)$. The image decoder $g^d$ maps the denoised $\mathbf{z}_0$ from the generation process back to the image space as $\mathbf{x}_0 = g^d(\mathbf{z}_0)$. For the text encoder, it encodes the textual prompt $P$ using $f^t$, which is then incorporated into the noise prediction network $\epsilon_\theta(f^t(P), \mathbf{z}_t, t)$.

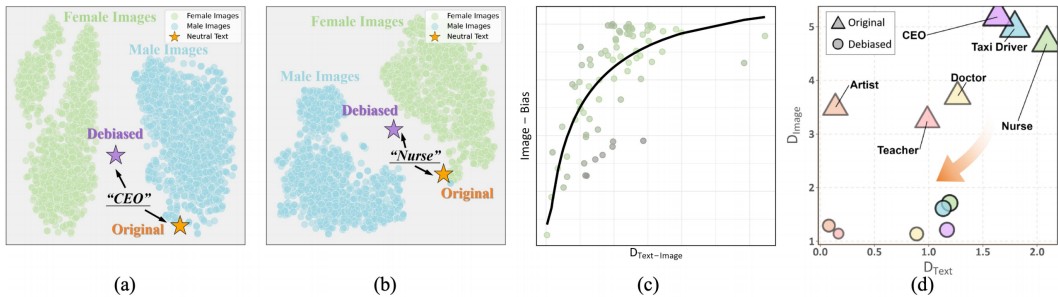

Figure 2: (a)-(b) T-SNE visualization of original and debiased ‘`CEO`’/‘`Nurse`’ text embeddings alongside male and female image embeddings. (c) Visualization of distance differences ($D_{\text{Text-Image}}$) and generated image bias across $80$ occupations. (d) Visualization of original/our debiased text embedding distance differences ($D_{\text{Text}}$) and image embedding distance differences ($D_{\text{Image}}$).

**Fair Diffusion Model.** Following the notations in [51, 70, 91], the textual prompt $P = prompt(a, c)$ is typically composed of $a$, an attribute word from the set $A$, and $c$, a main word from the set $C$. For example, $prompt(\text{‘female’}, \text{‘doctor’})$ represents "`Photo portrait of a female doctor`". We denote textual prompts without an attribute word as $prompt(\cdot, c)$, for instance, $prompt(\cdot, \text{‘doctor’}) = $ "`Photo portrait of a doctor`". Currently, fair diffusion models have two goals:

**Goal 1: Equalized Odds** encourages equal generation frequency for images with different attributes when given an unspecified attribute prompt $prompt(\cdot, c)$, expressed as:

$$\mathbb{P}(a_i | prompt(\cdot, c)) = \mathbb{P}(a_j | prompt(\cdot, c)), \quad \forall a_i, a_j \in A. \tag{3}$$

This probability is typically computed using an additional attribute classifier $h(\cdot)$.

**Goal 2: Equalized Quality** promotes equal image quality for different attribute images, expressed as:

$$Q(prompt(a_i, c)) = Q(prompt(a_j, c)), \quad \forall a_i, a_j \in A, \tag{4}$$

where $Q(prompt(a_i, c))$ represents the quality score of the images generated using $prompt(a_i, c)$. The quality score is typically measured using metrics such as CLIP [75] or DINO [68].

## 4 `LightFair`

In this section, we explore achieving a fair diffusion model through lightweight fine-tuning. We first identify the text encoder as a key structure contributing to bias (Sec. 4.1) and propose the collaborative distance-constrained debiasing strategy to address it without auxiliary networks (Sec. 4.2). We then analyze the diffusion process to determine the optimal timing for applying the debiased text encoder, mitigating its impact on performance (Sec. 4.3). Following the setup in [70, 84, 15, 27, 51], this paper focuses on addressing attribute bias in Stable Diffusion [78] and uses the example of gender bias in occupations to illustrate the discussion. A table of symbol definitions is provided in Sec. A.

### 4.1 A Closer Look at the Pre-trained Text Encoder

Stable Diffusion employs a pre-trained text encoder (*e.g.*, CLIP) to encode textual inputs, which then guide the noise prediction network (*e.g.*, U-Net). **The noise prediction network usually has more parameters than the text encoder (details are provided in Sec. B).** Thus, we aim to investigate whether fine-tuning the text encoder can correct biases, which is lightweight and underexplored.

To investigate the bias within the text encoder, we propose two progressive research questions. **(RQ 1)** How can we measure bias within the text embeddings? Existing methods for assessing bias in SD typically rely on performing attribute classification on generated images. However, the text encoder outputs embeddings, making it challenging to directly quantify the bias. **(RQ 2)** Does the bias in the text encoder affect the output of the subsequent noise prediction network?

To address **(RQ 1)**, we investigate the relationship between distance and bias within the text-image semantic space aligned by CLIP. First, we conduct a simple empirical experiment as an initial

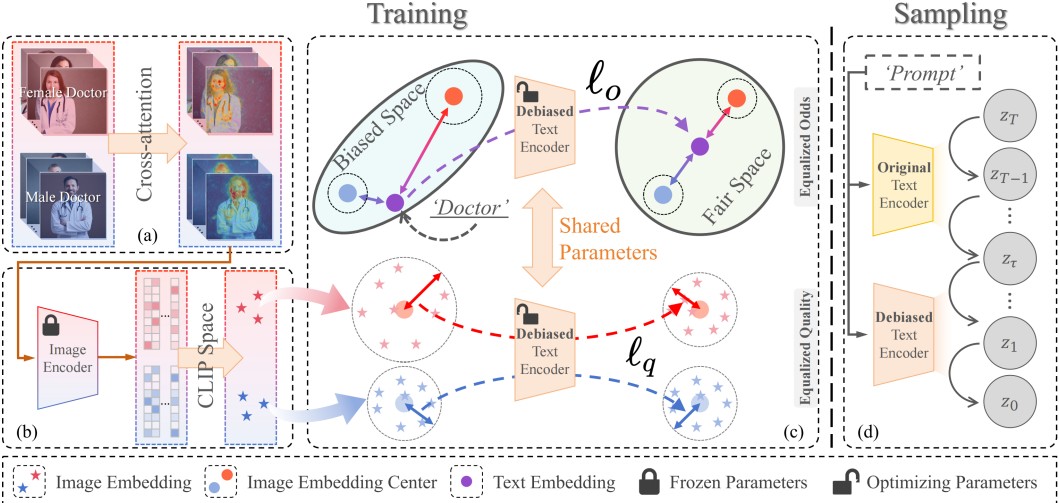

Figure 3: **An overview of our** `LightFair`. a) We first perform adaptive foreground extraction on images with different attributes. b) Then, the foreground images are encoded by an image encoder to obtain the centroid for each attribute. c) Lightweight fair fine-tuning is conducted using collaborative distance constraints. d) A two-stage text-guided sampling strategy ensures generation quality.

exploration. It is widely acknowledged that SD tends to generate male images for the prompt `CEO'` and female images for the prompt `Nurse'` [24]. In the experiment, we find that the text embedding for `CEO'` is closer to the centroid of male images, while the embedding for `Nurse'` is closer to the centroid of female images, as shown in Fig. 2(a) and (b). Next, we quantify the relationship between this embedding distance and the bias in generated images. Specifically, we calculate the distance difference between the text embeddings of 80 occupations (details are provided in Sec. C) and the semantic centroids for `male'` ($\male$) and `female'` ($\female$):

$$D_{\text{Text-Image}} = \left| s\left(\text{emb}_c^T(\cdot), \mathbb{E}[\text{emb}_c^I(\male)]\right) - s\left(\text{emb}_c^T(\cdot), \mathbb{E}[\text{emb}_c^I(\female)]\right) \right|, \tag{5}$$

where $s(a, b)$ represents the cosine distance between $a$ and $b$. $\text{emb}_c^T(\cdot)$ and $\text{emb}_c^I(\cdot)$ are shorthand notations for $f^t(prompt(\cdot, c))$ and $f^i(M(prompt(\cdot, c)))$, respectively. $M(prompt(\cdot))$ denotes the images generated using $prompt(\cdot)$. $f^t(\cdot)$ and $f^i(\cdot)$ refer to the encoding operations performed by the CLIP text and image encoders. Additionally, we use these text embeddings to generate 500 images for each occupation and measure the gender bias in the images (details are provided in Sec. H.1, Bias-Odds). The results, shown in Fig. 2(c), indicate a clear trend: greater distance differences correspond to stronger gender bias in the generated images. **Therefore, distance is a good measure to reflect bias in text embeddings.**

To answer **(RQ 2)**, we investigate how the bias in text embeddings changes after passing through the noise prediction network. Following the conclusion from **(RQ 1)**, we use distance as a measure of bias. Specifically, we use six occupations as the main word $c$ for image generation. We then calculate the bias in the text embeddings from the text encoder ($D_{\text{Text}}$) and image embeddings of the output from the noise prediction network ($D_{\text{Image}}$), as follows:

$$D_{\text{Text}} = \left| s\left(\text{emb}_c^T(\cdot), \text{emb}_c^T(\male)\right) - s\left(\text{emb}_c^T(\cdot), \text{emb}_c^T(\female)\right) \right|, \tag{6}$$

$$D_{\text{Image}} = \left| s\left(\mathbb{E}[\text{emb}_c^I(\cdot)], \mathbb{E}[\text{emb}_c^I(\male)]\right) - s\left(\mathbb{E}[\text{emb}_c^I(\cdot)], \mathbb{E}[\text{emb}_c^I(\female)]\right) \right|. \tag{7}$$

We plot $D_{\text{Text}}$ and $D_{\text{Image}}$ in Fig. 2(d). The results show that the text encoder introduces bias into the model, and the noise prediction network further amplifies this bias during image generation. Our debiased text encoder produces text embeddings with less bias, leading to images with less bias. Therefore, we can get the following insight:

**Insight.1.** *The text encoder is one of the key yet overlooked structures contributing to attribute bias in Stable Diffusion.*

**Remark 1.** *The noise prediction network is not entirely independent of the text encoder, as encoded textual inputs directly influence the denoising process. **It is acknowledged that when there is no bias***

*in the text embeddings (e.g., for specified attributes), the noise prediction network exhibits minimal bias. This connection suggests that biases in the text encoder can propagate to the noise prediction network and be further amplified during training, underscoring the importance of analyzing and mitigating biases in the text encoder. Meanwhile, since the text encoder is trained independently of the noise prediction network, fine-tuning the text encoder separately is a feasible solution.*

## 4.2 Collaborative Distance-constrained Debiasing Strategy

We perform lightweight fine-tuning of the text encoder using a collaborative distance-constrained debiasing strategy to eliminate bias. A brief overview is provided in Fig. 3(a)-(c).

### 4.2.1 Debiasing Through Distance Constraints

To achieve **equalized odds**, we aim to generate images with equal probabilities for each attribute. However, it is challenging to obtain the probability distribution of generated images. To facilitate optimization, we theoretically explore the equivalence between Equalized Odds and Equalized Distance in Thm. 4.1.

**Theorem 4.1.** *Under Thm. D.1, D.2 and D.3, for any attributes $a_i, a_j \in A$, achieving Equalized Odds $\mathbb{P}(a_i|P(\cdot,c)) = \mathbb{P}(a_j|P(\cdot,c))$ is equivalent to ensuring Equalized Distance:*

$$\left\| f(a_i,c) - f^t\big(P(\cdot,c)\big) \right\|^2 = \left\| f(a_j,c) - f^t\big(P(\cdot,c)\big) \right\|^2 , \tag{8}$$

*where $f(a_i,c)$ represents the encoding of the concepts $a_i$ and $c$, $f^t(\cdot)$ denotes the encoding performed by the CLIP text encoder and $P(\cdot)$ is shorthand for $prompt(\cdot)$.*

The proof is deferred to Sec. D. Since the image $M(P(a,c))$ generated by the prompt $P(a,c)$ can serve as the encoding for the concepts $a$ and $c$, we approximate $f(a,c)$ by using the semantic center of multiple image embeddings $\mathbb{E}\left[\text{emb}_c^I(a)\right]$, as shown in Fig. 3(b). Ultimately, we can correct the bias by shifting the text embedding to a position where its distance from the embedding center of each attribute image is equal. The loss function can be expressed as:

$$\ell_o = \sqrt{\frac{1}{|A|} \sum_{i=1}^{|A|} \left[ s\Big(\text{emb}_c^T(\cdot), \mathbb{E}\left[\text{emb}_c^I(a_i)\right]\Big) - \overline{s}\right]^2}, \tag{9}$$

where $\overline{s} = \frac{1}{|A|}\sum_{i=1}^{|A|} s\Big(\text{emb}_c^T(\cdot), \mathbb{E}\left[\text{emb}_c^I(a_i)\right]\Big)$, and $s(a,b)$ represents the cosine distance between $a$ and $b$.

To ensure **equalized quality**, we aim to generate images that share the same CLIP score for each attribute. We compute the CLIP score of a single image as $s\Big(\text{emb}_c^T(a), \text{emb}_c^I(a)\Big)$. To find the quality distribution, we calculate the CLIP score for each generated image. This computation can be simplified by using the average image embedding for each attribute. Specifically, we set a constraint so that the distance between the image embedding center of each attribute and its corresponding text embedding is equal.

$$\ell_q = \sqrt{\frac{1}{|A|} \sum_{i=1}^{|A|} \left[ s\Big(\text{emb}_c^T(a_i)), \mathbb{E}\left[\text{emb}_c^I(a_i)\right]\Big) - \overline{s'}\right]^2}, \tag{10}$$

where $\overline{s'} = \frac{1}{|A|}\sum_{i=1}^{|A|} s\Big(\text{emb}_c^T(a_i)), \mathbb{E}\left[\text{emb}_c^I(a_i)\right]\Big)$.

We introduce an additional regularization term to constrain the text embeddings from deviating too far from the image embedding center.

$$\ell_{reg} = 1 - s\Big(\text{emb}_c^T(\cdot), \mathbb{E}_{i \in [1,|A|]}\left[\mathbb{E}\left[\text{emb}_c^I(a_i)\right]\right]\Big), \tag{11}$$

where $\mathbb{E}_{i \in [1,|A|]}\left[\mathbb{E}\left[\text{emb}_c^I(a_i)\right]\right]$ represents the centroid of all attribute image embeddings.

During fine-tuning, the loss function is constructed by jointly using Equ. (9), Equ. (10), and Equ. (11):

$$\ell = \ell_o + \lambda_1 \ell_q + \lambda_2 \ell_{reg}, \tag{12}$$

where $\lambda_1$ and $\lambda_2$ are hyperparameters. The entire process does not require additional auxiliary networks or the computation of complex gradient chains, ensuring lightweight fine-tuning.

### 4.2.2 Adaptive Foreground Extraction

When using image embeddings to represent semantic centers, we notice that the background of the image may introduce distractions. As shown in Fig. 4, in the case of the prompt "`Photo portrait of a white receptionist`", the generated image includes elements like *desk* and *door*. To address this issue, we use text guidance to highlight the pixels corresponding to the main word in the image. Specifically, it is achieved using a cross-attention layer:

$$\text{emb}_c^{I\prime}(a_i) = \text{Softmax}(\frac{\mathbf{Q}\mathbf{K}^T}{\sqrt{d}})\mathbf{V}, \quad (13)$$

where $\mathbf{Q} = \text{emb}_c^I(a_i), \mathbf{K} = \mathbf{V} = \text{emb}_c^T(a_i)$ are the query, key, value matrices of the attention

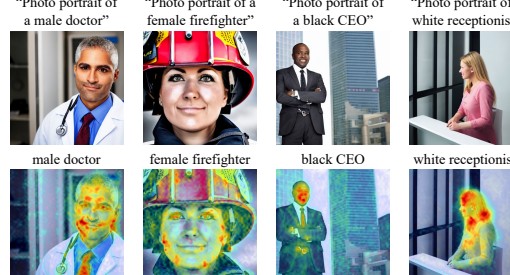

Figure 4: The first row includes images generated by SD using designed templates. The second row shows the visualization of the generated images after highlighting the main content.

operation, $d$ is the embedding dimension of $\mathbf{K}$. The highlighted image replaces the original image as input to the image encoder, as shown in Fig. 3(a). By using the highlighted image, the model can better focus on the target concept while reducing the influence of background information. As illustrated in Fig. 4, the phrase "`white receptionist`" directs the model's attention to the person, distinguishing her from the surrounding environment. In Sec. 5.3, we present additional experiments to demonstrate the effectiveness of this module.

### 4.3 Two-Stage Text-Guided Sampling Strategy

Although we aim to minimize the impact of debiasing on the model's generative performance by using multiple constraints, there is no free lunch. Fine-tuning inevitably affects the model's output quality. To mitigate this, we apply fine-tuned guidance only at critical times during the generation process rather than entirely replacing the original text encoder. This approach requires a detailed analysis of the diffusion model's generation process [52, 77, 8].

First, we identify frequency signal patterns in the diffusion denoising process:

**Proposition 4.2.** *The recovery rate of low-frequency signals during the diffusion denoising process is higher than that of high-frequency signals.*

The proof is deferred to Sec. E. The main word $c$ can be considered as low-frequency information because it describes macroscopic features, while the attribute $a$ represents high-frequency information because it focuses on detailed features. Consequently, Thm. 4.2 indicates that attribute information always emerges during the later stages of denoising.

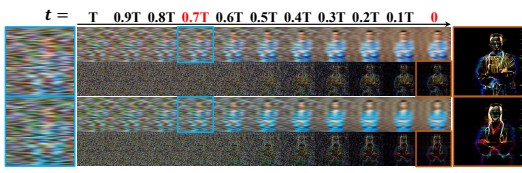

Figure 5: Results of **low-pass and high-pass filtering** applied during the denoising process with text guidance for '`male doctor`' (top two rows) and '`female doctor`' (bottom two rows). Each pair shows low-pass filtered images on top and high-pass filtered images below. For clarity, some images are enlarged and highlighted on both sides. More images are provided in Sec. F.

As shown in Fig. 5, we visualize the process of progressive denoising from Gaussian noise ($t = T$) to a clear image of a '`doctor`' ($t = 0$). It can be observed that low-frequency information emerges in the early stages of denoising, with the concept of '`doctor`' gradually taking shape, while high-frequency information remains obscured by noise. At $0.7T$, gender attributes are almost indistinguishable. Only in the later stages of denoising do gender-related attributes, such as hair and facial features, gradually appear. This confirms the correctness of Thm. 4.2.

Table 1: Selected representative quantitative results on gender and race attributes. The champion and the runner-up are highlighted in **bold** and underline. Complete results are provided in Tab. 6 and 7.

| Method | Gender | | | | | | | | Race | | | | | | | |
|---|---|---|---|---|---|---|---|---|---|---|---|---|---|---|---|---|
| | Fairness | | Quality | | | | | | Fairness | | Quality | | | | | |
| | Bias-O↓ | Bias-Q↓ | CLIP-T↑ | CLIP-I↑ | FID↓ | IS↑ | AS-R↑ | AS-A↓ | Bias-O↓ | Bias-Q↓ | CLIP-T↑ | CLIP-I↑ | FID↓ | IS↑ | AS-R↑ | AS-A↓ |
| **Stable Diffusion v1.5** | | | | | | | | | | | | | | | | |
| SD [78] | 0.73 (±0.05) | 1.90 (±0.67) | 29.31 (±0.06) | - | 275.13 (±6.75) | 1.26 (±0.03) | 4.78 (±0.08) | 2.65 (±0.04) | 0.54 (±0.02) | 1.60 (±0.67) | 29.31 (±0.06) | - | 275.13 (±6.75) | 1.26 (±0.03) | 4.78 (±0.08) | 2.65 (±0.04) |
| FairD [24] | 0.79 (±0.04) | 3.25 (±1.15) | 28.79 (±0.11) | 75.91 (±0.56) | 269.62 (±4.42) | 1.30 (±0.03) | 4.57 (±0.09) | 2.82 (±0.05) | 0.50 (±0.02) | 1.50 (±0.38) | 28.95 (±0.10) | 74.33 (±0.68) | 262.72 (±4.84) | 1.28 (±0.03) | 4.55 (±0.08) | 2.83 (±0.06) |
| UCE [27] | 0.78 (±0.07) | 1.79 (±0.46) | 28.91 (±0.13) | 82.72 (±0.81) | 273.95 (±5.53) | 1.26 (±0.03) | 4.71 (±0.09) | 2.64 (±0.04) | 0.44 (±0.03) | 1.40 (±0.24) | 29.13 (±0.14) | 90.15 (±0.70) | 281.16 (±5.18) | 1.26 (±0.02) | 4.76 (±0.08) | 2.69 (±0.05) |
| FinetuneFD [84] | 0.38 (±0.07) | 2.31 (±0.35) | 29.34 (±0.13) | 76.17 (±0.68) | 278.21 (±7.53) | 1.24 (±0.02) | 4.38 (±0.06) | 2.86 (±0.04) | 0.20 (±0.03) | 1.41 (±0.23) | 29.02 (±0.15) | 74.57 (±0.53) | 270.09 (±5.99) | 1.26 (±0.02) | 4.33 (±0.06) | 2.87 (±0.05) |
| FairMapping [51] | 0.46 (±0.05) | 2.16 (±0.72) | 29.30 (±0.16) | 76.00 (±0.66) | 278.81 (±5.84) | 1.26 (±0.02) | 4.34 (±0.07) | 2.90 (±0.03) | 0.34 (±0.02) | 1.75 (±0.47) | 29.29 (±0.15) | 76.54 (±0.71) | 280.95 (±5.02) | 1.26 (±0.03) | 4.53 (±0.08) | 2.80 (±0.05) |
| BalancingAct [70] | 0.41 (±0.05) | 1.70 (±0.55) | 29.30 (±0.11) | 77.37 (±0.64) | 272.08 (±5.16) | 1.28 (±0.02) | 4.71 (±0.06) | 2.68 (±0.04) | 0.34 (±0.02) | 1.13 (±0.36) | 29.34 (±0.11) | 77.44 (±0.72) | 271.91 (±5.35) | 1.29 (±0.03) | 4.72 (±0.10) | 2.66 (±0.05) |
| LightFair (Ours) | 0.30 (±0.08) | 0.99 (±0.55) | 30.57 (±0.16) | 80.00 (±0.76) | 233.53 (±5.50) | 1.30 (±0.03) | 4.79 (±0.08) | 2.60 (±0.04) | 0.18 (±0.04) | 1.06 (±0.43) | 31.34 (±0.20) | 86.31 (±0.70) | 259.96 (±7.75) | 1.33 (±0.03) | 4.80 (±0.10) | 2.55 (±0.04) |
| **Stable Diffusion v2.1** | | | | | | | | | | | | | | | | |
| SD [78] | 0.85 (±0.05) | 1.84 (±0.63) | 29.90 (±0.15) | - | 259.36 (±4.81) | 1.23 (±0.03) | 5.12 (±0.05) | 2.24 (±0.03) | 0.63 (±0.01) | 2.06 (±0.35) | 29.90 (±0.15) | - | 259.36 (±4.81) | 1.23 (±0.03) | 5.12 (±0.05) | 2.24 (±0.03) |
| debias VL [15] | 0.43 (±0.09) | 1.44 (±0.48) | 28.20 (±0.22) | 70.01 (±0.96) | 245.11 (±3.72) | 1.35 (±0.03) | 3.53 (±0.11) | 2.93 (±0.06) | 0.49 (±0.03) | 1.91 (±0.92) | 28.15 (±0.26) | 67.42 (±0.96) | 242.78 (±4.21) | 1.33 (±0.03) | 3.57 (±0.11) | 2.85 (±0.06) |
| UCE [27] | 0.90 (±0.04) | 1.67 (±0.71) | 29.41 (±0.13) | 87.94 (±0.86) | 268.52 (±3.92) | 1.22 (±0.02) | 5.12 (±0.05) | 2.32 (±0.03) | 0.50 (±0.03) | 1.95 (±0.37) | 29.44 (±0.12) | 80.46 (±1.13) | 250.57 (±3.49) | 1.23 (±0.03) | 5.17 (±0.08) | 2.25 (±0.03) |
| LightFair (Ours) | 0.33 (±0.10) | 1.40 (±0.28) | 30.82 (±0.19) | 75.29 (±0.99) | 231.46 (±3.30) | 1.35 (±0.02) | 5.14 (±0.09) | 2.24 (±0.06) | 0.40 (±0.03) | 1.82 (±0.44) | 30.26 (±0.16) | 77.47 (±1.05) | 230.59 (±6.53) | 1.35 (±0.01) | 5.29 (±0.11) | 2.14 (±0.06) |

Based on this, **Insight.2** encapsulates our fine-grained exploration of the diffusion generation process.

> **Insight.2.** *The diffusion model generates the main word concept in the early denoising stages and the attribute concepts in the later denoising stages.*

Therefore, as shown in Fig. 3(d), we propose a two-stage text-guided sampling strategy. In the early stages of sampling, when attribute-related information is minimal, the output of the original text encoder continues to provide guidance. In the later stages, the fine-tuned text encoder's output directs the generation of images with fair attributes. Specifically, the noise prediction is expressed as:

$$\epsilon_\theta(P, \mathbf{z}_t, t) = \begin{cases} \epsilon_\theta(f_{orig}^t(P), \mathbf{z}_t, t) \,, t \geq \tau \\ \epsilon_\theta(f_{new}^t(P), \mathbf{z}_t, t) \,, t < \tau \end{cases} \,, \tag{14}$$

where $\tau$ represents the optimal switching time for the text encoder. This strategy introduces almost no computational burden, making it lightweight as well. **Overall, Sec. G provides the pseudo-code for** `LightFair`**.**

## 5 Experiments

### 5.1 Experimental Setups

We apply our method to SD v1.5 and v2.1 to mitigate gender and racial biases. For gender, we consider 'Male' and 'Female' attributes. For racial, we include 'White', 'Black', and 'Asian' attributes. We use the prompt template "Photo portrait of a/an {occupation}, a person", where the occupation is taken from [24]. We generate 100 images per prompt, repeat evaluation 5 times, and report the mean and variance across 2 fairness and 6 quality metrics. We compare our method against 16 recent advances in fair T2I diffusion. Detailed introductions are deferred to Sec. H.

### 5.2 Overall Performance

**Quantitative Analysis.** *Due to space limitations, Tab. 1 presents results for 7 representative quantitative comparisons. The full experimental results, including comparisons with 16 baseline methods and evaluations using 2 additional metrics, are provided in Sec. I.1.* Based on Tab. 1, we draw the following conclusions: First, the Stable Diffusion, whether v1.5 or v2.1, displays strong attribute biases. Specifically, both gender and race biases exceed 0.5, with SD v2.1 exhibiting a particularly high gender bias of 0.85. Second, current debiasing methods provide limited improvement and, in some cases, worsen the biases, as observed with FairD. This may occur because over-correction shifts the model's bias from one attribute to another. Additionally, some methods, while reducing odds bias, negatively affect quality fairness. For example, FinetuneFD lowers Bias-O but increases Bias-Q. Our method focuses on the key structure contributing to attribute bias while preserving quality fairness in the generated content. It successfully debiases multiple versions of SD. For instance, for SD v1.5, it reduces gender/race biases by 0.43/0.36, while ensuring that quality biases are reduced by 0.91/0.54. We note that our method outperforms all competitors in terms of generation quality, except for the CLIP-I metric. Since debiasing alters certain image attributes, a decline in this metric is an expected trade-off. Nonetheless, our method still ranks as the runner-up, demonstrating its effectiveness. Finally, the time and space complexity analysis provided in Fig. 1 highlights the lightweight nature of our `LightFair`.

Table 2: Expansion of diverse prompts in gender-debiased SD.

| Prompt | Method | Stable Diffusion v1.5 | | | Stable Diffusion v2.1 | | |
|---|---|---|---|---|---|---|---|
| | | Bias-O $\downarrow$ | Bias-Q $\downarrow$ | CLIP-T $\uparrow$ | Bias-O $\downarrow$ | Bias-Q $\downarrow$ | CLIP-T $\uparrow$ |
| Non-templated | SD | 0.61 (±0.25) | 1.32 (±0.19) | 32.06 (±1.65) | 0.46 (±0.19) | 1.43 (±0.24) | 32.02 (±2.04) |
| | Ours | **0.48** (±0.27) | **1.02** (±0.15) | **32.62** (±2.02) | **0.34** (±0.23) | **1.13** (±0.15) | **32.77** (±1.99) |
| Two People | SD | 0.35 (±0.04) | 1.23 (±0.23) | 30.46 (±0.14) | 0.65 (±0.03) | 1.76 (±0.22) | 32.32 (±0.17) |
| | Ours | **0.13** (±0.04) | **0.89** (±0.12) | **30.90** (±0.22) | **0.54** (±0.04) | **1.11** (±0.13) | **32.50** (±0.25) |
| Three People | SD | 0.46 (±0.05) | 1.77 (±0.31) | 31.17 (±0.18) | 0.70 (±0.04) | 2.01 (±0.42) | 32.99 (±0.18) |
| | Ours | **0.30** (±0.04) | **1.05** (±0.20) | **32.49** (±0.04) | **0.62** (±0.04) | **1.43** (±0.21) | **33.89** (±0.25) |

Table 3: Ablation study on the effectiveness of different modules.

| $\ell_o$ | $\ell_q$ | $\ell_{reg}$ | AFE | Bias-O $\downarrow$ | Bias-Q $\downarrow$ | CLIP-T $\uparrow$ |
|---|---|---|---|---|---|---|
| | | | | 0.70 (±0.03) | 1.15 (±0.49) | 30.34 (±0.08) |
| ✓ | | | | 0.67 (±0.08) | 1.07 (±0.46) | 30.45 (±0.11) |
| ✓ | ✓ | | | 0.56 (±0.08) | 0.93 (±0.46) | 30.51 (±0.12) |
| ✓ | | ✓ | | 0.49 (±0.06) | 1.12 (±0.73) | 28.38 (±0.11) |
| ✓ | ✓ | ✓ | | 0.45 (±0.09) | 0.88 (±0.51) | 30.29 (±0.44) |
| ✓ | ✓ | ✓ | ✓ | **0.34** (±0.10) | **0.81** (±0.65) | **32.19** (±0.07) |

**Qualitative Analysis.** Fig. 6 presents the qualitative results of our debiased SD. The original SD shows a tendency to generate male CEOs and white doctors, marginalizing other identities. In contrast, our debiased SD significantly improves the representation of minorities while preserving the original image layout and details. Additional qualitative results are provided in Sec. I.2.

**Generalization to diverse prompts.** In Tab. 2, we further explore the effectiveness of our method across a broader range of prompts. For non-templated prompts, we conduct experiments on 30 occupation-related prompts from the LAION-Aesthetics V2 dataset [81] (see Sec. I.3). For scenarios involving multiple people, we consider prompts such as "Photo portrait of two/three {occupation}, two/three people". The results show that our method is equally effective across diverse prompts, demonstrating its scalability. The qualitative results are provided in Sec. I.4.

**Generalization to diverse attributes.** In Sec. I.5 and Sec. I.6, we explore the results of debiasing on the cross-attribute Gender×Race and the Age attribute. The results demonstrate that our method can generalize to other attributes and multi-attribute debiasing scenarios.

**Generalization to diverse target distributions**. In Sec. I.7, we explore the effectiveness of debiasing under imbalanced target distributions. The results demonstrate that our method can adapt to diverse target distributions by tuning only a single hyperparameter.

## 5.3 Ablation Study

We perform several ablation studies to test the effectiveness of different modules and hyperparameters. All experiments are conducted in gender-debiased Stable Diffusion v1.5.

**The Effectiveness of Different Modules.** Tab. 3 presents our step-by-step ablation study on the three loss functions and Adaptive Foreground Extraction (AFE) mechanism for foreground extraction. Compared to the baseline, $\ell_o$ reduces Bias-O, while $\ell_q$ reduces Bias-Q. However, the improvements remain limited due to overfitting. The regularization loss $\ell_{reg}$ prevents the model from deviating excessively from the original semantics, resulting in a reduction of Bias-O and Bias-Q by $0.25$ and $0.27$, respectively. Additionally, AFE enhances semantic information extraction from the foreground, further reducing Bias-O and Bias-Q while maintaining the quality of the generated images. Fig. 4 illustrates qualitative results achieved with AFE. Sec. I.8 provides additional ablation experiments evaluating AFE under complex background conditions.

**Ablation Study on Hyper-Parameters.** Fig. 7a ablates the hyperparameter $\lambda_1$, which sets the weight of $\ell_q$. The sequence $\lambda_1 = 5 \to 2 \to 1 \to 0.5$ forms a smooth downward curve. But $\lambda_1 = 10$ and $0.1$ deviate from the main pattern due to over- and under-regularization. The optimal value is $\lambda_1 = 1$. A small $\lambda_1$ reduces constraints on equalized quality, increasing Bias-Q. In contrast, a large $\lambda_1$ causes model overfitting, worsening Bias-O. Fig. 7b ablates the hyperparameter $\lambda_2$, which adjusts the weight of $\ell_{reg}$. The sequence $\lambda_2 = 1 \to 0.5 \to 0.2 \to 0.1$ creates a consistent slope, while $\lambda_2 = 0.05$ and $0.01$ fall outside the stable range. The best value is $\lambda_2 = 0.1$. If $\lambda_2$ is too small, excessive parameter changes lower the generation quality. If $\lambda_2$ is too large, the model has difficulty converging, reducing the effectiveness of debiasing. For additional ablation studies on hyperparameters, see Sec. I.9.

**Different $\tau$ Values During Inference.** Fig. 8 illustrates the effect of different $\tau$ values during sampling. Since the early denoising stages primarily capture non-attribute information, choosing $\tau \in (\frac{3}{4}T, T]$ ensures a lower Bias-O. In contrast, during the later stages, attribute features have already formed and are difficult to reverse, resulting in Bias-O values comparable to those observed without the debias model. However, intervening too early can degrade the quality of generated images, as evidenced by irrelevant semantic artifacts (*e.g.*, extraneous objects in the bottom-right corner of the image in Fig. 8(a)). Based on experimental results, we recommend setting $\tau = \frac{3}{4}T$.

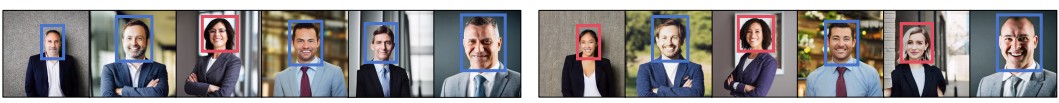

(a) Prompt: "Photo portrait of a **CEO**, a person". Left: original SD v1.5. Right: our **gender**-debiased **SD v1.5**.

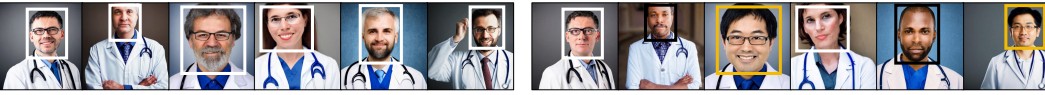

(b) Prompt: "Photo portrait of a **doctor**, a person". Left: original SD v2.1. Right: our **race**-debiased **SD v2.1**.

Figure 6: Qualitative results. Images generated by the original SD (left) and our debiased SD (right). For the same prompt, the images in corresponding positions are generated using the same random noise. Bounding boxes denote detected faces (Gender: Male, Female; Race: White, Asian, Black). More images are provided in Fig. 12, Fig. 13, Fig. 14 and Fig. 15.

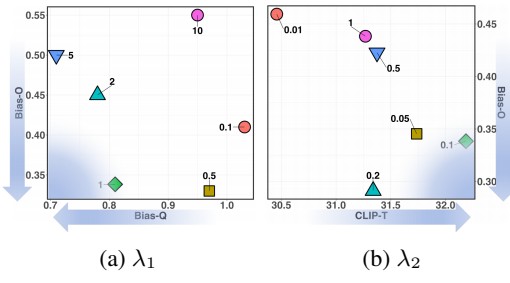

(a) $\lambda_1$       (b) $\lambda_2$

Figure 7: Ablation Study on Hyper-Parameters.

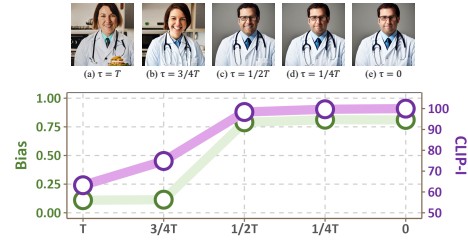

Figure 8: Visualization and performance of image generation with different $\tau$ values.

## 5.4 Further Exploration

In Sec. I.10, we show that `LightFair` can serve as a plug-in alongside other debiasing methods. Sec. I.11 further shows that `LightFair` is equally applicable to SD models built on the DiT architecture. Sec. I.12 presents user studies indicating that our method delivers a superior user experience. Sec. I.13 verifies that our debiasing preserves the model's semantic understanding of original attributes. Moreover, Sec. I.14 confirms it does not affect generation on general prompts.

## 6 Conclusion

This paper explores a novel lightweight approach, named `LightFair`, to achieve fairness in T2I DMs. First, we reveal the text encoder's adverse effects on fairness. Then, we propose a collaborative distance-constrained debiasing strategy that achieves equalized odds and equalized quality without relying on auxiliary networks. Next, we introduce a two-stage text-guided sampling strategy. It applies the debiased text encoder only during later sampling stages, which preserves the original model's fidelity. Finally, extensive experiments confirm the effectiveness of our `LightFair`.

## Acknowledgments

This work was supported in part by National Natural Science Foundation of China: 62525212, 62236008, 62441232, U21B2038, U23B2051, 62502496, 62206264 and 92370102, in part by Youth Innovation Promotion Association CAS, in part by the Strategic Priority Research Program of the Chinese Academy of Sciences, Grant No. XDB0680201, in part by the China Postdoctoral Science Foundation (CPSF) under Grant No.2025M771492, and in part by the Postdoctoral Fellowship Program of CPSF under Grant No. GZB20240729.

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

# Contents

## A   Symbol Definitions

In this section, Tab. 4 includes a summary of key notations and descriptions in this work.

Table 4: A summary of key notations and descriptions in this work.

| Notations | Descriptions |
|---|---|
| $g^e$ | Image encoder of the latent diffusion model. |
| $g^d$ | Image decoder of the latent diffusion model. |
| $\mathbf{x}_0, \cdots, \mathbf{x}_T$ | Samples of the diffusion model at $t = 0, \cdots, T$. |
| $\mathbf{z}_0, \cdots, \mathbf{z}_T$ | Latent space samples of the diffusion model at $t = 0, \cdots, T$. |
| $\alpha_t$ | Hyperparameter controlling the noise level, $\alpha_t \in (0, 1)$. |
| $\overline{\alpha_t}$ | $\overline{\alpha_t} = \prod_{i=1}^{t} \alpha_i$. |
| $\beta_t$ | $\beta_t = 1 - \alpha_t$. |
| $f^t$ | Text encoder of diffusion model / Text encoder of CLIP. |
| $f^i$ | Image encoder of CLIP. |
| $prompt(\cdot)$ / $P(\cdot)$ | Prompt used for text-to-image generation. |
| $\epsilon_\theta(f^t(P), \mathbf{z}_t, t)$ | Noise prediction network of the text-to-image diffusion model. |
| $A$ | Attribute set. |
| $C$ | Main word set. |
| $a$ | Attribute word from the set $A$, $a \in A$. |
| $c$ | Main word from the set $C$, $c \in C$. |
| $M(prompt(\cdot))$ | Images generated using $prompt(\cdot)$. |
| $\mathrm{emb}_c^T(\cdot)$ | Text embedding, shorthand for $f^t(prompt(\cdot, c))$. |
| $\mathrm{emb}_c^i(\cdot)$ | Image embedding, shorthand for $f^i(M(prompt(\cdot, c)))$. |
| $\mathbb{E}[\mathrm{emb}_c^i(\cdot)]$ | Centroid of the image embeddings. |
| $\tau$ | The optimal starting point for fine-tuning the text encoder. |

## B   Parameter Counts of Different Components in Stable Diffusion

Tab. 5 presents the parameter counts of different components in Stable Diffusion, including the CLIP text encoder and U-Net. The results indicate that U-Net has more parameters than the CLIP text encoder. Therefore, fine-tuning the text encoder alone is a lightweight approach.

Table 5: Parameter Counts of Different Components in Stable Diffusion.

| Method | CLIP Text Encoder | U-Net |
|---|---|---|
| Stable Diffusion v1.5 | 123.060480 M | 859.520964 M |
| Stable Diffusion v2.1 | 340.387840 M | 865.910724 M |

## C   Occupation List

We obtain the following 80 occupations from [24], which are used for plotting Fig. 2(c).

[*'aerospace engineer'*, *'author'*, *'baker'*, *'bartender'*, *'butcher'*, *'carpenter'*, *'ceo'*, *'childcare worker'*, *'claims appraiser'*, *'cleaner'*, *'coach'*, *'compliance officer'*, *'computer programmer'*, *'computer support specialist'*, *'computer systems analyst'*, *'construction worker'*, *'cook'*, *'correctional officer'*, *'dentist'*, *'designer'*, *'detective'*, *'director'*, *'dispatcher'*, *'doctor'*, *'drywall installer'*, *'electrical engineer'*, *'electrician'*, *'engineer'*, *'event planner'*, *'facilities manager'*, *'file clerk'*, *'financial manager'*, *'firefighter'*, *'head cook'*, *'health technician'*, *'hostess'*, *'industrial engineer'*, *'inventory clerk'*, *'it specialist'*, *'janitor'*, *'lawyer'*, *'logistician'*, *'machinery mechanic'*, *'machinist'*, *'manicurist'*, *'massage therapist'*, *'mechanical engineer'*, *'medical records specialist'*, *'mover'*, *'musician'*, *'network administrator'*, *'nurse'*, *'occupational therapist'*, *'office clerk'*, *'painter'*, *'pilot'*, *'plane mechanic'*, *'plumber'*, *'police officer'*, *'postal worker'*, *'printing press operator'*, *'producer'*, *'programmer'*, *'radiologic technician'*, *'real estate broker'*, *'repair worker'*, *'roofer'*, *'sales manager'*, *'salesperson'*, *'school bus driver'*, *'security guard'*, *'social assistant'*, *'software developer'*, *'supervisor'*, *'teacher'*, *'teaching assistant'*, *'waiter'*, *'web developer'*, *'wholesale buyer'*, *'writer'*]

# D  Proof of Theorem 4.1

**Assumption D.1** (Stochastic Neighbor Embedding [36]). Let $i$ represent an object and $j$ a potential neighbor. The probability $p_{ij}$ that object $i$ selects $j$ as its neighbor is defined as:

$$p_{ij} = \frac{\exp(-d_{ij}^2)}{\sum_{k \neq i} \exp(-d_{ik}^2)}, \tag{1}$$

where the dissimilarities $d_{ij}^2$ are calculated using the scaled squared Euclidean distance between two high-dimensional points $\mathbf{x}_i$ and $\mathbf{x}_j$:

$$d_{ij}^2 = \frac{\|\mathbf{x}_i - \mathbf{x}_j\|^2}{2\sigma_i^2}, \tag{2}$$

and $\sigma_i$ represents the variance parameter associated with object $i$.

**Assumption D.2.** The diffusion model is assumed to be well-trained, such that it can correctly generate the content specified by the prompt:

$$\mathbb{P}\big(x \mid prompt(x)\big) = 1. \tag{15}$$

**Assumption D.3.** The attribute $a$ and the concept $c$ are assumed to be statistically independent, that is:

$$\mathbb{P}(a, c) = \mathbb{P}(a)\mathbb{P}(c). \tag{16}$$

This implies that the attribute $a$ does not provide additional information about the concept $c$, and vice versa.

**Restate of Theorem 4.1.** *Under Thm. D.1, D.2 and D.3, for any attributes $a_i, a_j \in A$, achieving Equalized Odds $\mathbb{P}(a_i|P(\cdot, c)) = \mathbb{P}(a_j|P(\cdot, c))$ is equivalent to ensuring Equalized Distance:*

$$\big\|f(a_i, c) - f^t\big(P(\cdot, c)\big)\big\|^2 = \big\|f(a_j, c) - f^t\big(P(\cdot, c)\big)\big\|^2,$$

*where $f(a_i, c)$ represents the encoding of the concepts $a_i$ and $c$, $f^t(\cdot)$ denotes the encoding performed by the CLIP text encoder and $P(\cdot)$ is shorthand for $prompt(\cdot)$.*

*Proof.* First, since the input of the diffusion denoising process is the encoding of the prompt by the text encoder, **Equalized Odds** can be reformulated as:

$$\mathbb{P}\Big(a_i \mid f^t\big(P(\cdot, c)\big)\Big) = \mathbb{P}\Big(a_j \mid f^t\big(P(\cdot, c)\big)\Big), \quad \forall a_i, a_j \in A, \tag{17}$$

where $f^t(\cdot)$ denotes encoding by the CLIP text encoder, and $P(\cdot)$ is shorthand for $prompt(\cdot)$.

According to Thm. D.2, we have:

$$\mathbb{P}\Big(c \mid f^t\big(P(\cdot, c)\big)\Big) = 1. \tag{18}$$

Thus, by Thm. D.3, for any $a_i, a_j \in A$,

$$\mathbb{P}\Big(a_i, c \mid f^t\big(P(\cdot, c)\big)\Big) = \mathbb{P}\Big(a_j, c \mid f^t\big(P(\cdot, c)\big)\Big). \tag{19}$$

Let $f(a, c)$ represent the effective encoding of the concepts $a$ and $c$, Equ. (19) can be approximated as:

$$\mathbb{P}\Big(f(a_i, c) \mid f^t\big(P(\cdot, c)\big)\Big) = \mathbb{P}\Big(f(a_j, c) \mid f^t\big(P(\cdot, c)\big)\Big). \tag{20}$$

According to Thm. D.1 and [20], the conditional probability $\mathbb{P}\Big(f(a_i, c) \mid f^t\big(P(\cdot, c)\big)\Big)$ can be represented as the similarity between $f(a_i, c)$ and $f^t\big(P(\cdot, c)\big)$, and can be modeled using a Gaussian distribution. We thus measuring $\mathbb{P}\Big(f(a_i, c) \mid f^t\big(P(\cdot, c)\big)\Big)$ by calculating:

$$\mathbb{P}\Big(f(a_i, c) \mid f^t\big(P(\cdot, c)\big)\Big) = \frac{\exp\left(-\frac{\left\|f(a_i, c) - f^t\big(P(\cdot, c)\big)\right\|^2}{2\rho^2}\right)}{\sum_{a_k \in A} \exp\left(-\frac{\left\|f(a_k, c) - f^t\big(P(\cdot, c)\big)\right\|^2}{2\rho^2}\right)}, \tag{21}$$

where $\rho$ is a constant dependent only on $f^t\big(P(\cdot,c)\big)$, controlling the falloff of $\mathbb{P}$ with respect to distance. Combining Equ. (20) and Equ. (21), we obtain:

$$\frac{\exp\left(-\frac{\left\|f(a_i,c)-f^t\big(P(\cdot,c)\big)\right\|^2}{2\rho^2}\right)}{\sum_{a_k\in A}\exp\left(-\frac{\left\|f(a_k,c)-f^t\big(P(\cdot,c)\big)\right\|^2}{2\rho^2}\right)}=\frac{\exp\left(-\frac{\left\|f(a_j,c)-f^t\big(P(\cdot,c)\big)\right\|^2}{2\rho^2}\right)}{\sum_{a_k\in A}\exp\left(-\frac{\left\|f(a_k,c)-f^t\big(P(\cdot,c)\big)\right\|^2}{2\rho^2}\right)} \tag{22}$$

$$\exp\left(-\frac{\left\|f(a_i,c)-f^t\big(P(\cdot,c)\big)\right\|^2}{2\rho^2}\right)=\exp\left(-\frac{\left\|f(a_j,c)-f^t\big(P(\cdot,c)\big)\right\|^2}{2\rho^2}\right) \tag{23}$$

$$\left\|f(a_i,c)-f^t\big(P(\cdot,c)\big)\right\|^2=\left\|f(a_j,c)-f^t\big(P(\cdot,c)\big)\right\|^2 \tag{24}$$

This completed the proof. $\qquad\square$

To establish Thm. 4.1, we rely on an independence assumption (Thm. D.3). This is a mild requirement that can be validated on both **empirical** and **theoretical** grounds.

- **Empirically**, the assumption is weak and specific to the training process. Although attributes and concepts are seldom independent in the real world, the training images are generated by Stable Diffusion. By controlling the prompts, we can readily enforce independence between attributes and concepts. For example, we generate equal numbers of images for different attributes to compute semantic centroids. In such a controlled setting, the condition $\mathbb{P}(a,c)=\mathbb{P}(a)\mathbb{P}(c)$ clearly holds.

- **Theoretically**, we further relax Thm. D.3 to a softer condition: $1-\epsilon\leq\frac{\mathbb{P}(a|c)}{\mathbb{P}(a)}\leq 1+\epsilon$ (Thm. D.4), and derive Thm. D.5. It shows that, under this relaxed assumption, the induced probability error from the distance constraint is on the order of $O(\epsilon)$, where $\epsilon$ is a small constant. In our training data, $\epsilon$ is always less than 0.01.

**Assumption D.4.** For any attribute $a$ and concept $c$, there exists $\epsilon\in[0,1)$ such that

$$1-\epsilon\leq\frac{P(a\mid c)}{P(a)}\leq 1+\epsilon,\quad\forall a\in A,\ c\in C. \tag{25}$$

When $\epsilon=0$, this reduces to the original Thm. D.3.

**Theorem D.5.** *Under Thm. D.1, D.2 and D.4, let $d_i=\left\|f(a_i,c)-f^t\big(P(\cdot,c)\big)\right\|$, and let $\rho$ be the bandwidth of the Gaussian kernel. If Equalized Odds holds, i.e.,*

$$\mathbb{P}(a_i\mid P(\cdot,c))=\mathbb{P}(a_j\mid P(\cdot,c))\quad\text{for any }a_i,a_j\in A, \tag{26}$$

*then the corresponding embedding distances satisfy*

$$\left|d_i^2-d_j^2\right|\leq 2\rho^2\cdot\left|\ln\frac{1+\epsilon}{1-\epsilon}\right|\approx 4\rho^2\epsilon. \tag{27}$$

*In particular, as $\epsilon\to 0$, the bound vanishes. Equalized Odds then implies exact equality in embedding distances, recovering the original Thm. 4.1.*

*Proof.* First, since the input of the diffusion denoising process is the encoding of the prompt by the text encoder, **Equalized Odds** can be reformulated as:

$$\mathbb{P}\Big(a_i\mid f^t\big(P(\cdot,c)\big)\Big)=\mathbb{P}\Big(a_j\mid f^t\big(P(\cdot,c)\big)\Big),\quad\forall a_i,a_j\in A, \tag{28}$$

where $f^t(\cdot)$ denotes encoding by the CLIP text encoder, and $P(\cdot)$ is shorthand for $prompt(\cdot)$.

According to Thm. D.2, we have:

$$\mathbb{P}\Big(c\mid f^t\big(P(\cdot,c)\big)\Big)=1. \tag{29}$$

Under the **weak-independence** Thm. D.4, for any $a_i, a_j \in A$,

$$\frac{1-\epsilon}{1+\epsilon} \leq \frac{\mathbb{P}(a_i, c \mid f^t(P(\cdot, c)))}{\mathbb{P}(a_j, c \mid f^t(P(\cdot, c)))} \leq \frac{1+\epsilon}{1-\epsilon}. \tag{30}$$

Let $f(a, c)$ represent the joint encoding of the attribute $a$ and concept $c$.

Because $f(a, c)$ captures both $a$ and $c$, Equ. (30) can be rewritten as

$$\frac{1-\epsilon}{1+\epsilon} \leq \frac{\mathbb{P}(f(a_i, c) \mid f^t(P(\cdot, c)))}{\mathbb{P}(f(a_j, c) \mid f^t(P(\cdot, c)))} \leq \frac{1+\epsilon}{1-\epsilon}. \tag{31}$$

According to Thm. D.1 and [20], the conditional probability $\mathbb{P}\big(f(a_i, c) \mid f^t(P(\cdot, c))\big)$ can be represented as the similarity between $f(a_i, c)$ and $f^t(P(\cdot, c))$, and can be modeled using a Gaussian distribution. We thus measuring $\mathbb{P}\big(f(a_i, c) \mid f^t(P(\cdot, c))\big)$ by calculating:

$$\mathbb{P}\big(f(a_i, c) \mid f^t(P(\cdot, c))\big) = \frac{\exp\left(-\frac{\left\|f(a_i, c) - f^t(P(\cdot, c))\right\|^2}{2\rho^2}\right)}{\sum_{a_k \in A} \exp\left(-\frac{\left\|f(a_k, c) - f^t(P(\cdot, c))\right\|^2}{2\rho^2}\right)}, \tag{32}$$

where $\rho$ is a constant dependent only on $f^t(P(\cdot, c))$, controlling the falloff of $\mathbb{P}$ with respect to distance. Combining Equ. (31) and Equ. (32), we obtain:

$$\frac{1-\epsilon}{1+\epsilon} \leq \exp\left(-\frac{\|f(a_i, c) - f^t(P(\cdot, c))\|^2 - \|f(a_j, c) - f^t(P(\cdot, c))\|^2}{2\rho^2}\right) \leq \frac{1+\epsilon}{1-\epsilon}. \tag{33}$$

Taking natural logarithms and absolute values on Equ. (33) yields

$$\left| \|f(a_i, c) - f^t(P(\cdot, c))\|^2 - \|f(a_j, c) - f^t(P(\cdot, c))\|^2 \right| \leq 2\rho^2 \left| \ln \frac{1+\epsilon}{1-\epsilon} \right|. \tag{34}$$

Recalling the definition $d_i = \|f(a_i, c) - f^t(P(\cdot, c))\|$ and using $\ln \frac{1+\epsilon}{1-\epsilon} = 2\epsilon + O(\epsilon^3)$, we have

$$\left| d_i^2 - d_j^2 \right| \leq 2\rho^2 \left| \ln \frac{1+\epsilon}{1-\epsilon} \right| \approx 4\rho^2 \epsilon. \tag{35}$$

As $\epsilon \to 0$, the logarithmic term vanishes, so Equalized Odds enforces $d_i^2 = d_j^2$, recovering the exact equality of embedding distances established under the stronger independence assumption.

This completed the proof. $\qquad\square$

## E    Proof of Proposition 4.2

**Definition E.1** (Fourier Transform). The Fourier Transform of a function $f(x)$, denoted as $\mathcal{F}\{f(x)\}$, is defined as:

$$\mathcal{F}[f(x)](\omega) = F_x(\omega) = \int_{-\infty}^{\infty} f(x) e^{-i\omega x} \, dx, \tag{36}$$

where $\omega$ is the angular frequency variable in the Fourier domain.

**Lemma E.2** (Linearity Property of the Fourier Transform). *If the Fourier transforms of signals $f_1(x)$ and $f_2(x)$ are $F_{1x}(\omega)$ and $F_{2x}(\omega)$, respectively, i.e.,*

$$\mathcal{F}[f_1(x)](\omega) = F_{1x}(\omega),$$
$$\mathcal{F}[f_2(x)](\omega) = F_{2x}(\omega),$$

*then for any constants $a_1$ and $a_2$, the Fourier Transform satisfies:*

$$\mathcal{F}[a_1 f_1(t) + a_2 f_2(t)](\omega) = a_1 F_{1x}(\omega) + a_2 F_{2x}(\omega). \tag{37}$$

**Restate of Proposition 4.2.** *The recovery rate of low-frequency signals during the diffusion denoising process is higher than that of high-frequency signals.*

*Proof.* For the forward noising process of the diffusion model Denoising Diffusion Probabilistic Model (DDPM), we have

$$\mathbf{x}_t = \sqrt{\overline{\alpha_t}}\mathbf{x}_0 + \sqrt{1 - \overline{\alpha_t}}\epsilon \quad \text{with} \quad \epsilon \sim \mathcal{N}(\epsilon; 0, I), \tag{38}$$

where, $\overline{\alpha_t} = \prod_{i=1}^{t}\alpha_i$, $\alpha_t \in (0, 1)$ represents the noise attenuation factor at time $t$ during the diffusion process. $\mathbf{x}_0$ and $\mathbf{x}_t$ denote the initial noise-free sample and the noisy sample at time $t$, respectively. $\epsilon$ represents standard Gaussian noise.

According to Thm. E.1, applying the Fourier Transform to Equ. (38) yields:

$$\mathcal{F}\left[\mathbf{x}_t\right](\omega) = F_t(\omega) = \mathcal{F}\left[\sqrt{\overline{\alpha_t}}\mathbf{x}_0 + \sqrt{1 - \overline{\alpha_t}}\epsilon\right](\omega). \tag{39}$$

Next, due to the linearity property of the Fourier Transform in Thm. E.2, we have:

$$\mathcal{F}\left[\mathbf{x}_t\right](\omega) = \sqrt{\overline{\alpha_t}}\mathcal{F}\left[\mathbf{x}_0\right](\omega) + \sqrt{1 - \overline{\alpha_t}}\mathcal{F}\left[\epsilon\right](\omega),$$

$$F_t(\omega) = \sqrt{\overline{\alpha_t}}F_0(\omega) + \sqrt{1 - \overline{\alpha_t}}F_\epsilon(\omega). \tag{40}$$

Substituting $f = 2\pi\omega$, we obtain:

$$F_t(f) = \sqrt{\overline{\alpha_t}}F_0(f) + \sqrt{1 - \overline{\alpha_t}}F_\epsilon(f). \tag{41}$$

The DDPM denoising process can be viewed as an error-free transmission of image signals through a channel. The original noise-free image signal is the sum of image signals at different frequencies, expressed as $\mathbf{x}_0 = \sum_{f=0}^{+\infty} F_{\mathbf{x}_0}(f)$. The channel input is a combination of the attenuated image signal ($\sqrt{\overline{\alpha_T}}F_{\mathbf{x}_0}(f)$) and Gaussian noise ($\sqrt{1 - \overline{\alpha_T}}F_\epsilon(f)$). Due to the sufficiently large Gaussian noise, the image signal is masked, and the input can be considered as Gaussian noise, which corresponds to the random noise at timestep $\mathbf{x}_T$ in the DDPM reverse process, *i.e.*:

**Attenuated Image Signal**

$$F_T(f) = \boxed{\sqrt{\overline{\alpha_T}}F_0(f)} + \boxed{\sqrt{1 - \overline{\alpha_T}}F_\epsilon(f)}. \tag{42}$$

**Gaussian Noise**

For a diffusion model to successfully reconstruct an image, it must ensure that the attenuated image signal is completely transmitted. Simultaneously, during the transmission process, the DDPM weakens the original Gaussian noise through noise prediction. This process can be described as:

$$\boxed{\text{Image Signal}} + \boxed{\text{High Gaussian Noise}} \xrightarrow[\text{(DDPM)}]{\text{Channel Transmission}} \boxed{\text{Image Signal}} + \boxed{\text{Low Gaussian Noise}}.$$

Assuming the DDPM model is fully trained at any given time $t$, it can completely remove the noise. During this process, the signal-to-noise ratio at time $t$ is:

$$\text{SNR}_t(f) = \frac{\mathbb{E}\left[\left|F_T(f) - E_T(f)\right|^2\right]}{\mathbb{E}\left[\left|E_t(f)\right|^2\right]} \tag{43}$$

$$= \frac{\mathbb{E}\left[\left|F_T(f) - \sqrt{1 - \overline{\alpha_T}}F_\epsilon(f)\right|^2\right]}{\mathbb{E}\left[\left|\sqrt{1 - \overline{\alpha_t}}F_\epsilon(f)\right|^2\right]} \tag{44}$$

$$= \frac{\overline{\alpha_T}\left|F_0(f)\right|^2}{(1 - \overline{\alpha_t})\left|F_\epsilon(f)\right|^2}. \tag{45}$$

The variation efficiency of the signal-to-noise ratio:

$$\Delta\text{SNR}_i^{i+1}(f) = \text{SNR}_i(f) - \text{SNR}_{i+1}(f) \tag{46}$$

$$= \frac{\overline{\alpha_T}\,|F_0(f)|^2}{(1-\overline{\alpha_i})\,|F_\epsilon(f)|^2} - \frac{\overline{\alpha_T}\,|F_0(f)|^2}{(1-\overline{\alpha_{i+1}})\,|F_\epsilon(f)|^2} \tag{47}$$

$$= \frac{\overline{\alpha_i} - \overline{\alpha_{i+1}}}{(1-\overline{\alpha_i})(1-\overline{\alpha_{i+1}})} \cdot \frac{\overline{\alpha_T}\,|F_0(f)|^2}{|F_\epsilon(f)|^2} \tag{48}$$

Previous studies [12, 22, 23, 89, 90] have observed that the average power spectrum of natural images follows the form $1/f^\beta$ with $\beta \sim 2$. Therefore, we have:

$$|F_0(f)|^2 \propto \frac{1}{f^\beta}, \quad \beta \sim 2. \tag{49}$$

Since $\epsilon \sim \mathcal{N}(\epsilon; 0, I)$ is Gaussian white noise, the power is constant across different frequencies $f$. It can be expressed as:

$$|F_\epsilon(f)|^2 = C. \tag{50}$$

Substituting Equ. (49) and Equ. (50) into Equ. (48) yields:

$$\Delta\text{SNR}_i^{i+1}(f) \propto \underbrace{\frac{\overline{\alpha_i} - \overline{\alpha_{i+1}}}{(1-\overline{\alpha_i})(1-\overline{\alpha_{i+1}})}}_{(1)} \cdot \underbrace{\frac{\overline{\alpha_T}}{C}}_{(2)} \cdot \underbrace{\frac{1}{f^\beta}}_{(3)} \tag{51}$$

Since $\overline{\alpha_t} = \prod_{i=1}^{t} \alpha_i$ and $\alpha_t \in (0, 1]$, part (1) is always positive for any $i$. Part (2) is a constant, and part (3) is a positive term inversely proportional to $f$. Therefore:

- For any $i \in [0, T-1]$, $\Delta\text{SNR}_i^{i+1}(f) > 0$.

- For $f_1 > f_2$, we have $\Delta\text{SNR}_i^{i+1}(f_1) < \Delta\text{SNR}_i^{i+1}(f_2)$.

So, the recovery rate of low-frequency signals during the diffusion denoising process is higher than that of high-frequency signals.

This completed the proof.

$\square$

# F Expanded Version of Filtering Results in the Denoising Process

Here, we provide an expanded version of the filtering results in the denoising process, as shown in Fig. 9, Fig. 10 and Fig. 11.

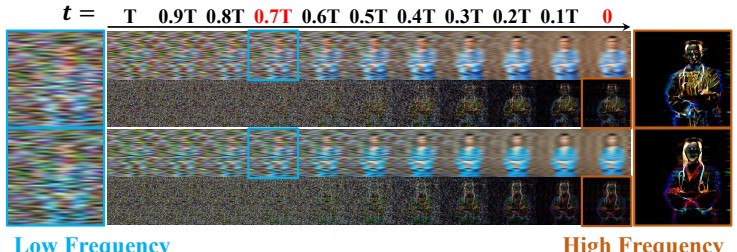

Figure 9: **An enlarged version of Fig. 5.** Results of **low-pass and high-pass filtering** applied during the denoising process with text guidance for 'male doctor' (top two rows) and 'female doctor' (bottom two rows). Each pair shows low-pass filtered images on top and high-pass filtered images below. For clarity, some images are enlarged and highlighted on both sides.

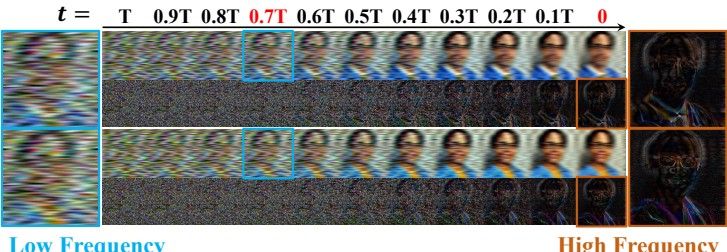

Figure 10: Expanded results of **low-pass and high-pass filtering** applied during the denoising process with text guidance for 'male doctor' (top two rows) and 'female doctor' (bottom two rows). Each pair shows low-pass filtered images on top and high-pass filtered images below. For clarity, some images are enlarged and highlighted on both sides.

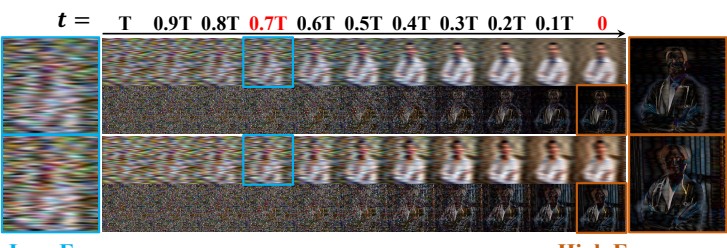

Figure 11: Expanded results of **low-pass and high-pass filtering** applied during the denoising process with text guidance for 'male doctor' (top two rows) and 'female doctor' (bottom two rows). Each pair shows low-pass filtered images on top and high-pass filtered images below. For clarity, some images are enlarged and highlighted on both sides.

# G  Algorithm of `LightFair`

---

**Algorithm 1:** Training

---

**Input:** CLIP text encoder $f_{orig}^{text}$; CLIP image encoder $f^{image}$; U-Net $\epsilon_\theta$; main word $c$; attribute set $A = \{a_1, \dots, a_{|A|}\}$; training epochs $E_{total}$; batch size $N_b$

**Output:** Fine-tuned text encoder $f_{new}^{text}$

1:  $f_{new}^{text} \leftarrow f_{orig}^{text}$;
2:  **for** $epoch = 1$ **to** $E_{total}$ **do**
3:      ▷ Generate Training Images
4:      **with** `no_grad()` **do**
5:          $SD \leftarrow (f_{new}^{text}, \epsilon_\theta)$;                    `# Build SD using text encoder and U-Net`
6:          **for** $i = 1$ **to** $|A|$ **do**
7:              $image_i = [SD(P(a_i, c))]_{N_b}$;                `# Generate images with attributes`
8:          **end for**
9:      **end with**
10:     ▷ Extracting Foreground
11:     **for** $i = 1$ **to** $|A|$ **do**
12:         $image_i' = CA(image_i, P(a_i, c))$;                    `# Extracting foreground using cross-attention`
13:     **end for**
14:     ▷ Calculating Loss Function and Optimization
15:     $images \leftarrow [image_1', \dots, image_{|A|}']$;
16:     $texts \leftarrow [P(\cdot, c), P(a_1, c), \dots, P(a_{|A|}, c)]$;
17:     **with** `no_grad()` **do**
18:         $image\_emb = f^{image}(images).norm$;
19:     **end with**
20:     $image\_emb\_centroid \leftarrow [\mathbb{E}[image\_emb], \mathbb{E}[image\_emb_1], \dots, \mathbb{E}[image\_emb_{|A|}]]$;
21:     $text\_emb = f_{new}^{text}(texts).norm$;
22:     $s = image\_emb\_centroid \cdot text\_emb^T$;                `# Calculate the similarity matrix`
23:     $\ell_o \leftarrow s[1 : |A|, 0]$;
24:     $\ell_q \leftarrow s[k, k], k \in [1, |A|]$;
25:     $\ell_{reg} \leftarrow s[0, 0]$;
26:     Calculate $\ell = \ell_o + \lambda_1 \ell_q + \lambda_2 \ell_{reg}$ with Equ. (9), Equ. (10), and Equ. (11);
27:     Backpropagation updates $f_{new}^{text}$ parameters;
28: **end for**
29: **return** $f_{new}^{text}$.

---

**Algorithm 2:** Sampling

---

**Input:** CLIP original text encoder $f_{orig}^{text}$; CLIP fine-tuned text encoder $f_{new}^{text}$; U-Net $\epsilon_\theta$; prompt $P$; Stable Diffusion image decoder $g^d$; hyperparameters $\alpha_t$, $\sigma_t$ and $\tau$

**Output:** Clean Image $\mathbf{x}_0$

1:  $\mathbf{z}_T \sim \mathcal{N}(0, I)$;
2:  **for** $t = T$ **to** $1$ **do**
3:      $\epsilon_t \sim \mathcal{N}(0, I)$ if $t > 1$, else $\epsilon_t = 0$;
4:      **if** $T \geq \tau$ **then**
5:          $\mathbf{z}_{t-1} = \frac{1}{\sqrt{\alpha_t}}(\mathbf{z}_t - \frac{1-\alpha_t}{\sqrt{1-\bar{\alpha}_t}}\epsilon_\theta(f_{orig}^{text}(P), \mathbf{z}_t, t)) + \sigma_t \epsilon_t$;
6:      **else**
7:          $\mathbf{z}_{t-1} = \frac{1}{\sqrt{\alpha_t}}(\mathbf{z}_t - \frac{1-\alpha_t}{\sqrt{1-\bar{\alpha}_t}}\epsilon_\theta(f_{new}^{text}(P), \mathbf{z}_t, t)) + \sigma_t \epsilon_t$;
8:      **end if**
9:  **end for**
10: $\mathbf{x}_0 \leftarrow g^d(\mathbf{z}_0)$;
11: **return** $\mathbf{x}_0$.

# H Additional Experimental Settings

In this section, we make a supplementation to Sec. 5.1.

## H.1 Evaluation metrics

We use the gender and race classifiers from [80] for evaluation. For each prompt, 100 images are generated. To reduce experimental randomness, the evaluation is repeated five times, and the mean and variance of the reported metrics are calculated. Here we give a more detailed summary of the evaluation metrics mentioned in the experiments.

To evaluate the **Fairness** of the model, we use the following two metrics:

- **Bias-Odds** (Bias-O) quantifies the degree of bias in the frequency of different attributes in generated images when unspecified attributes are involved. Specifically, for a given prompt $prompt(\cdot, c)$, Bias-O is calculated as Bias-O$(prompt(\cdot, c))$ = $\frac{1}{|A|(|A|-1)/2} \sum_{a_i, a_j \in |A|: i < j} |\mathsf{freq}(a_i) - \mathsf{freq}(a_j)|$, where $\mathsf{freq}(a_i)$ represents the frequency of attribute $a_i$ appearing in the generated images, and $|A|$ denotes the number of elements in the attribute set.

- **Bias-Quality** (Bias-Q) quantifies the degree of bias in the generation quality of different attributes in the generated images when unspecified attributes are involved. Specifically, for a given prompt $prompt(\cdot, c)$, Bias-Q is calculated as Bias-Q$(prompt(\cdot, c))$ = $\frac{1}{|A|(|A|-1)/2} \sum_{a_i, a_j \in |A|: i < j} |\mathsf{qual}(a_i) - \mathsf{qual}(a_j)|$, where $\mathsf{qual}(a_i)$ represents the generation quality of images containing attribute $a_i$, calculated using the CLIP Score (The calculation is the same as that in the following CLIP-T.).

To evaluate the **Quality** of the model, we use the following six metrics:

- **CLIP-T** (CLIP Score of Text) [34] measures the semantic alignment between generated images and their corresponding textual descriptions. By calculating the similarity between image and text embeddings, it evaluates their semantic relevance. In our experiments, the *clip-vit-large-patch14* [2] model is used to compute this metric. A higher CLIP-T reflects better alignment between images and descriptions.

- **CLIP-I** (CLIP Score of Image) [34] measures the similarity between the generated image and the original Stable Diffusion (SD) output for the same prompt and noise. By calculating the similarity between image embeddings, it evaluates the consistency of the generated image relative to the original generation. Similar to CLIP-T, the *clip-vit-large-patch14* model is used to compute this metric. A higher CLIP-I indicates a smaller impact of fine-tuning on the model's performance.

- **FID** (Fréchet Inception Distance) [35] assesses the quality of generated images by comparing the distribution of generated images with that of real images in the feature space. In our experiments, the FairFace dataset [46] is used as a reference. A lower FID score indicates that the generated images are closer to the real images, reflecting higher realism and visual quality.

- **IS** (Inception Score) [79] evaluates the quality and diversity of generated images by analyzing the prediction distribution of a classifier, such as the Inception network, on the generated images. A higher IS value indicates better image quality and greater diversity in content.

- **AS-R** (Aesthetic Score - Rating) [99] evaluates the overall aesthetic quality of a given image. The metric is computed using a regression head trained on human-rated aesthetic datasets, and it takes the pooled visual features from a CLIP-based vision encoder as input. A higher AS-R score indicates better visual appeal and artistic quality.

- **AS-A** (Aesthetic Score - Artifacts) [94] quantifies the presence of visual artifacts or distortions in an image. It is computed using a separate regression model trained. A lower AS-A score reflects fewer artifacts and higher perceptual quality.

---

[2] https://huggingface.co/openai/clip-vit-large-patch14

## H.2 Competitors

Here we give a more detailed summary of the competitors mentioned in the experiments.

- **SD** (Stable Diffusion) [78] is an efficient latent space diffusion model capable of generating high-quality images. By mapping the image generation process to a lower-dimensional latent space, it significantly reduces computational costs. In this paper, we consider versions v1.5 [3] and v2.1 [4].

- **FairD** (Fair Diffusion) [24] reduces bias or unfairness in generated images by randomly incorporating additional text prompts to adjust the generation process. The original paper conducts experiments based on SDv1.5.

- **UCE** (Unified Concept Editing) [27] provides a universal framework for concept editing, enabling the effective removal, modification, or replacement of specific concepts in generated images by updating the cross-attention layers. The original paper conducts experiments based on SDv1.5 and SDv2.1.

- **FinetuneFD** (Finetune Fair Diffusion) [84] employs biased fine-tuning combined with a distributional alignment loss to reduce bias in generated images. This method conducts experiments based on SDv1.5 in the original paper.

- **FairMapping** (Fair Mapping) [51] introduces a linear network to map textual conditioning embeddings into a debiased space, enabling demographically fair image generation. Additionally, an auxiliary detector is used to determine whether to activate the linear network based on the input prompt. The original paper conducts experiments using SDv1.5. We reproduced using the hyperparameters reported in the original paper due to the absence of official code.

- **BalancingAct** (Balancing Act) [70] introduces an auxiliary network called the Attribute Distribution Predictor, which maps UNet latent features to attribute distributions and guides the generation process toward a prescribed demographic distribution. The original paper conducts experiments using SDv1.5.

- **Debias VL** (Debiasing Vision-Language Model) [15] eliminates bias in vision-language foundation models by projecting out biased directions in text embeddings. The original paper conducts experiments using SDv2.1.

- **TI** (Textual Inversion) [26] enables personalized image generation by learning new pseudo-words to represent specific visual concepts using some example images. TI can mitigate bias by replacing biased concepts with embeddings learned from unbiased datasets. We reconduct experiments using SDv1.5.

- **AITTI** (Adaptive Inclusive Token for Text-to-Image) [40] introduces an adaptive mapping network that learns concept-specific inclusive tokens to mitigate stereotypical biases in T2I generation. The original paper conducts experiments using SDv1.5.

- **TIME** (Text-to-Image Model Editing) [69] edits implicit assumptions in pre-trained diffusion models by aligning under-specified prompts with user-desired alternatives through modifying cross-attention projection matrices. We reconduct experiments using SDv1.5.

- **MIST** (Mitigating Intersectional Bias with Disentangled Cross-Attention Editing) [105] isolates and adjusts biased attribute concepts while preserving unrelated content by editing the cross-attention layers in a disentangled manner. The original paper conducts experiments using SDv1.5.

- **FairSM** (Fair Sampling with Switching Mechanism) [14] obfuscates attribute-specific information while preserving semantic content by switching the conditioning of sensitive attributes at a learned transition point during the denoising process. The original paper conducts experiments using SDv1.5.

- **SANER** (Societal Attribute Neutralizer) [37] removes attribute information from CLIP text features, retaining only attribute-neutral descriptions. The original paper focuses solely on debiasing CLIP, while we transfer the debiased model to SDv1.5.

---

[3] https://huggingface.co/stable-diffusion-v1-5/stable-diffusion-v1-5
[4] https://huggingface.co/stabilityai/stable-diffusion-2-1

- **DEAR** (Debiasing with Additive Residuals) [83] learns additive residual image representations to offset the original representations, ensuring fair output representations. The original paper focuses solely on debiasing CLIP, while we transfer the debiased model to SDv1.5.

- **EntiGen** (Ethical Natural Language Interventions in Text-to-Image Generation) [1] encourages models to generate images representing diverse social groups across gender, skin color, and culture by appending natural language ethical interventions to prompts. The original paper only modifies the input to the CLIP text encoder, while we apply it to SDv1.5 and SDv2.1.

- **ITI-GEN** (Inclusive Text-to-Image Generation) [107] learning a set of prompt embeddings to generate images that can effectively represent all desired attribute categories. The original paper conducts experiments using SDv1.5.

### H.3 Implementation Details

We perform all experiments on an NVIDIA 4090 GPU. We generate images for six occupations (doctor, CEO, taxi driver, nurse, artist, and teacher) using the prompt template unless otherwise specified. In the main experiments, we fine-tune LoRA [41] with a rank of $50$ applied to the text encoder. Following the default setting for Stable Diffusion v1.5 and v2.1, we fix the classifier-free guidance (CFG) scale at $7.5$ for all experiments and visualizations. We use the *Adam with Weight Decay (AdamW)* [59] optimizer with a weight decay of $0.01$. The initial learning rate is set within the range of $9 \times 10^{-6}$ to $7 \times 10^{-5}$, depending on the version of Stable Diffusion and the specific attribute category. The batch size is fixed at $50$ across all scenarios, and the total number of epochs is set to $160$.

## I   Additional Experimental Results

### I.1   Expanded Version of Quantitative Results

Here, we present an expanded version of the quantitative results. First, we provide comparisons against a broader range of baseline methods. Then, we report results across additional evaluation metrics.

**Comparison with More Baselines.** Tab. 6 shows a comprehensive comparison with 16 baseline methods, including the original SD, 11 debiasing methods designed for diffusion models, and 4 debiasing methods tailored for CLIP. Our `LightFair` continues to achieve SOTA performance. Notably, since our approach targets the CLIP text encoder within the diffusion model, we include comparisons with CLIP debiasing methods. Existing CLIP debiasing approaches can be broadly categorized into three groups:

1. **Joint fine-tuning of the image and text encoders** (*e.g.*, [19, 65]): These methods are mainly designed for classification tasks. However, due to mismatched optimization objectives, the fine-tuned text embeddings often become incompatible with the U-Net used in diffusion models. Consequently, replacing the original text encoder with a jointly fine-tuned one often results in generation failures, producing noisy and semantically meaningless outputs.

2. **Fine-tuning only the image encoder** (*e.g.*, DEAR in Tab. 6): Since SD primarily relies on the CLIP text encoder for guiding generation, methods that modify only the image encoder do not address the core bias issues in generative tasks.

3. **Fine-tuning only the text encoder** (*e.g.*, SANER, EntiGen, and ITI-GEN in Tab. 6): These methods directly perform debiasing on the CLIP text encoder and apply it to SD-based image generation. They effectively demonstrate the importance of addressing bias at the level of text encoding for achieving fairness in generative models. Our `LightFair` belongs to this category. What distinguishes `LightFair` is its theoretically grounded loss functions targeting equalized odds and equalized quality, which contribute to its superior performance.

**Results on Additional Evaluation Metrics.** In addition to the 2 fairness metrics and 6 quality evaluation metrics used in our paper, several other evaluation metrics have been proposed in related work. For example, [70] employs Fairness Discrepancy (FD) to assess fairness, and [84] uses DINO

features to evaluate image quality. We incorporate both of these additional metrics in our evaluation, with the results presented in Tab. 7.

The results show that FD exhibits a similar trend to Bias-O, while the DINO-based quality scores align closely with those of CLIP-I. **Notably, our LightFair consistently ranks first or second across all ten evaluation metrics, underscoring its overall effectiveness in balancing fairness and generation quality.**

Table 6: Complete quantitative results on gender and race attributes. The champion and the runner-up are highlighted in **bold** and underline, respectively. Methods marked with * are reproduced using the model architectures and hyperparameters reported in the original papers due to the absence of official code. For clarity, methods added beyond those in Tab. 1 are highlighted in red.

| Method | Gender | | | | | | | | Race | | | | | | | |
|---|---|---|---|---|---|---|---|---|---|---|---|---|---|---|---|---|
| | Fairness | | Quality | | | | | | Fairness | | Quality | | | | | |
| | Bias-O↓ | Bias-Q↓ | CLIP-T↑ | CLIP-I↑ | FID↓ | IS↑ | AS-R↑ | AS-A↓ | Bias-O↓ | Bias-Q↓ | CLIP-T↑ | CLIP-I↑ | FID↓ | IS↑ | AS-R↑ | AS-A↓ |
| **Stable Diffusion v1.5** | | | | | | | | | | | | | | | | |
| SD [78] | 0.73 (±0.05) | 1.90 (±0.67) | 29.31 (±0.06) | - | 275.13 (±6.75) | 1.26 (±0.03) | 4.78 (±0.08) | 2.65 (±0.04) | 0.54 (±0.02) | 1.60 (±0.67) | 29.31 (±0.06) | - | 275.13 (±6.75) | 1.26 (±0.03) | 4.78 (±0.08) | 2.65 (±0.04) |
| FairD [24] | 0.79 (±0.04) | 3.25 (±1.15) | 28.79 (±0.11) | 75.91 (±0.56) | 269.62 (±4.42) | 1.30 (±0.03) | 4.57 (±0.09) | 2.82 (±0.05) | 0.50 (±0.02) | 1.50 (±0.38) | 28.95 (±0.10) | 74.33 (±0.68) | 262.72 (±4.84) | 1.28 (±0.03) | 4.55 (±0.08) | 2.83 (±0.06) |
| UCE [27] | 0.78 (±0.07) | 1.79 (±0.46) | 28.91 (±0.13) | 82.72 (±0.81) | 273.95 (±5.53) | 1.26 (±0.05) | 4.71 (±0.09) | 2.64 (±0.04) | 0.44 (±0.03) | 1.40 (±0.24) | 29.13 (±0.14) | 90.15 (±0.70) | 281.16 (±5.18) | 1.26 (±0.02) | 4.76 (±0.08) | 2.69 (±0.05) |
| FinetuneFD [84] | 0.38 (±0.07) | 2.31 (±0.40) | 29.34 (±0.13) | 76.17 (±0.68) | 278.21 (±7.53) | 1.24 (±0.05) | 4.38 (±0.06) | 2.86 (±0.04) | 0.20 (±0.03) | 1.41 (±0.23) | 29.02 (±0.15) | 74.57 (±0.53) | 270.09 (±5.99) | 1.26 (±0.02) | 4.33 (±0.06) | 2.87 (±0.05) |
| FairMapping* [51] | 0.46 (±0.05) | 2.16 (±0.72) | 29.30 (±0.16) | 76.00 (±0.66) | 278.81 (±5.84) | 1.26 (±0.02) | 4.34 (±0.07) | 2.90 (±0.03) | 0.34 (±0.02) | 1.75 (±0.47) | 29.29 (±0.15) | 76.54 (±0.71) | 280.95 (±5.02) | 1.26 (±0.03) | 4.53 (±0.08) | 2.80 (±0.05) |
| BalancingAct [70] | 0.41 (±0.05) | 1.70 (±0.55) | 29.30 (±0.11) | 77.37 (±0.64) | 272.08 (±5.16) | 1.28 (±0.02) | 4.71 (±0.06) | 2.68 (±0.04) | 0.34 (±0.02) | 1.13 (±0.36) | 29.34 (±0.11) | 77.44 (±0.72) | 271.91 (±5.35) | 1.29 (±0.03) | 4.72 (±0.10) | 2.66 (±0.04) |
| TI [26] | 0.56 (±0.06) | 1.88 (±0.37) | 28.76 (±0.10) | 75.43 (±0.54) | 278.92 (±6.22) | 1.27 (±0.02) | 4.45 (±0.07) | 2.74 (±0.04) | 0.47 (±0.03) | 1.45 (±0.27) | 28.67 (±0.17) | 67.96 (±0.84) | 275.20 (±5.07) | 1.25 (±0.03) | 4.43 (±0.04) | 2.81 (±0.04) |
| AITTI* [40] | 0.41 (±0.06) | 1.34 (±0.44) | 29.03 (±0.09) | 77.25 (±0.44) | 267.23 (±5.14) | 1.29 (±0.02) | 4.61 (±0.08) | 2.69 (±0.04) | 0.25 (±0.04) | 1.20 (±0.31) | 29.03 (±0.11) | 85.43 (±0.47) | 271.13 (±5.24) | 1.29 (±0.03) | 4.78 (±0.08) | 2.73 (±0.03) |
| TIME [69] | 0.65 (±0.04) | 1.76 (±0.35) | 28.45 (±0.12) | 73.71 (±0.69) | 279.17 (±4.43) | 1.25 (±0.02) | 4.45 (±0.07) | 2.86 (±0.04) | 0.39 (±0.04) | 1.51 (±0.34) | 27.97 (±0.15) | 76.53 (±0.68) | 275.72 (±6.84) | 1.26 (±0.02) | 4.57 (±0.04) | 2.75 (±0.02) |
| MIST* [105] | 0.39 (±0.05) | 1.35 (±0.43) | 29.10 (±0.13) | 76.67 (±0.35) | 254.33 (±4.76) | 1.27 (±0.02) | 4.69 (±0.05) | 2.64 (±0.04) | 0.26 (±0.03) | 1.19 (±0.24) | 29.08 (±0.09) | 83.25 (±0.75) | 265.83 (±6.78) | 1.28 (±0.02) | 4.74 (±0.07) | 2.57 (±0.05) |
| FairSM [14] | 0.65 (±0.04) | 1.83 (±0.21) | 27.98 (±0.11) | 74.23 (±0.59) | 265.60 (±5.04) | 1.26 (±0.03) | 4.39 (±0.07) | 2.83 (±0.03) | 0.42 (±0.03) | 1.61 (±0.37) | 28.68 (±0.06) | 72.83 (±0.60) | 271.58 (±5.83) | 1.27 (±0.03) | 4.58 (±0.07) | 2.84 (±0.03) |
| SANER* [37] | 0.52 (±0.02) | 1.65 (±0.34) | 28.13 (±0.08) | 75.28 (±0.77) | 275.34 (±5.40) | 1.25 (±0.04) | 4.53 (±0.09) | 2.76 (±0.04) | 0.45 (±0.03) | 1.41 (±0.33) | 28.50 (±0.13) | 73.64 (±0.51) | 273.21 (±6.42) | 1.24 (±0.02) | 4.38 (±0.09) | 2.67 (±0.05) |
| DEAR [63] | 0.73 (±0.05) | 1.90 (±0.67) | 29.31 (±0.06) | - | 275.13 (±6.75) | 1.26 (±0.03) | 4.78 (±0.08) | 2.65 (±0.04) | 0.54 (±0.02) | 1.60 (±0.67) | 29.31 (±0.06) | - | 275.13 (±6.75) | 1.26 (±0.03) | 4.78 (±0.08) | 2.65 (±0.04) |
| EntiGen [1] | 0.46 (±0.05) | 2.63 (±0.88) | 28.57 (±0.14) | 71.89 (±0.68) | 263.76 (±5.51) | 1.29 (±0.04) | 4.53 (±0.09) | 2.78 (±0.06) | 0.37 (±0.04) | 2.88 (±0.63) | 27.97 (±0.16) | 69.56 (±0.74) | 265.94 (±4.25) | 1.31 (±0.04) | 4.77 (±0.07) | 2.57 (±0.04) |
| ITI-GEN [107] | 0.39 (±0.06) | 1.27 (±0.82) | 28.36 (±0.12) | 68.82 (±0.59) | 246.55 (±5.97) | 1.29 (±0.02) | 4.36 (±0.11) | 2.86 (±0.07) | 0.31 (±0.04) | 1.62 (±0.37) | 28.13 (±0.14) | 66.97 (±0.57) | 269.84 (±6.61) | 1.33 (±0.03) | 4.14 (±0.10) | 2.85 (±0.05) |
| LightFair (Ours) | 0.30 (±0.08) | 0.99 (±0.55) | 30.57 (±0.16) | 80.09 (±0.76) | 233.53 (±5.50) | 1.30 (±0.03) | 4.79 (±0.08) | 2.60 (±0.04) | 0.18 (±0.04) | 1.06 (±0.43) | 31.34 (±0.20) | 86.31 (±0.70) | 259.96 (±7.75) | 1.33 (±0.03) | 4.80 (±0.10) | 2.55 (±0.04) |
| **Stable Diffusion v2.1** | | | | | | | | | | | | | | | | |
| SD [78] | 0.85 (±0.05) | 1.84 (±0.63) | 29.90 (±0.15) | - | 259.36 (±4.81) | 1.23 (±0.03) | 5.12 (±0.05) | 2.24 (±0.03) | 0.63 (±0.01) | 2.06 (±0.35) | 29.90 (±0.15) | - | 259.36 (±4.81) | 1.23 (±0.03) | 5.12 (±0.05) | 2.24 (±0.03) |
| debias VL [15] | 0.43 (±0.09) | 1.44 (±0.48) | 28.20 (±0.22) | 70.01 (±0.96) | 245.11 (±3.72) | 1.35 (±0.03) | 3.53 (±0.11) | 2.93 (±0.06) | 0.49 (±0.03) | 1.91 (±0.92) | 28.15 (±0.26) | 67.42 (±0.96) | 242.78 (±4.21) | 1.33 (±0.03) | 3.57 (±0.11) | 2.85 (±0.06) |
| UCE [27] | 0.90 (±0.04) | 1.67 (±0.71) | 29.41 (±0.13) | 87.94 (±0.86) | 268.52 (±3.92) | 1.22 (±0.02) | 5.12 (±0.05) | 2.32 (±0.03) | 0.50 (±0.03) | 1.95 (±0.37) | 29.44 (±0.12) | 80.46 (±1.13) | 250.57 (±4.49) | 1.23 (±0.03) | 5.17 (±0.08) | 2.25 (±0.03) |
| EntiGen [1] | 0.42 (±0.03) | 2.10 (±0.38) | 29.25 (±0.16) | 69.22 (±1.12) | 255.01 (±3.60) | 1.24 (±0.02) | 4.91 (±0.08) | 2.42 (±0.04) | 0.55 (±0.03) | 3.07 (±0.39) | 28.12 (±0.12) | 65.34 (±1.02) | 253.53 (±3.83) | 1.23 (±0.03) | 5.28 (±0.07) | 2.21 (±0.05) |
| LightFair (Ours) | 0.33 (±0.10) | 1.40 (±0.28) | 30.82 (±0.19) | 75.29 (±0.99) | 231.46 (±3.30) | 1.35 (±0.02) | 5.14 (±0.09) | 2.24 (±0.06) | 0.40 (±0.03) | 1.82 (±0.44) | 30.26 (±0.16) | 77.47 (±0.51) | 230.59 (±6.53) | 1.35 (±0.01) | 5.29 (±0.11) | 2.14 (±0.06) |

Table 7: Results on two additional evaluation metrics (FD and DINO). The champion and the runner-up are highlighted in **bold** and underline, respectively.

| Method | Gender | | | | Race | | | |
|---|---|---|---|---|---|---|---|---|
| | Bias-O↓ | FD↓ | CLIP-I↑ | DINO↑ | Bias-O↓ | FD↓ | CLIP-I↑ | DINO↑ |
| **Stable Diffusion v1.5** | | | | | | | | |
| SD | 0.73 (±0.05) | 0.45 (±0.03) | - | - | 0.54 (±0.02) | 0.17 (±0.01) | - | - |
| FairD | 0.79 (±0.04) | 0.45 (±0.02) | 75.91 (±0.56) | 0.53 (±0.02) | 0.50 (±0.02) | 0.15 (±0.01) | 74.33 (±0.68) | 0.53 (±0.02) |
| UCE | 0.78 (±0.07) | 0.48 (±0.04) | **82.72** (±0.81) | **0.70** (±0.02) | 0.44 (±0.03) | 0.13 (±0.01) | **90.15** (±0.70) | **0.83** (±0.02) |
| FinetuneFD | 0.38 (±0.07) | 0.22 (±0.04) | 76.17 (±0.68) | 0.57 (±0.01) | 0.20 (±0.03) | 0.07 (±0.01) | 74.57 (±0.53) | 0.54 (±0.01) |
| FairMapping | 0.46 (±0.05) | 0.30 (±0.03) | 76.00 (±0.66) | 0.53 (±0.02) | 0.34 (±0.02) | 0.10 (±0.01) | 76.54 (±0.71) | 0.54 (±0.02) |
| BalancingAct | 0.41 (±0.05) | 0.24 (±0.03) | 77.37 (±0.64) | 0.55 (±0.02) | 0.34 (±0.02) | 0.07 (±0.01) | 77.44 (±0.72) | 0.55 (±0.02) |
| TI | 0.56 (±0.06) | 0.33 (±0.04) | 75.43 (±0.54) | 0.54 (±0.02) | 0.47 (±0.03) | 0.14 (±0.01) | 67.96 (±0.84) | 0.43 (±0.04) |
| AITTI | 0.41 (±0.06) | 0.25 (±0.02) | 77.25 (±0.44) | 0.56 (±0.02) | 0.25 (±0.04) | 0.08 (±0.02) | 85.43 (±0.47) | 0.79 (±0.01) |
| TIME | 0.65 (±0.04) | 0.40 (±0.02) | 73.71 (±0.69) | 0.52 (±0.02) | 0.39 (±0.04) | 0.12 (±0.01) | 76.53 (±0.68) | 0.54 (±0.02) |
| MIST | 0.39 (±0.05) | 0.22 (±0.03) | 76.67 (±0.35) | 0.55 (±0.01) | 0.26 (±0.03) | 0.08 (±0.01) | 83.25 (±0.75) | 0.76 (±0.02) |
| FairSM | 0.65 (±0.04) | 0.43 (±0.02) | 74.23 (±0.59) | 0.52 (±0.02) | 0.42 (±0.03) | 0.13 (±0.01) | 72.83 (±0.60) | 0.51 (±0.03) |
| SANER | 0.52 (±0.02) | 0.35 (±0.02) | 75.28 (±0.77) | 0.52 (±0.02) | 0.45 (±0.03) | 0.13 (±0.02) | 73.64 (±0.51) | 0.50 (±0.02) |
| DEAR | 0.73 (±0.05) | 0.45 (±0.03) | - | - | 0.54 (±0.02) | 0.17 (±0.01) | - | - |
| EntiGen | 0.46 (±0.05) | 0.27 (±0.03) | 71.89 (±0.68) | 0.50 (±0.01) | 0.37 (±0.04) | 0.09 (±0.01) | 69.56 (±0.74) | 0.47 (±0.01) |
| ITI-GEN | 0.39 (±0.06) | 0.18 (±0.04) | 68.82 (±0.59) | 0.45 (±0.02) | 0.31 (±0.04) | 0.10 (±0.01) | 66.97 (±0.57) | 0.42 (±0.01) |
| LightFair (Ours) | **0.30** (±0.08) | **0.17** (±0.04) | 80.09 (±0.76) | 0.63 (±0.02) | **0.18** (±0.04) | **0.06** (±0.01) | 86.31 (±0.70) | 0.81 (±0.02) |
| **Stable Diffusion v2.1** | | | | | | | | |
| SD | 0.85 (±0.05) | 0.54 (±0.03) | - | - | 0.63 (±0.01) | 0.21 (±0.01) | - | - |
| debias VL | 0.43 (±0.09) | 0.28 (±0.05) | 70.01 (±0.96) | 0.49 (±0.02) | 0.49 (±0.03) | 0.14 (±0.01) | 67.42 (±0.96) | 0.46 (±0.02) |
| UCE | 0.90 (±0.04) | 0.59 (±0.02) | **87.94** (±0.86) | **0.71** (±0.02) | 0.50 (±0.03) | 0.16 (±0.01) | **80.46** (±1.13) | **0.64** (±0.02) |
| EntiGen | 0.42 (±0.03) | 0.25 (±0.02) | 69.22 (±1.12) | 0.49 (±0.02) | 0.55 (±0.03) | 0.14 (±0.01) | 65.34 (±1.02) | 0.45 (±0.02) |
| LightFair (Ours) | **0.33** (±0.10) | **0.21** (±0.05) | 75.29 (±0.99) | 0.63 (±0.02) | **0.40** (±0.03) | **0.11** (±0.01) | 77.47 (±1.05) | 0.53 (±0.03) |

## I.2 Expanded Version of Qualitative Results

Here, we present an expanded version of qualitative results. Fig. 12 showcases the outcomes of our gender-debiased SD v1.5 and v2.1, while Fig. 13 highlights the results of our race-debiased SD v1.5 and v2.1. Fig. 14 and Fig. 15 present the visual comparison between our debiased SD and those of other competitors.

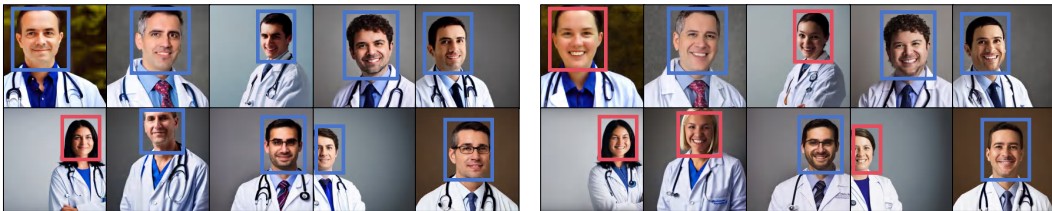

(a) Prompt: "Photo portrait of a **doctor**, a person". Left: original SD v1.5. Right: our **gender**-debiased **SD v1.5**.

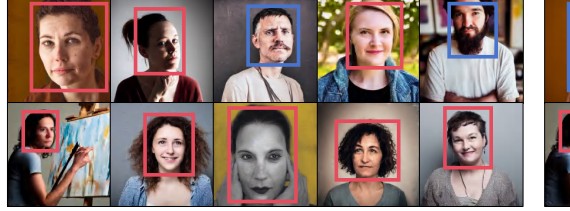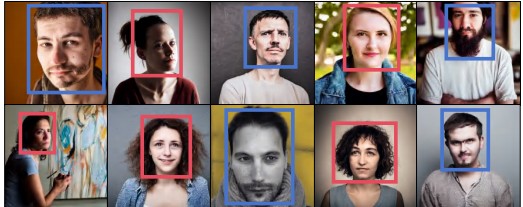

(b) Prompt: "Photo portrait of an **artist**, a person". Left: original SD v1.5. Right: our **gender**-debiased **SD v1.5**.

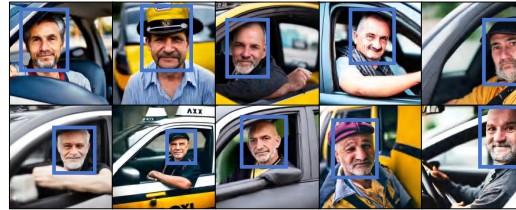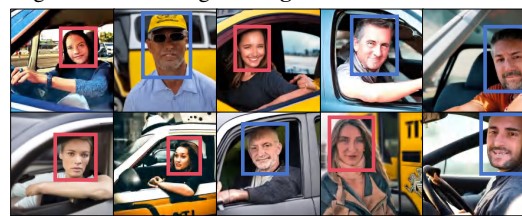

(c) Prompt: "Photo portrait of a **taxi driver**, a person". Left: original SD v2.1. Right: our **gender**-debiased **SD v2.1**.

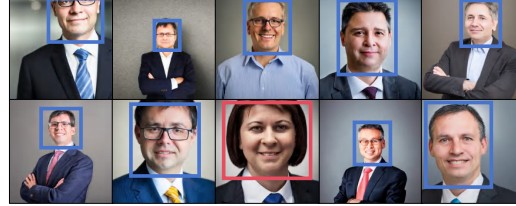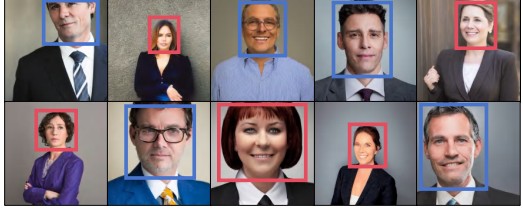

(d) Prompt: "Photo portrait of a **CEO**, a person". Left: original SD v2.1. Right: our **gender**-debiased **SD v2.1**.

Figure 12: Expanded version of qualitative results. Images generated by the original SD (left) and our debiased SD (right). For the same prompt, the images in corresponding positions are generated using the same random noise. Bounding boxes denote detected faces (Gender: Male, Female; Race: White, Asian, Black).

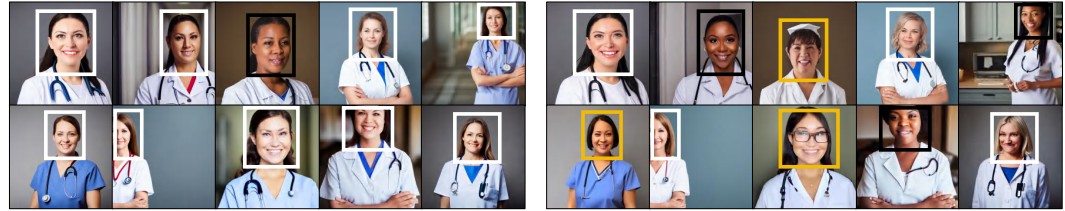

(a) Prompt: "Photo portrait of a **nurse**, a person". Left: original SD v1.5. Right: our **race**-debiased **SD v1.5**.

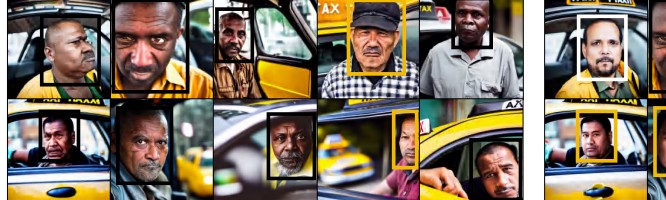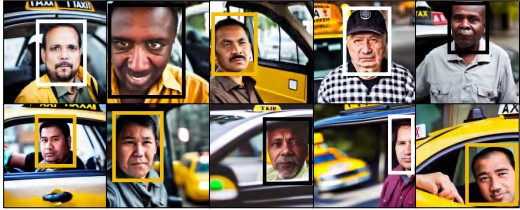

(b) Prompt: "Photo portrait of a **taxi driver**, a person". Left: original SD v1.5. Right: our **race**-debiased **SD v1.5**.

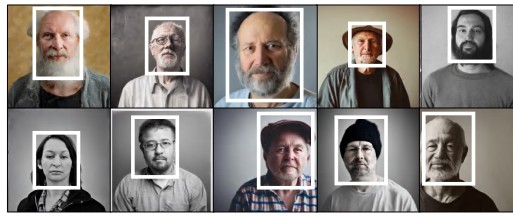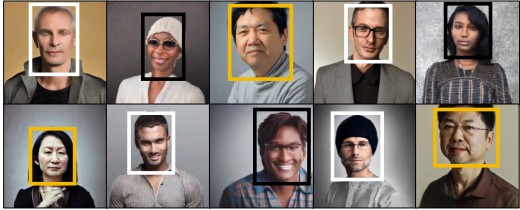

(c) Prompt: "Photo portrait of an **artist**, a person". Left: original SD v2.1. Right: our **race**-debiased **SD v2.1**.

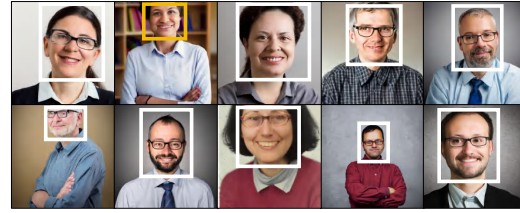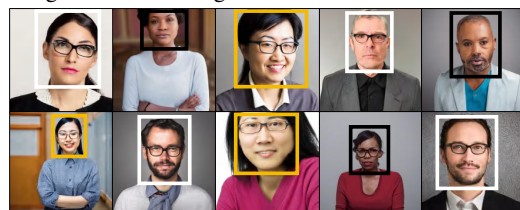

(d) Prompt: "Photo portrait of a **teacher**, a person". Left: original SD v2.1. Right: our **race**-debiased **SD v2.1**.

Figure 13: Expanded version of qualitative results. Images generated by the original SD (left) and our debiased SD (right). For the same prompt, the images in corresponding positions are generated using the same random noise. Bounding boxes denote detected faces (Gender: Male, Female; Race: White, Asian, Black).

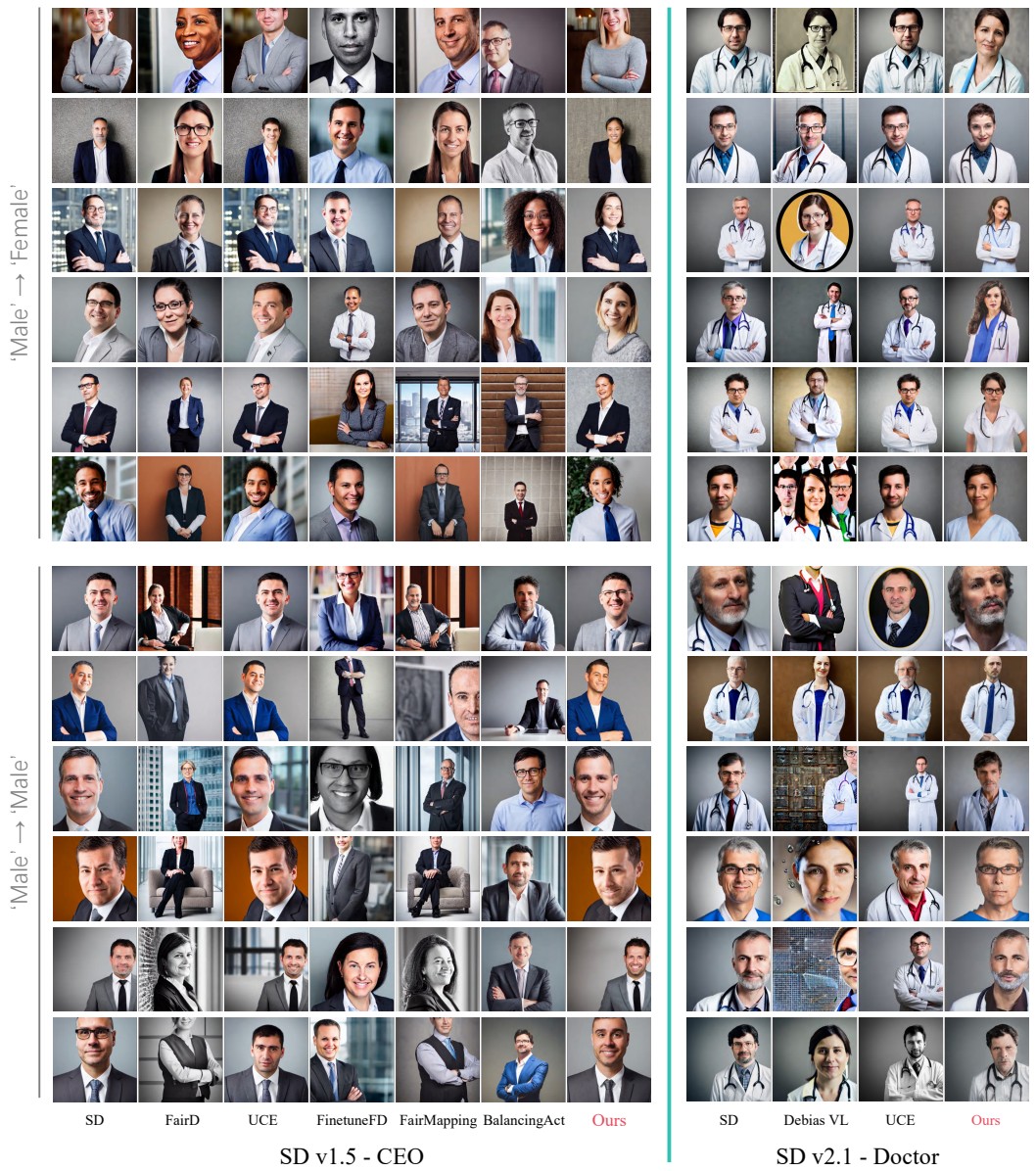

Figure 14: Expanded version of qualitative results. The performance of our **gender**-debiased SD and other competitors in terms of attribute transformation ('Male' → 'Female') and preservation ('Male' → 'Male'). For the same row, the images in corresponding positions are generated using the same random noise.

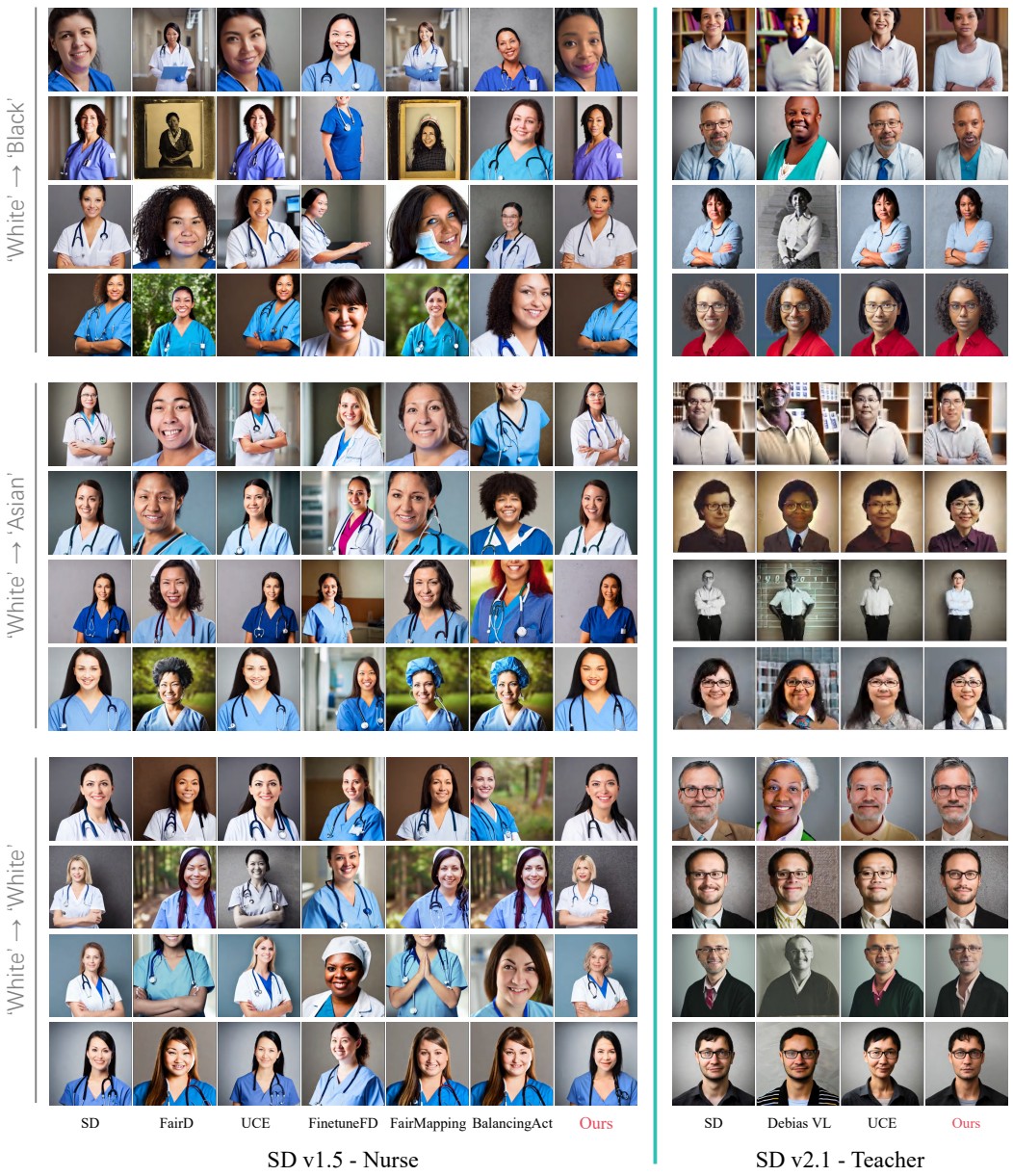

Figure 15: Expanded version of qualitative results. The performance of our **race**-debiased SD and other competitors in terms of attribute transformation ('White' → 'Black', 'White' → 'Asian') and preservation ('White' → 'White'). For the same row, the images in corresponding positions are generated using the same random noise.

## I.3   Prompts from LAION-Aesthetics V2

We obtain the following 30 prompts from the LAION-Aesthetics V2 dataset [81], following the settings of [84]. In addition, we use the sentence structure "Photo of a person, describing {prompt}" as the prompt to ensure that the output is a person.

['*Photograph of a doctor holding a headset sitting in front of a laptop*', '*a doctor examines a patients thyroid*', '*A doctor in a white coat on a computer screen*', '*A doctor speaks with a cancer patient*', '*A doctor giving a patient an eye exam*', '*Leadership Management Strategy. Concept as a ceo rowing a giant 3D illustration king crown in water as a business and financial metaphor for navigating and stock illustration*', '*a CEO talking to an advisor on video call*', '*Close-up image of a ceo pointing at*

*the data on the board while talking with colleagues on the foreground', 'Portrait of a CEO With a Team of Business Executives Standing by a Window', 'A CEO tries to get out of a speeding ticket by using a rather creative excuse, but fails', 'A taxi driver from Crimea', 'portrait of a taxi driver with cab', 'A taxi driver in Malaga wearing a face mask with a Spanish flag amid the coronavirus outbreak', 'A taxi driver reads a newspaper while waiting for customers at a taxi rank in Ljubljana, Slovenia, on Friday, May 10, 2013', 'TAXI – A taxi driver went on a tirade about Uber drivers at a council meeting on Dec. 16', 'Ladies leg in Cast being treated by a Nurse', 'A nurse and an old man in a wheelchair high five', 'white backround vector illustration of a nurse baby Illusztráció', 'A nurse and a smiling elderly patient in a wheelchair in the hall', 'A nurse places electrodes on Natalia Rodriguez's forehead, prepping her for bilateral ECT treatment', 'Young Artist Paints Sacre Coeur from the Ancient Rue Narvins by Ed Clark', 'An artist sketching a portrait of His Holiness the Dalai Lama on the third day of 'Thank You India' program at Kozhikode, Kerala, 13 Sept 2016', 'The two hands of an artist making a pot on the Ganges Art Studio Tour, one of the best things to do in Salt Spring Island in the summer', 'An artist paints a creature from ancient epics to decorate the royal crematorium and funeral complex. (AP Photo/Sakchai Lalit)', 'Doo Style', an artist tagging the basement levels at Le Bloc', 'A teacher in front of the school building vector image', 'A teacher leads class in Yixing Middle School in Lianshui County, Jiangsu Province, China', 'A teacher asks a question during a class at the Yeshiva high school Chachme Lev in Jerusalem. March 15, 2016', 'A teacher talking at the front of a classroom, Ghana', 'A teacher put a beaker of water on a hot plate'*]

### I.4   Qualitative Results on Diverse Prompts

In this section, we present qualitative results on diverse prompts. Fig. 16 showcases the results for non-templated prompts. Fig. 17 highlights the results in scenarios involving multiple people.

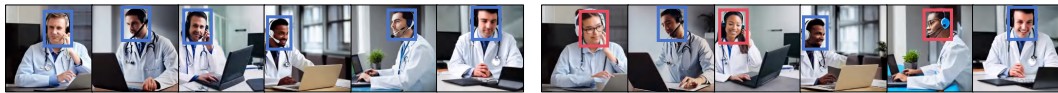

(a) Prompt: "Photograph of a doctor holding a headset sitting in front of a laptop". Left: original SD v1.5. Right: our debiased **SD v1.5**.

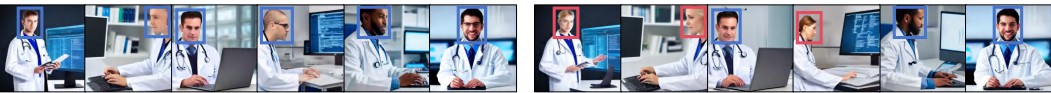

(b) Prompt: "A doctor in a white coat on a computer screen". Left: original SD v1.5. Right: our debiased **SD v1.5**.

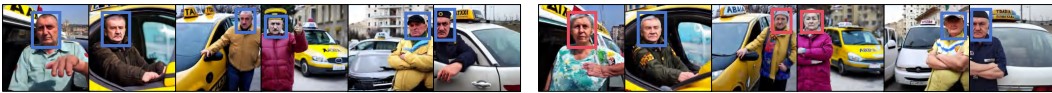

(c) Prompt: "A taxi driver from Crimea". Left: original SD v1.5. Right: our debiased **SD v1.5**.

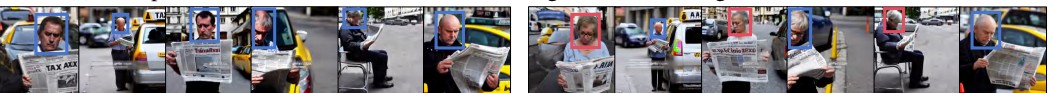

(d) Prompt: "A taxi driver reads a newspaper while waiting for customers at a taxi rank in Ljubljana, Slovenia, on Friday, May 10, 2013". Left: original SD v1.5. Right: our debiased **SD v1.5**.

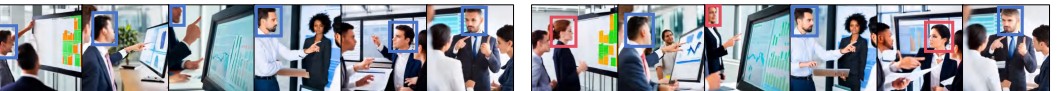

(e) Prompt: "Close-up image of a ceo pointing at the data on the board while talking with colleagues on the foreground". Left: original SD v2.1. Right: our debiased **SD v2.1**.

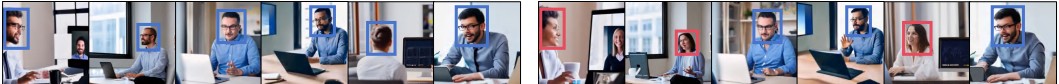

(f) Prompt: "a CEO talking to an advisor on video call". Left: original SD v2.1. Right: our debiased **SD v2.1**.

Figure 16: Qualitative results on non-templated prompts. Images generated by the original SD (left) and our gender-debiased SD (right). For the same prompt, the images in corresponding positions are generated using the same random noise. Bounding boxes denote detected faces (Gender: Male, Female).

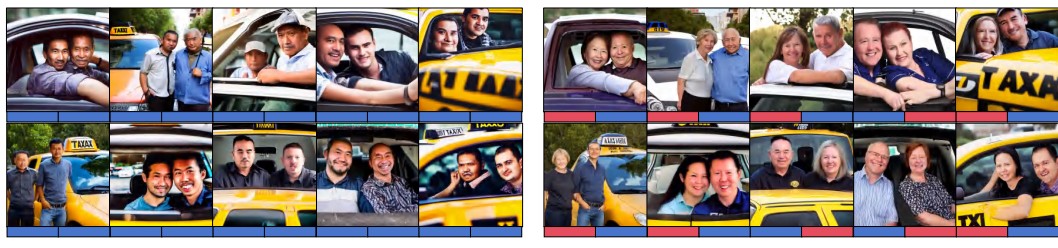

(a) Prompt: "Photo portrait of two taxi drivers, **two people**". Left: original SD v1.5. Right: our debiased **SD v1.5**.

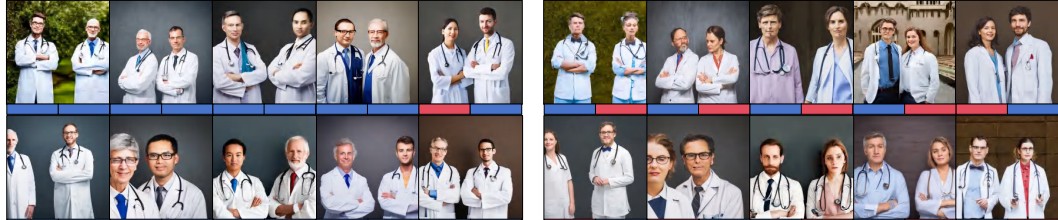

(b) Prompt: "Photo portrait of two doctors, **two people**". Left: original SD v2.1. Right: our debiased **SD v2.1**.

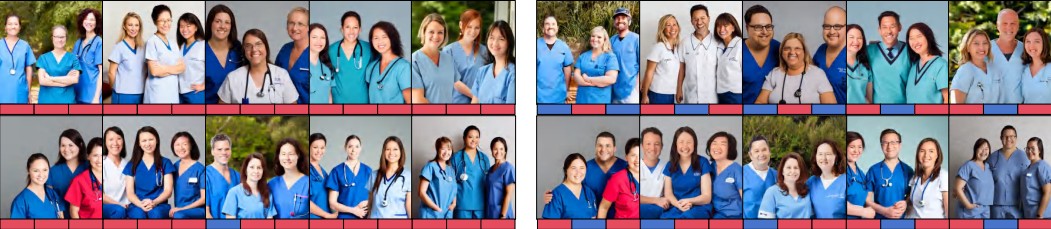

(c) Prompt: "Photo portrait of three nurses, **three people**". Left: original SD v1.5. Right: our debiased **SD v1.5**.

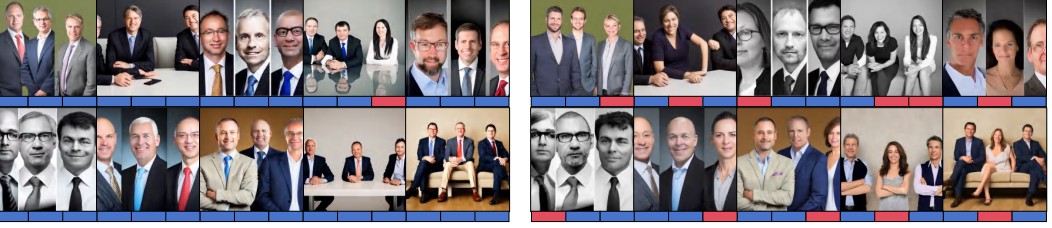

(d) Prompt: "Photo portrait of three CEOs, **three people**". Left: original SD v2.1. Right: our debiased **SD v2.1**.

Figure 17: Qualitative results on multiple people scenarios. Images generated by the original SD (left) and our gender-debiased SD (right). For the same prompt, the images in corresponding positions are generated using the same random noise. Each stripe below the images corresponds to one person in the generated image (Gender: Male, Female).

## I.5 Results of Mitigating Gender×Race Bias

In this section, we explore the performance of debiasing both Gender and Race attributes simultaneously. Specifically, we consider the attributes 'Male White', 'Male Black', 'Male Asian', 'Female White', 'Female Black', and 'Female Asian'. Since we have already performed debiasing for gender and race attributes separately in Sec. 5.2, we leverage the previously obtained training results to decouple the cross-attribute problem into two single-attribute problems. Specifically, we load both the gender-debiased and race-debiased LoRA modules simultaneously and conduct testing. The results are shown in Tab. 8 and Fig. 18. The results indicate that our model achieves excellent cross-attribute debiasing on SD v1.5 and v2.1, further demonstrating that our debiasing modules can be combined to effectively address multiple attributes simultaneously.

Table 8: Quantitative results on Gender×Race attributes.

| Backbone | Method | Fairness | | Quality | | | |
| | | Bias-O ↓ | Bias-Q ↓ | CLIP-T ↑ | CLIP-I ↑ | FID ↓ | IS ↑ |
|---|---|---|---|---|---|---|---|
| SD v1.5 | SD | 0.29 (±0.01) | 1.31 (±0.54) | 29.32 (±0.06) | - | 275.85 (±6.29) | 1.26 (±0.03) |
| | Ours | **0.14** (±0.01) | **0.91** (±0.32) | **31.34** (±0.20) | **62.82** (±5.57) | **259.96** (±7.75) | **1.33** (±0.03) |
| SD v2.1 | SD | 0.32 (±0.01) | 1.12 (±0.37) | 29.90 (±0.15) | - | 259.36 (±4.81) | 1.23 (±0.03) |
| | Ours | **0.23** (±0.02) | **0.89** (±0.22) | **30.32** (±0.19) | **56.41** (±2.04) | **230.39** (±5.95) | **1.25** (±0.01) |

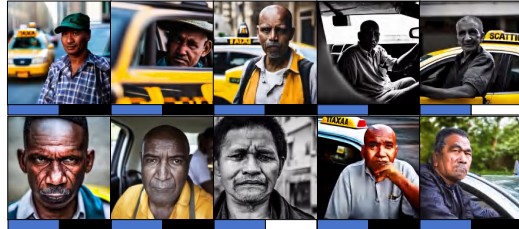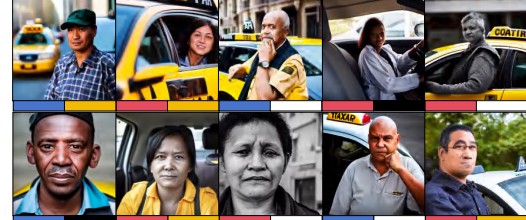

(a) Prompt: "Photo portrait of a taxi driver, a person". Left: original SD v1.5. Right: our debiased **SD v1.5**.

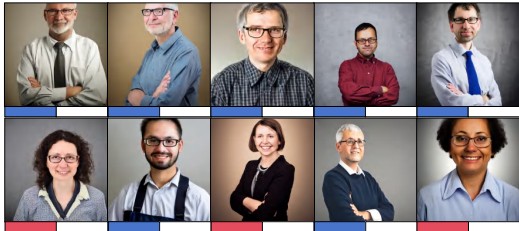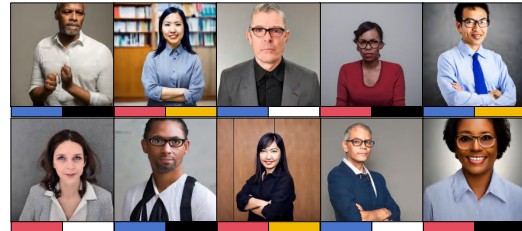

(b) Prompt: "Photo portrait of a teacher, a person". Left: original SD v2.1. Right: our debiased **SD v2.1**.

Figure 18: Qualitative results on gender×race debiasing. Images generated by the original SD (left) and our gender×race-debiased SD (right). For the same prompt, the images in corresponding positions are generated using the same random noise. Each stripe below the images represents a specific attribute (Gender: Male, Female; Race: White, Asian, Black).

## I.6 Results of Mitigating Age Bias

In this section, we focus on debiasing the age attribute. Specifically, we consider two attributes: 'Young (ages 0 to 39)' and 'Old (ages 39 and above)'. Our goal is to achieve fairness generation across these two attributes. The experimental settings are consistent with those used in Sec. 5. The results, presented in Tab. 9 and Fig. 19, demonstrate that our method achieves effective debiasing on both versions of SD. This highlights the strong generalization capability of our approach, extending beyond gender and race to other attributes like age.

Table 9: Quantitative results on Age attributes.

| Backbone | Method | Fairness | | Quality | | | |
| --- | --- | --- | --- | --- | --- | --- | --- |
| | | Bias-O ↓ | Bias-Q ↓ | CLIP-T ↑ | CLIP-I ↑ | FID ↓ | IS ↑ |
| SD v1.5 | SD | 0.65 (±0.04) | 1.23 (±0.44) | 29.23 (±0.06) | - | 311.56 (±11.95) | 1.23 (±0.02) |
| | Ours | **0.34** (±0.02) | **0.95** (±0.33) | **30.15** (±0.06) | **79.56** (±4.11) | **278.66** (±9.54) | **1.25** (±0.02) |
| SD v2.1 | SD | 0.83 (±0.07) | 1.14 (±0.32) | 29.30 (±0.11) | - | 287.64 (±10.67) | 1.24 (±0.01) |
| | Ours | **0.32** (±0.04) | **0.89** (±0.23) | **31.23** (±0.07) | **82.22** (±6.23) | **246.36** (±9.03) | **1.26** (±0.01) |

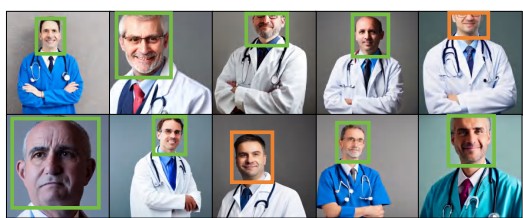 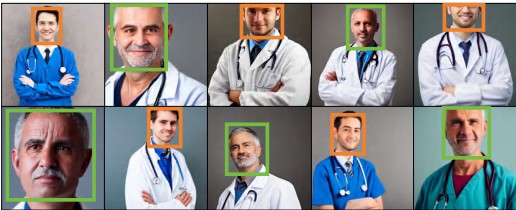

(a) Prompt: "Photo portrait of a doctor, a person". Left: original SD v1.5. Right: our debiased **SD v1.5**.

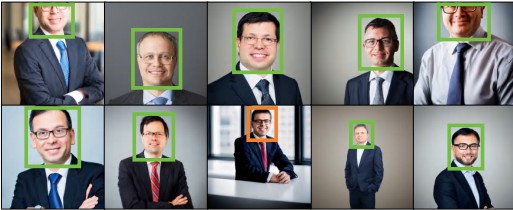 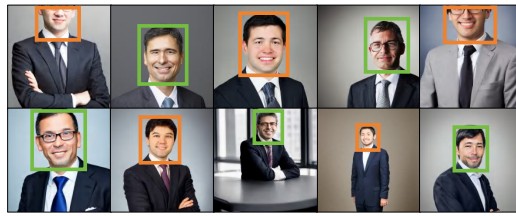

(b) Prompt: "Photo portrait of a CEO, a person". Left: original SD v2.1. Right: our debiased **SD v2.1**.

Figure 19: Qualitative results on age debiasing. Images generated by the original SD (left) and our age-debiased SD (right). For the same prompt, the images in corresponding positions are generated using the same random noise. Each stripe below the images represents a specific attribute (Age: Young, Old).

## I.7 Results of Debiasing under Diverse Target Distributions

We evaluate our `LightFair` under imbalanced target distributions. To achieve this, we modify the centroid-to-attribute distance to support debiasing toward arbitrary attribute distributions. Specifically, we revise Equ. (9) as follows:

$$\ell_o = \sqrt{\frac{1}{|A|} \sum_{i=1}^{|A|} \left[ \gamma_i s\left( \text{emb}_c^T(\cdot), \mathbb{E}\left[ \text{emb}_c^I(a_i) \right] \right) - \overline{s} \right]^2},$$ (52)

where $\gamma_i$ controls the target distribution.

In the original SD, the gender bias for the concept 'doctor' exhibits a male-to-female ratio of $9 : 1$. We set the target distributions to $7 : 3$ and $3 : 7$, respectively. Using `LightFair`, we generate 1000 images and record the resulting gender distribution, as shown in Figure 1. The results demonstrate that `LightFair` can be extended to support debiasing toward arbitrary attribute distributions.

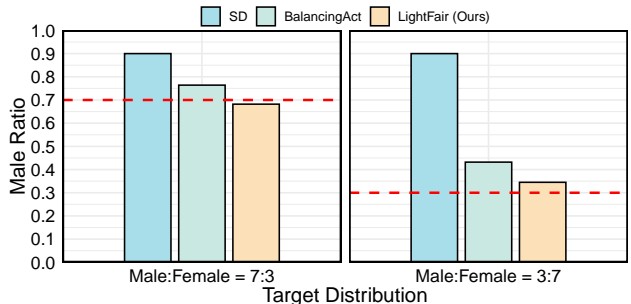

Figure 20: Results of debiasing under diverse target distributions.

## I.8 Ablation Study Results on the Impact of Adaptive Foreground Extraction

In Sec. 5.3, we conducted an initial ablation of the adaptive foreground extraction (AFE) module. In this section, we further investigate whether semantic information contained in background elements affects the results.

The AFE module is designed to follow the highest-salience regions linked to the prompt's main subject. Our prompts specify foreground-related content, such as "male doctor", so the module continues functioning even when background elements include attribute-related semantics, like uniforms.

To test a worst-case scenario, we generate 100 images using prompts that explicitly require uniforms in the background and examine the resulting attention maps. In 94 of these cases, the peak activations still concentrate on the foreground subject. As shown in Tab. 10, applying our `LightFair` to these images required only about 10% additional training epochs to achieve the same level of debiasing as with standard prompts. This finding suggests that the method remains robust in practice. **We conclude that for more complex scenes, a modest increase in training epochs is sufficient to maintain performance.**

Table 10: Results of Debiasing with AFE under Distracting Backgrounds.

| Method | Bias-O ($\downarrow$) | Bias-Q ($\downarrow$) |
|---|---|---|
| SD | 0.7 | 1.15 |
| Ours (normal) | 0.34 | 0.81 |
| Ours (worst-case) | 0.43 | 0.97 |
| Ours (worst-case + 10% training epochs) | 0.36 | 0.85 |

## I.9 Ablation Study Results on the Impact of Batch Size and Training Epochs

In this section, we present an extended version of the ablation study on hyperparameters, conducting detailed experiments on batch size and training epochs.

Fig. 21a ablates the hyperparameter batch size, which represents the number of images used to approximate attribute centroids in each iteration. The optimal value is 50. A smaller batch size fails to approximate the attribute centroids effectively, leading to insufficient debiasing. In contrast, a larger batch size does not further reduce bias but requires more images during training, resulting in additional computational overhead. Fig. 21b ablates the hyperparameter training epochs, with the optimal value being 160. A smaller number of epochs results in insufficient training, leading to inadequate debiasing. On the other hand, a larger number of epochs provides only marginal improvements in bias reduction while introducing significant computational costs that outweigh the benefits.

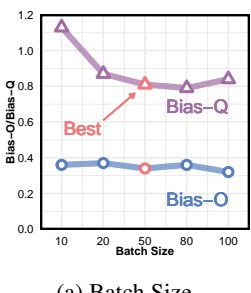
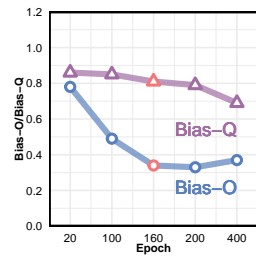

(a) Batch Size.  (b) Training Epochs.

Figure 21: Expanded Version of Ablation Study on Hyper-Parameters.

## I.10 Result of Collaborating with Other Debiasing Methods

The sources of bias in diffusion models are complex. In this work, we emphasize the critical role of debiasing the CLIP component and propose an effective approach for doing so. Our results show that debiasing CLIP alone is sufficient to achieve SOTA performance. **Notably, `LightFair` functions as a plug-and-play module that can be seamlessly integrated with existing debiasing methods targeting other components, further improving overall fairness**. We conduct experiments by combining our `LightFair` with debiased U-Nets from UCE and FinetuneFD. The results, presented in Tab. 11, demonstrate that `LightFair` can effectively complement other debiasing techniques.

Table 11: Result of collaborating with other debiasing methods.

| Method | Bias-O ↓ | Bias-Q ↓ | CLIP-T ↑ |
|---|---|---|---|
| UCE | 0.78 | 1.79 | 28.91 |
| +LightFair | **0.31** | **1.02** | **30.64** |
| FinetuneFD | 0.38 | 2.31 | 29.34 |
| +LightFair | **0.23** | **0.92** | **31.01** |

## I.11 Results of SD Models Based on the DiT Architecture

In this section, we replace the U-Net denoising network with a DiT-based architecture and conduct a preliminary evaluation. `LightFair` **only applies lightweight modifications to the text encoder and places no constraints on the denoising network architecture.** This is one reason why our method is easily generalized to various diffusion models. The results are shown in Tab. 12. They indicate that our method remains effective even with a DiT-style backbone.

Table 12: Result of SD models based on the DiT architecture.

| **Method** | **Bias-O (↓)** | **Bias-Q (↓)** |
|---|---|---|
| SD (DiT) | 0.33 | 1.42 |
| +LightFair | **0.29** | **1.23** |

## I.12 Results of User Studies

In this section, we conduct an additional user study to support the quantitative results with human judgment. We recruit 30 participants and show each of them images generated by four baselines (SD, FinetuneFD, FairMapping, BalancingAct) and our `LightFair`.

Participants rate perceived fairness, diversity, and image quality using a five-point Likert scale. The results are shown in Tab. 13. `LightFair` achieves a mean fairness score of $4.3$, compared to $3.8$ for the best-performing baseline. It also receives the highest scores for diversity and image quality.

Inter-rater agreement, measured by Fleiss' kappa, reaches 0.16. These findings confirm that the improvements are clearly perceived by human evaluators without reducing variety.

Table 13: Results of user studies.

| Method | Fairness | Diversity | Quality |
|--------|----------|-----------|---------|
| SD | 1.9 | 3.1 | 3.3 |
| FinetuneFD | 3.2 | 3.4 | 3.6 |
| FairMapping | 3.8 | 3.9 | **4** |
| BalancingAct | 3.5 | 3.6 | 3.7 |
| LightFair | **4.3** | **4.2** | **4** |

### I.13 Evaluation on Prompts with Attribute

This section demonstrates that **eliminating bias does not impact the semantic understanding of the attributes themselves**. We use our debiased Stable Diffusion model to generate 20 images for each specified attribute prompt, shown in Fig. 22. In this case, we perform debiasing on gender, but it does not affect the semantics of the terms 'male' and 'female'. First, our model correctly identifies the term 'male' without generating female images (Fig. 22a & Fig. 22c), and it correctly identifies the term 'female' without generating male images (Fig. 22b & Fig. 22d). Second, no semantic bias is introduced, whether the prompt used during training (Fig. 22a & Fig. 22b) or a new prompt (Fig. 22c & Fig. 22d) is employed.

Next, we verify that **our method does not lead to the generation of neutral images**. Specifically, for the gender attribute, our approach avoids producing androgynous or ambiguous images. To evaluate this, we measure the number of generated "doctor" images falling near the gender decision boundary ($0.45 \leq P_{male} \leq 0.55$) out of 100 samples, comparing the original SD with our debiased version. The results are as follows: SD (6/100), LightFair (5/100). These results indicate that only a small fraction of images fall within this ambiguous range, confirming that our method does not induce neutrality in attribute expression.

This happens because the U-Net, trained with sufficient data, learns the relevant associations and directs image generation toward a single attribute. Since we do not modify the U-Net, our method avoids generating neutral images. This phenomenon is also confirmed in [51].

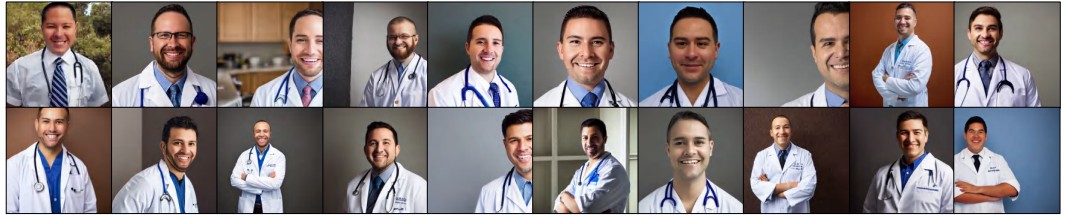

(a) Prompt: "Photo portrait of a male doctor, a person".

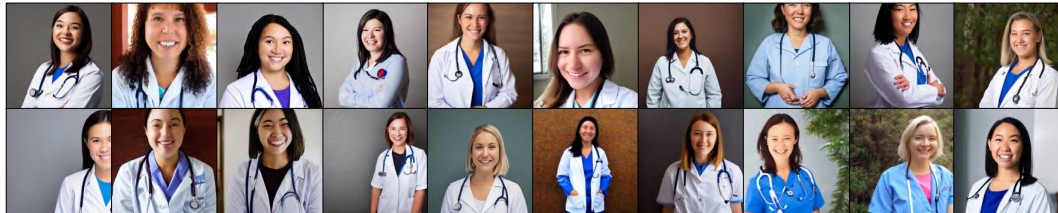

(b) Prompt: "Photo portrait of a female doctor, a person".

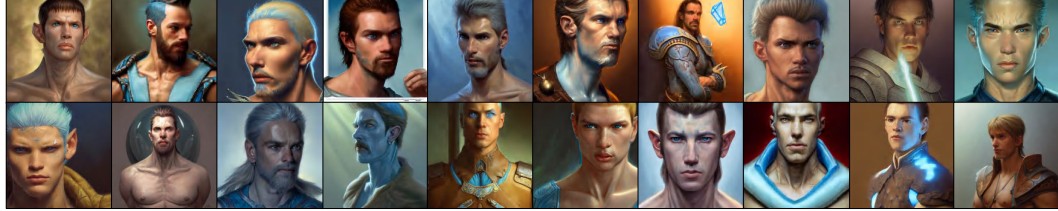

(c) Prompt: "a portrait of a male with light blue skin, gills on his neck, style by donato giancola, wayne reynolds, jeff easley dramatic light, high detail, cinematic lighting, artstation, dungeons and dragons".

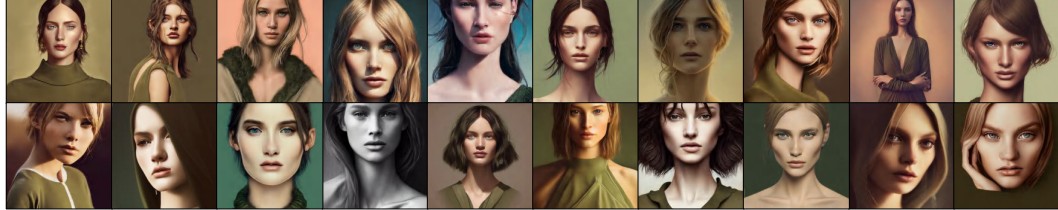

(d) Prompt: "close up portrait of beautiful female supermodel wearing olive dress, hotography by amy leibowitz, wlop, jeremy lipkin, beeple, intricate, symmetrical front portrait, artgerm, ilya kuvshinov".

Figure 22: Images generated using prompts with attribute. All images are generated using our gender-debiased SD v1.5.

## I.14 Evaluaton on General Prompts

This section examines the effect of our method on image generation for general prompts, which are not necessarily related to specific occupations. We randomly select 16 prompts from the DiffusionDB [94] dataset, written by real users. For each prompt, we generate six images using both the original Stable Diffusion model (SD v1.5 & SD v2.1) and our debiased version, with the same set of noises. The generated images are displayed in Fig. 23, Fig. 24, Fig. 25 and Fig. 26.

**We find that our debiased SD generates images almost identical to those produced by the original SD, ensuring that fine-tuning does not affect the semantics of general prompts.** Our debiased SD maintains a strong understanding of various concepts, including people such as 'Cristiano Ronaldo' (Fig. 23a) and 'Taylor Swift' (Fig. 23b), animals like 'dog' (Fig. 23c) and 'tiger' (Fig. 23d), plants such as 'sunflower' (Fig. 24a) and 'rose' (Fig. 24b), landscapes like 'grand canyon' (Fig. 24c) and 'old ruin' (Fig. 24d), cartoons like 'magic ritual place cartoon' (Fig. 25a) and 'lion cartoon' (Fig. 25b), oil paintings such as 'babylon' (Fig. 25c) and 'abandoned stone brick ruin' (Fig. 25d), artistic styles like 'Van Gogh' (Fig. 26a) and 'Cassius Marcellus Coolidge' (Fig. 26b). At the same time, the debiased SD still retains its creativity, such as generating dinosaurs kissing (Fig. 26c) or UFOs seamlessly integrated into realistic scenes (Fig. 26d).

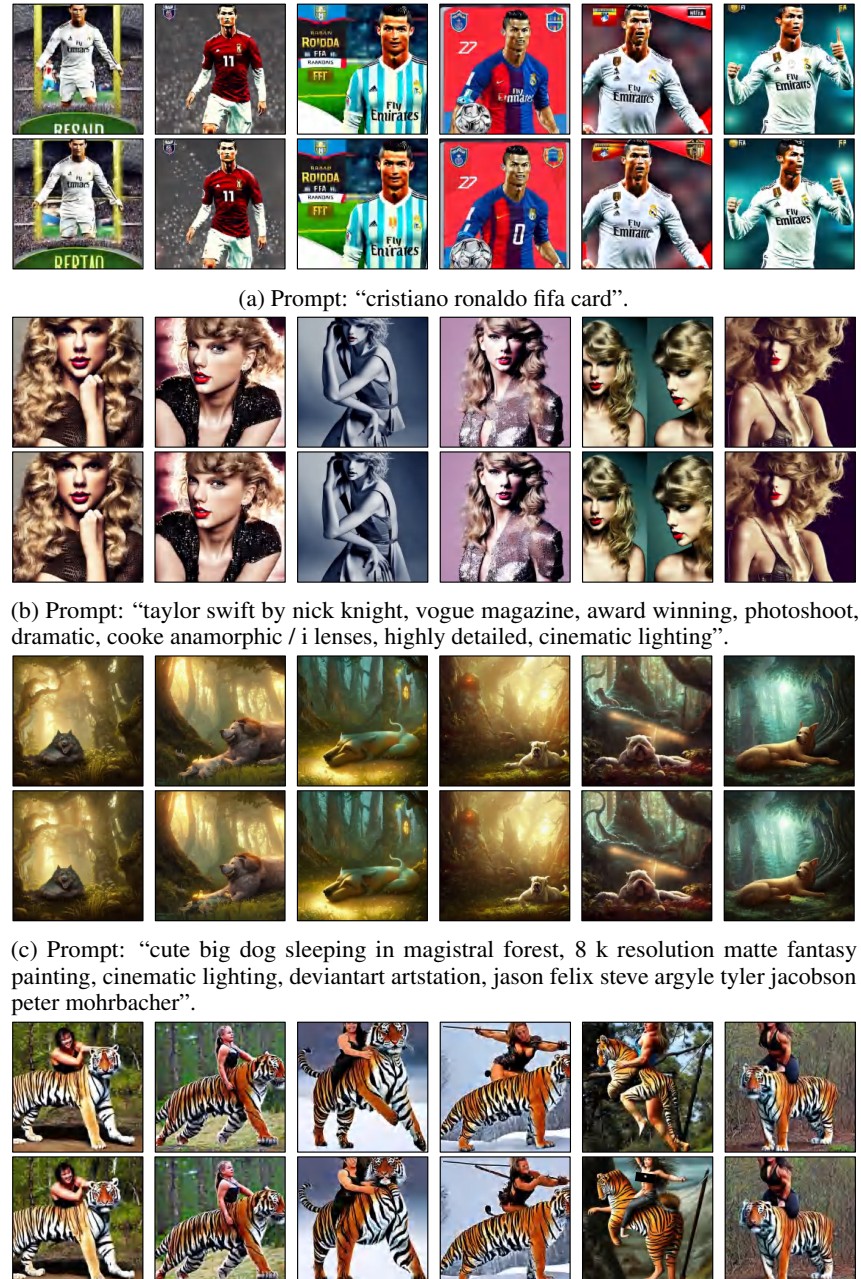

(a) Prompt: "cristiano ronaldo fifa card".

(b) Prompt: "taylor swift by nick knight, vogue magazine, award winning, photoshoot, dramatic, cooke anamorphic / i lenses, highly detailed, cinematic lighting".

(c) Prompt: "cute big dog sleeping in magistral forest, 8 k resolution matte fantasy painting, cinematic lighting, deviantart artstation, jason felix steve argyle tyler jacobson peter mohrbacher".

(d) Prompt: "a muscled warrior girl mounted on a large siberian tiger".

Figure 23: Images generated using general prompts. For every subfigure, the top row is generated using the **original SD v1.5**, and the bottom row is generated using **our gender-debiased SD v1.5**. The pair of images in the same column are generated using the same noise.

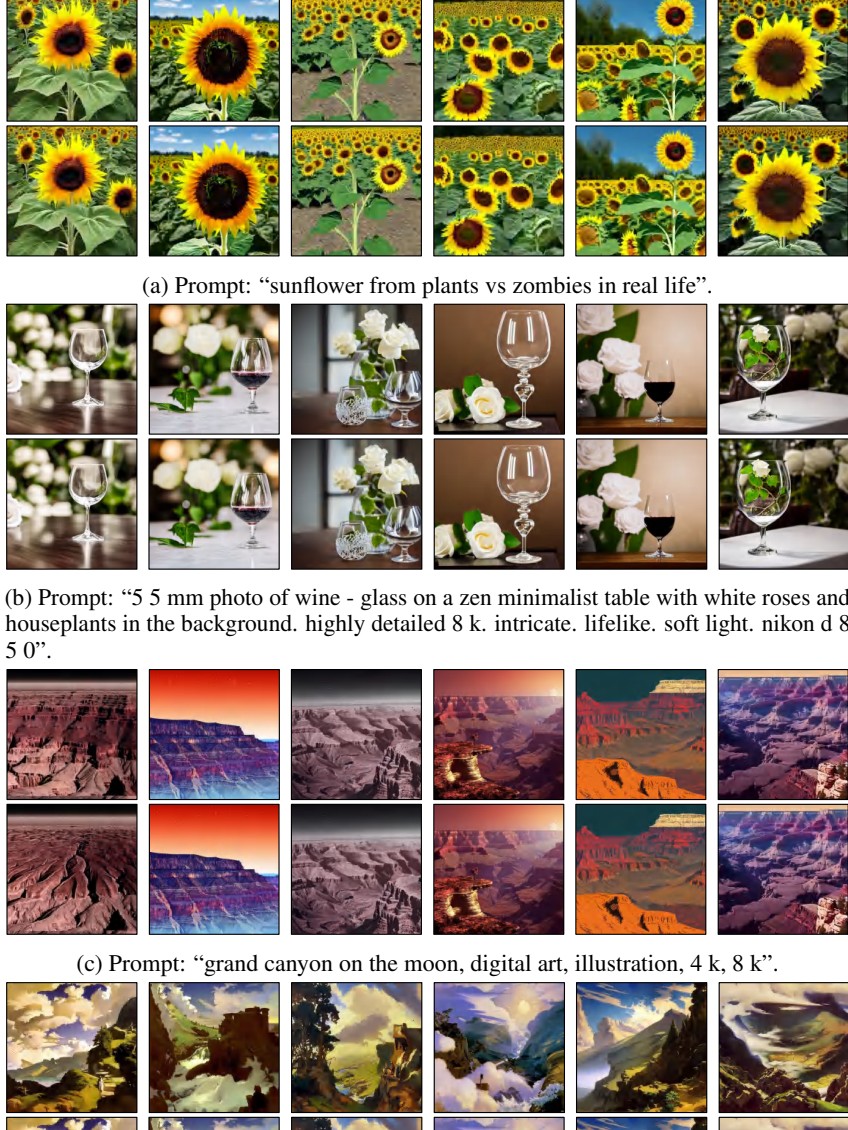

(a) Prompt: "sunflower from plants vs zombies in real life".

(b) Prompt: "5 5 mm photo of wine - glass on a zen minimalist table with white roses and houseplants in the background. highly detailed 8 k. intricate. lifelike. soft light. nikon d 8 5 0".

(c) Prompt: "grand canyon on the moon, digital art, illustration, 4 k, 8 k".

(d) Prompt: "painting by sargent and leyendecker and greg hildebrandt, apollinaris vasnetsov, savrasov levitan polenov, studio ghibly style mononoke, huge old ruins giovanni paolo panini, middle earth above the layered low clouds waterfall road between forests big lake wide river trees sunrise sea bay view faroe azores overcast storm masterpiece".

Figure 24: Images generated using general prompts. For every subfigure, the top row is generated using the **original SD v1.5**, and the bottom row is generated using **our race-debiased SD v1.5**. The pair of images in the same column are generated using the same noise.

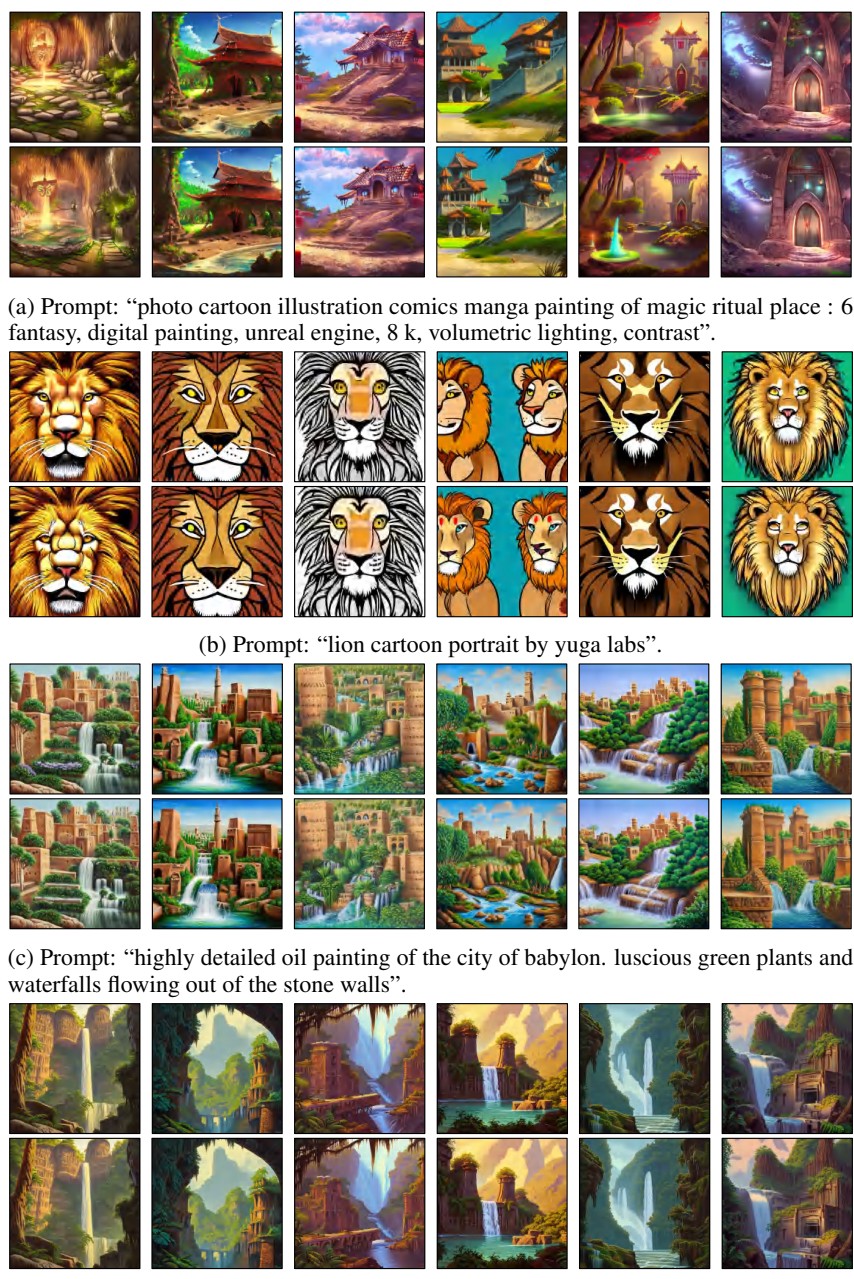

(a) Prompt: "photo cartoon illustration comics manga painting of magic ritual place : 6 fantasy, digital painting, unreal engine, 8 k, volumetric lighting, contrast".

(b) Prompt: "lion cartoon portrait by yuga labs".

(c) Prompt: "highly detailed oil painting of the city of babylon. luscious green plants and waterfalls flowing out of the stone walls".

(d) Prompt: "classic oil painting, abandoned stone brick ruins, as a dnd environment, surrounded by jungle and waterfalls, as a book cover illustration, readability, cottagecore, extremely detailed, concept art, smooth, sharp focus, art by brothers hildebrandt".

Figure 25: Images generated using general prompts. For every subfigure, the top row is generated using the **original SD v2.1**, and the bottom row is generated using **our gender-debiased SD v2.1**. The pair of images in the same column are generated using the same noise.

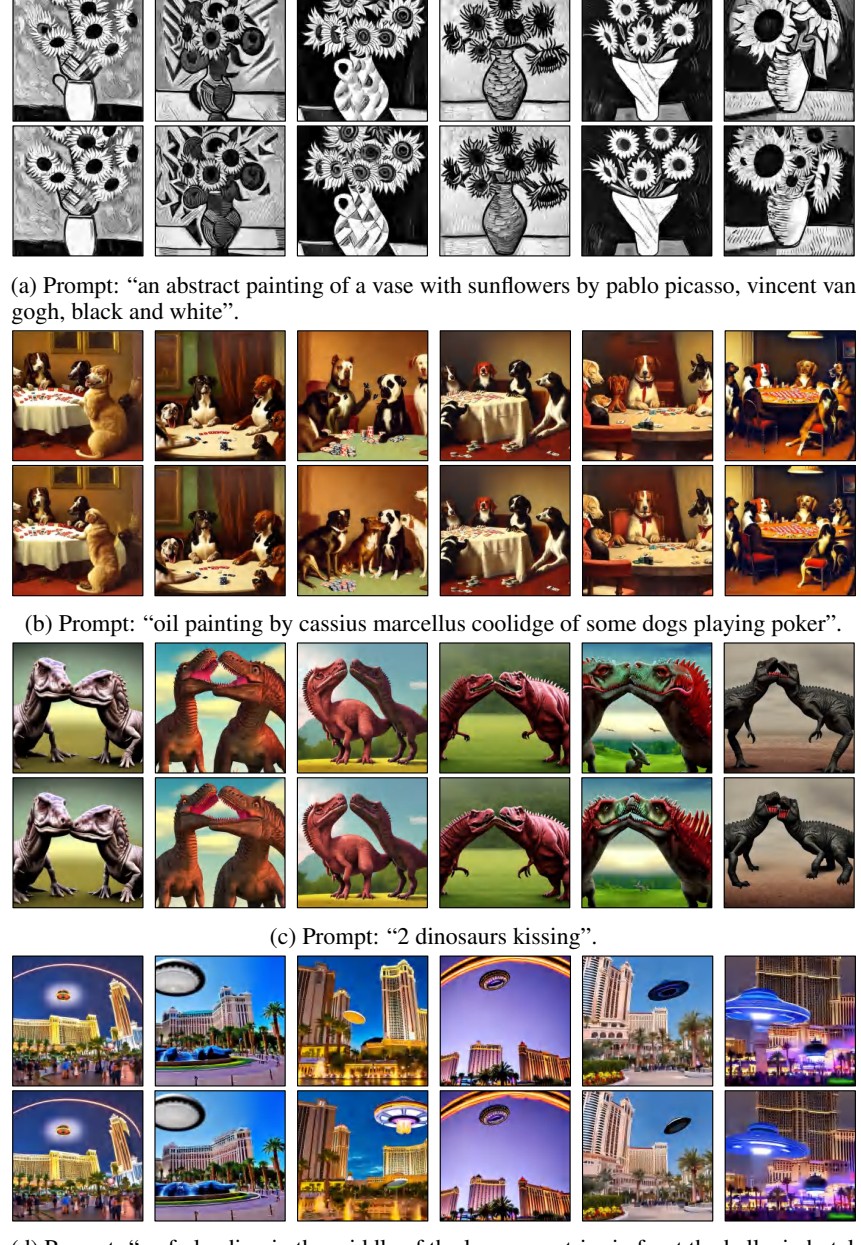

(a) Prompt: "an abstract painting of a vase with sunflowers by pablo picasso, vincent van gogh, black and white".

(b) Prompt: "oil painting by cassius marcellus coolidge of some dogs playing poker".

(c) Prompt: "2 dinosaurs kissing".

(d) Prompt: "a ufo landing in the middle of the las vegas strip. in front the bellagio hotel. professional photography".

Figure 26: Images generated using general prompts. For every subfigure, the top row is generated using the **original SD v2.1**, and the bottom row is generated using **our race-debiased SD v2.1**. The pair of images in the same column are generated using the same noise.

## I.15 Analysis of Time and Spatial Complexity

In this section, we analyze the spatial and time complexities of our `LightFair` compared to other methods, as summarized in Tab. 14. For spatial complexity, we report the number of trainable parameters. For time complexity, we measure the time required for each training iteration and the time needed for each denoising step. The results show that, compared to post-processing methods (FairD and Debias VL), our method achieves faster sampling speeds by eliminating the need for additional auxiliary networks. Furthermore, compared to fine-tuning methods (UCE, FinetuneFD, FairMapping and BalancingAct), our method identifies the key structures causing bias more precisely, resulting in fewer parameters to fine-tune and faster training speeds.

Table 14: Analysis of the complexity (time complexity (TC), spatial complexity (SC)) and effectiveness (Bias-O, Bias-Q, CLIP-T) of different methods.

| Method | SC (Parameters) | TC (Training) | TC (Sampling) | Bias-O | Bias-Q | CLIP-T |
|---|---|---|---|---|---|---|
| **Stable Diffusion v1.5** | | | | | | |
| FairD | - | - | 0.1179 s | 0.79 | 3.25 | 28.79 |
| UCE | 859.5210 M | 10.8213 s | 0.0662 s | 0.78 | 1.79 | 28.91 |
| FinetuneFD | 18.2592 M | 14.4 s | 0.0699 s | 0.38 | 2.31 | 29.34 |
| FairMapping | **0.7855 M** | 3.6383 s | 0.0658 s | 0.46 | 2.16 | 29.30 |
| BalancingAct | 8.1921 M | 3.0273 s | 0.1379 s | 0.41 | 1.70 | 29.30 |
| LightFair (Ours) | 3.6864 M | **2.4221 s** | **0.0631 s** | **0.30** | **0.99** | **30.57** |
| **Stable Diffusion v2.1** | | | | | | |
| Debias VL | - | - | 0.0874 s | 0.43 | 1.44 | 28.20 |
| UCE | 865.9107 M | 8.1834 s | 0.0585 s | 0.90 | 1.67 | 29.41 |
| LightFair (Ours) | **9.4208 M** | **3.3960 s** | **0.0567 s** | **0.33** | **1.40** | **30.82** |

# J  Limitations and Future Works

## J.1  Limitations

- Evaluation metrics themselves may introduce bias, potentially affecting model assessment. This is a common challenge across nearly all fairness evaluations in generative models. Nevertheless, these metrics are widely adopted in the generative modeling community, and we follow standard practice by using them as well. To mitigate the limitations of any single metric and reduce evaluation bias, we adopt a comprehensive evaluation protocol comprising 3 fairness metrics and 7 quality metrics, making our results more robust and persuasive.

- Some of the baseline methods (marked with * in Tab. 6) do not have official code. We re-implement them based on the descriptions in their original papers, strictly adhering to the reported configurations, including model architectures and hyperparameters. However, certain experimental details (*e.g.*, data augmentation strategies and random seed settings) are not specified in the original works. In these cases, we adopted the same settings used in our `LightFair` implementation. As a result, the reported metrics may differ slightly from those in the original papers. We have thoroughly examined these differences and confirmed that they are minimal. We will provide comprehensive comparisons once the official code of these methods becomes available.

- Our debiased SD occasionally generates artifacts such as non-smooth images, as shown in Fig. 17a and Fig. 23d. However, since the original SD exhibits similar issues, it is challenging to determine whether these artifacts are caused by our `LightFair` or are inherited characteristics of the original SD.

- As text-to-image generation models continue to evolve, a diverse array of model architectures is emerging. Our method is specifically designed for models with a text encoder and noise prediction network structure, and it is not yet applicable to other architectures.

## J.2  Future Works

We aim to develop methods for generating higher-quality debiased images and to explore fair-generation techniques for text-to-image models with diverse architectures. While our current method generalizes across multiple attributes, we acknowledge that fully continuous or user-defined attributes remain an open challenge. In the future, we plan to support continuous attributes by sampling

representative anchors along the spectrum or by optimizing against attribute regressors. At the same time, precisely defining a model's "fairness" remains challenging, as it largely depends on specific contexts and external factors. We envision that achieving genuine fairness will ultimately require joint efforts from researchers, policymakers, and practitioners.

# K  Statement

The "biases" discussed in this work are confined to those stemming from inherent statistical imbalances in training datasets, which often manifest as unequal representations of physical attributes such as gender, race, or age. Our objective is to address these biases to foster fairer and more accurate model outputs, particularly in scenarios where these outputs may significantly impact downstream applications.

That said, our approach has inherent limitations in mitigating biases affecting individuals whose identities do not conform to conventional societal categories, such as those with non-binary gender identities or mixed racial backgrounds.

It is important to clarify that this work does not seek to redefine or challenge societal norms or beliefs, nor does it attempt to provide solutions to the multifaceted and systemic issues of societal bias. Instead, our focus remains within the technical domain of machine learning, aiming to improve the robustness and fairness of generative models based on clear and measurable criteria.

Finally, while this study underscores the ethical significance of addressing bias in artificial intelligence, we acknowledge that technical interventions alone are insufficient to tackle deeper societal inequities. We advocate for multidisciplinary collaboration among researchers, policymakers, and practitioners to ensure that AI advancements align with and support broader societal values.

