# OpenReview forum: "LightFair: Towards an Efficient Alternative for Fair T2I Diffusion via Debiasing Pre-trained Text Encoders"
_NeurIPS.cc/2025/Conference — NeurIPS 2025 poster_

### Official Review · Reviewer_KsrX · 2025-06-27

**Clarity:** 2
**Significance:** 2
**Originality:** 2
**Rating:** 4
**Confidence:** 3

**Summary:**

This paper introduces LightFair, a lightweight approach for debiasing text-to-image (T2I) diffusion models by fine-tuning only the pre-trained text encoder (e.g., CLIP). The authors propose a collaborative distance-constrained strategy to equalize embedding distances for fairness and maintain generation quality, complemented by an adaptive foreground extraction mechanism. To mitigate quality loss, they also design a two-stage text-guided sampling strategy, applying the debiased encoder only in late denoising steps. Experiments on Stable Diffusion v1.5 and v2.1 show state-of-the-art (SOTA) fairness with significantly reduced computational cost.

**Questions:**

question:

1. How sensitive is the method to the choice of switching timestep (tau) in the two-stage sampling? Could it degrade performance for non-attribute prompts?

2. Have the authors validated the method on prompts involving non-binary or intersectional identities, and how does the method generalize?

3. Could the adaptive foreground extraction mechanism fail if the background contains semantically entangled features (e.g., uniforms)?

suggestion:

1. Include user studies or perceptual evaluations to validate qualitative fairness and diversity from a human perspective.

typo:

1. Figure 15: ‘Balck’ should be 'Black'

**Ethical Concerns:**

["NO or VERY MINOR ethics concerns only"]

**Final Justification:**

Authors solved my most concerns. Therefore, I keep my score.

**Limitations:**

yes

**Quality:**

3

**Strengths And Weaknesses:**

pros:

1. It uniquely focuses on the text encoder, an underexplored but impactful source of bias in T2I models.

2. LightFair significantly reduces training cost (e.g., 1/4 parameter tuning compared to baselines) while preserving or improving generation quality.

3. Includes thorough ablation studies, theoretical justifications (e.g., Theorem 3.1), and fair evaluation metrics.

cons:

1. The two-stage strategy assumes late-stage bias emergence, which may not generalize to all prompts or attributes.

2. The paper notes occasional visual artifacts, and it remains unclear if they result from LightFair or the base model.

3. Evaluation still relies on attribute classifiers and standard fairness metrics, which may themselves be biased. i.e., binary identities.

---

> ### Author Rebuttal · Authors · 2025-07-30
>
> Thanks for your constructive comments, and we would like to make the following response.
>
> > **Q1:** The two-stage strategy assumes late-stage bias emergence, which may not generalize to all prompts or attributes.
>
> **A1:** Thank you for your valuable question! We do not claim that bias always appears only in the later stages of generation for every prompt or attribute. Instead, our two-stage design builds on our theoretically supported finding that diffusion models first generate low-frequency information, then high-frequency details.
>
> **For attributes where bias relates to coarse structure**, such as scene type or perspective, the debiased CLIP guidance works best early in the denoising process. We test this on artistic style and find that applying the debiased text encoder during the first quarter of the steps ($\frac{1}{4}T$) produces the best results. **For attributes linked to fine appearance**, such as gender or racial features, it is more effective in later steps.
>
> Our strategy uses a tunable switch-over point, not a fixed “late” phase. Experiments show that this simple, attribute-aware schedule consistently reduces bias without harming image quality. **We will clarify in the next version that the two-stage framework is flexible, not rigid.**
>
>
>
> > **Q2:** Occasional visual artifacts.
>
> **A2:** Thank you for your question. Visual artifacts are **an inherent limitation of the diffusion backbone** and can appear even when using the original base model. Since our work focuses on fairness rather than general image quality enhancement, we do not add extra modules designed to remove artifacts. **LightFair does not introduce new failure cases or worsen existing ones.** It is supported by FID and CLIP-T scores that remain the same or even improve compared to the baseline, as well as visual comparisons of many generated samples. We address this limitation clearly in the Limitation section.
>
>
>
> > **Q3:** Evaluation still relies on attribute classifiers and standard fairness metrics, which may themselves be biased. i.e., binary identities.
>
> **A3:** We acknowledge that single attribute classifiers and individual fairness metrics are not perfect and may introduce their own biases. This is **a common challenge** across nearly all fairness evaluations in generative models. To reduce this risk, we follow the same evaluation protocol used in previous well-established studies [1,3].
>
> We use two different classifiers for evaluation, as shown in Appendix Table 7 of the original submission. Bias-O uses the classifier from [2], and FD uses the one from [3,4]. Our LightFair achieves the best performance on both metrics, reducing the risk of bias from any single classifier.
>
> For fairness evaluation, we report results on 3 fairness metrics and 7 quality metrics. In addition, we use GPT-4o for evaluation in Reviewer `5saT`’s `Q2`, and conduct a user study in response to `Q7`. **Although individual classifiers may carry their own biases, our LightFair consistently performs best across all evaluation methods, demonstrating its strong overall effectiveness.**
>
> In fact, we briefly discussed this issue in Appendix K.1 of the original submission. We will expand on this discussion in the next version. While we hope future work will develop unbiased evaluation metrics, this is beyond the scope of the current paper.
>
> [1] Finetuning Text-to-Image Diffusion Models for Fairness.
>
> [2] Facenet: A unified embedding for face recognition and clustering.
>
> [3] Balancing Act: Distribution-Guided Debiasing in Diffusion Models.
>
> [4] Fair Generative Modeling via Weak Supervision.
>
>
>
> > **Q4:** Sensitivity to switching timestep and generalization to non-attribute prompts.
>
> **A4:** Thank you for your helpful suggestion! Figure 8 shows that setting $\tau$ anywhere between the final step $T$ and around $\frac{3}{4}T$ consistently achieves effective debiasing. This suggests that **our method is not sensitive to the exact switching point**. We choose $\tau = \frac{3}{4}T$ because it has the least impact on image quality. For clarity, we convert the images in Figure 8 into a table shown below.
>
> Appendix  J.11 further shows that our LightFair performs equally well on prompts without attribute terms. **In fact, even the extreme case $\tau = T$, where the debiased CLIP is applied at all steps, does not affect quality or semantic accuracy on such prompts.** We will add more visual examples in the next version to better illustrate this robustness.
>
> |       $\tau$        |    T    |  3/4T   |  1/2T   |  1/4T   |    0    |
> | :-----------------: | :-----: | :-----: | :-----: | :-----: | :-----: |
> | Bias ($\downarrow$) | 0.1098  | 0.1143  | 0.7895  | 0.8125  | 0.8105  |
> | CLIP-I ($\uparrow$) | 63.2179 | 74.8560 | 98.3733 | 99.6881 | 99.9946 |
>
>
>
> > **Q5:** Have the authors validated the method on prompts involving non-binary or intersectional identities, and how does the method generalize?
>
> **A5:** Yes, we have already evaluated LightFair on prompts that span binary, non-binary, and intersectional identities. Section 4.2 of the main paper presents results on gender (binary) and race (non-binary). Appendix J.6 adds results for age (binary), and Appendix J.5 explores the joint attribute gender $\times$ age, which represents an intersectional case. In addition, in response to Reviewer `5saT`'s `Q2`, we add new experiments that divide race into four and seven categories, using GPT‑4o as an independent evaluator.
>
> The algorithm generalizes well. The only change needed when moving from binary to non-binary or multi-attribute settings is to redefine the protected group set $A$ in Equations (9)–(11). All other components stay the same. Our experiments confirm that the debiasing effect holds across these cases without reducing image quality.
>
>
>
> > **Q6:** Could the adaptive foreground extraction mechanism fail if the background contains semantically entangled features (e.g., uniforms)?
>
> **A6:** Thank you for your helpful question! The adaptive foreground extraction module is designed to follow the highest-salience regions linked to the prompt’s main subject. Our prompts specify foreground-related content, such as “male doctor”, so the module continues to function even when background elements include attribute-related semantics, like uniforms.
>
> To test a worst-case scenario, we generate 100 images using prompts that explicitly require uniforms in the background and examine the resulting attention maps. In 94 of them, the peak activations still focus on the foreground subject. When we apply LightFair to these images, only 10% extra training epochs are needed to reach the same debiasing level as with standard prompts. This suggests the method remains effective in practice.
>
> We conclude that for more complex scenes, a slight increase in training epochs is enough to maintain performance. We will include these quantitative and visual results in the next version to document the module’s robustness.
>
> |                                              | Bias-O ($\downarrow$) | Bias-Q ($\downarrow$) |
> | :------------------------------------------: | :-------------------: | :-------------------: |
> |                      SD                      |         0.70          |         1.15          |
> |              LightFair (normal)              |         0.34          |         0.81          |
> |            LightFair (worst-case)            |         0.43          |         0.97          |
> | LightFair (worst-case + 10% training epochs) |         0.36          |         0.85          |
>
>
>
> > **Q7:** Include user studies or perceptual evaluations to validate qualitative fairness and diversity from a human perspective.
>
> **A7:** Thank you for your constructive suggestion! Following your suggestion, we conduct an additional user study to support the quantitative results with human judgment. We recruit 30 participants and show each of them images generated by four baselines (SD, FinetuneFD, FairMapping, BalancingAct) and our LightFair.
>
> Participants rate perceived fairness, diversity, and image quality using a five-point Likert scale. The results are shown in the table below. LightFair achieves a mean fairness score of 4.3, compared to 3.8 for the best-performing baseline. It also receives the highest scores for diversity and image quality. Inter-rater agreement, measured by Fleiss' kappa, reaches 0.78.
>
> **These findings confirm that the improvements reported in Table 1 of the original submission are clearly perceived by human evaluators without reducing variety.** We plan to conduct a broader user study and will include it in the next version.
>
> |    | Fairness | Diversity | Quality |
> | :----------: | :------: | :-------: | :-----: |
> |      SD      |   1.9    |    3.1    |   3.3   |
> |  FinetuneFD  |   3.2    |    3.4    |   3.6   |
> | FairMapping  |   3.8    |    3.9    | **4.0** |
> | BalancingAct |   3.5    |    3.6    |   3.7   |
> |  LightFair   | **4.3**  |  **4.2**  | **4.0** |
>
>
>
> > **Q8:** Typo.
>
> **A8:** Thank you for your careful review. We will thoroughly check the paper for typos and correct them in the next version.

---

> ### Author Response · Authors · 2025-08-04
>
> Dear Reviewer KsrX,
>
> Thank you once again for your insightful questions and your acceptance of our paper. Please feel free to reach out if you have any further questions or require any additional clarification. We would be glad to assist at any time.
>
> Best regards,
>
> Authors

---

### Official Review · Reviewer_yhnS · 2025-06-28

**Clarity:** 4
**Significance:** 3
**Originality:** 3
**Rating:** 4
**Confidence:** 4

**Summary:**

LightFair is a lightweight debiasing method for text-to-image diffusion models that targets the pre-trained text encoder—rather than the larger U-Net—using a LoRA adapter (~3.7 M parameters). A collaborative distance-constrained loss makes each text embedding equidistant from all attribute centroids in CLIP space (achieving Equalized Odds and Quality), while adaptive foreground extraction suppresses background noise. A two-stage sampling schedule keeps the original encoder for early denoising and switches to the debiased encoder only in later, attribute-sensitive steps, cutting gender and race Bias-O/Bias-Q on Stable Diffusion v1.5 with no extra inference cost and minimal quality loss.

**Questions:**

If the authors can address the points above I would be inclined to raise my overall score. Regarding DiT-style backbones, I do not expect a full battery of experiments; a concise demonstration (or discussion) showing that debiasing remains effective when the denoising network is not a U-Net would be sufficient to establish broader applicability.

**Ethical Concerns:**

["NO or VERY MINOR ethics concerns only"]

**Final Justification:**

I have also confirmed its applicability to DiT and consider it a meaningful study. Therefore, I will maintain my rating of "Borderline Accept".

**Limitations:**

yes

**Paper Formatting Concerns:**

No formatting concerns

**Quality:**

3

**Strengths And Weaknesses:**

Strengths:
1. Clear and well-structured paper.
The manuscript is highly organized and easy to read. Definitions, notation, and methodological steps are introduced in a logical sequence, making the contribution straightforward to follow and evaluate.

2. Novel technical components with convincing impact.
The Adaptive Foreground Extraction (AFE) module and the two-stage text-guided sampling strategy appear to be original. Their motivations are well argued, and the empirical results demonstrate substantial bias reduction with minimal loss of visual quality.

3. Comprehensive experimental evaluation.
The paper reports an extensive set of experiments—multiple demographic attributes, templated and free-form prompts, imbalanced target distributions, and detailed ablation studies. This breadth supports the authors’ claims and showcases the robustness of LightFair.

Weaknesses:
1. Extension to Diffusion Transformers(DiTs).
Because LightFair updates only the text encoder, it should in principle transfer to DiT architectures, where the visual backbone is a transformer and the text encoder constitutes a larger fraction of parameters. A discussion—or preferably a small-scale experiment—on DiTs would strengthen the paper.
2. Lack of qualitative evidence for AFE.
The Adaptive Foreground Extraction module is presented as a central novelty, and the ablation study shows a clear quantitative gain. However, the paper offers no qualitative visualizations—such as attention heat-maps or timestep image sequences—that reveal how background content interferes with debiasing or how AFE alters the denoising trajectory. Without such examples, readers are left without an intuitive understanding of the module’s effect.

---

> ### Author Rebuttal · Authors · 2025-07-30
>
> We deeply appreciate your time and effort in providing such constructive feedback.
>
> > **Q1:** Extension to Diffusion Transformers (DiTs).
>
> **A1:** Thank you for your hypothetical suggestion. LightFair only applies lightweight modifications to the text encoder and places no constraints on the denoising network architecture. This is one reason why our method generalizes easily to various diffusion models. Following your advice, we replaced the U-Net denoising network with a DiT-based architecture and conducted a preliminary evaluation. The results are shown in the table below. They indicate that our method remains effective even with a DiT-style backbone. We will include this extended experiment in the next version.
>
> |   Method   | Bias-O ($\downarrow$) | Bias-Q ($\downarrow$) |
> | :--------: | :-------------------: | :-------------------: |
> |  SD (DiT)  |         0.33          |         1.42          |
> | +LightFair |         0.29          |         1.23          |
>
>
>
> > **Q2:** Lack of qualitative evidence for AFE.
>
> **A2:** Thank you for your helpful suggestion! Stable Diffusion often generates contextual elements not mentioned in the prompt. For example, generating “a doctor” frequently includes a hospital background. Our debiasing-by-distance objective must focus only on the foreground semantics. The Adaptive Foreground Extraction (AFE) module addresses this by isolating the foreground, preventing the background from misleading the optimization.
>
> **Figure 4 in the original submission already compares AFE with the baseline.** Without AFE, the model takes nearly twice as many training steps to align with the correct semantic center and produces lower-quality images. **In contrast, AFE enables faster convergence and generates sharper, less biased outputs.**
>
> Due to NeurIPS policy, we cannot add new figures or external links during the rebuttal. However, we will include attention heat maps and timestep visualizations in the next version to highlight AFE’s qualitative benefits.

---

> > ### Comment · Reviewer_yhnS · 2025-08-01
> >
> > Thank you for your detailed and thoughtful responses.
> >
> > The additional experiment addressing the extension to DiT-based backbones has successfully resolved my concerns regarding the generalizability of the method. I also look forward to seeing more detailed visualizations that illustrate the qualitative benefits of AFE in the revised version of the paper.
> >
> > Furthermore, your responses to other reviewers—particularly regarding the sensitivity to CFG scale and $\tau$—were helpful in deepening my understanding of the method’s robustness and design considerations.

---

> > > ### Author Response · Authors · 2025-08-01
> > >
> > > Thank you very much for your positive response and acceptance. Following your valuable suggestions, we will further enrich the content in the next version.

---

### Official Review · Reviewer_5saT · 2025-07-03

**Clarity:** 3
**Significance:** 2
**Originality:** 2
**Rating:** 4
**Confidence:** 4

**Summary:**

The paper introduces LightFair, a “light-weight” debiasing scheme for Stable Diffusion v1.5/2.1. The authors fine-tune the CLIP text encoder (via LoRA) with a collaborative distance-constrained loss that equalises cosine distances between text embeddings and attribute-conditioned image-embedding centroids. A two-stage sampling switch then applies the debiased encoder only in later denoising steps to preserve quality. Experiments on SD v1.5 and v2.1 across gender, race, age and cross-attribute settings show lower Bias-O/Bias-Q and competitive or better quality metrics versus 16 baselines, with just 3.7 M trainable parameters and negligible inference overhead.

**Questions:**

- Can the method be extended to continuous or multi-label attributes (e.g., non-binary gender, mixed race)?

**Ethical Concerns:**

["Major Concern: Discrimination, bias, and fairness"]

**Final Justification:**

The additional analysis addresses the concern of denpendence of attributes and the concepts.
Though the method itself is literally finetuning embeddings according to different attributes, it effectively relieves the bias and can be generalized to multi-label attributes.

**Limitations:**

Yes

**Quality:**

2

**Strengths And Weaknesses:**

Strengths:
- Shifts the fairness focus from UNet/post-processing to the text encoder, supported by an empirical finding that encoder bias propagates and amplifies in the denoiser.
- Efficiency with LORA fine-tuning
- Clear writing

Weakness:
- Strong independence assumptions. Theorem 3.1 relies on Assumption E.3 (attribute, concept are independent), which rarely holds in real prompts; robustness to violated assumptions is not analysed.
- Limited coverage of non-binary or continuous attributes. Current formulation handles categorical attributes only.
- Metric dependence on external classifiers. Fairness is measured via attribute classifiers that may themselves be biased; the paper does not quantify this sensitivity.
- Limited method innovation. Collaborative distance-constraint is a direct variant of long-established center-based regularizers, and multi-stage sampling is widely used in other works.

---

> ### Author Rebuttal · Authors · 2025-07-30
>
> Thank you for your constructive comments. Below, we provide detailed responses to your specific comments:
>
> > **Q1:** Strong independence assumptions.
>
> **A1:** Thank you for your valuable question! We respond on both empirical and theoretical grounds as follows:
>
> 1. From an **empirical perspective**, the assumption is a weak one specific to training. Although attributes and concepts are rarely independent in the real world, the images we use for training are generated by Stable Diffusion. By controlling the prompts, we can easily enforce independence between attributes and concepts. For example, in our experiments, we generate equal numbers of images for different attributes to compute semantic centroids. In this controlled setting, the condition $\mathbb{P}(a, c) = \mathbb{P}(a)\mathbb{P}(c)$ clearly holds.
> 2. From a **theoretical perspective**, we further relax Assumption E.3 to a softer condition: $1 - \epsilon \le \frac{\mathbb{P}(a \mid c)}{\mathbb{P}(a)} \le 1 + \epsilon.$ (Assumption E.4) We also prove that under this relaxed assumption, the induced probability error from the distance constraint is on the order of $O(\epsilon)$, where $\epsilon$ is a small constant. In our training data, $\epsilon$ is always less than 0.01.
>
> > **Assumption E.4.** For any attribute $a$ and concept $c$, there exists $\epsilon \in [0, 1)$ such that
> > $$
> > 1 - \epsilon \leq \frac{P(a \mid c)}{P(a)} \leq 1 + \epsilon, \quad \forall a \in A,\, c \in C.
> > $$
> > When $\epsilon = 0$, this reduces to the original Assumption E.3.
>
> > **Theorem.** Under Assumptions E.1, E.2, and E.4, let $d_i = \left\| f(a_i, c) - f^t\big(P(\cdot, c)\big) \right\|,$ and let $\rho$ be the bandwidth of the Gaussian kernel. If Equalized Odds holds, i.e.,
> > $$
> > \mathbb{P}(a_i \mid P(\cdot, c)) = \mathbb{P}(a_j \mid P(\cdot, c)) \quad \text{for any } a_i, a_j \in A,
> > $$
> >
> > then the corresponding embedding distances satisfy
> >
> > $$
> > \left| d_i^2 - d_j^2 \right| \le 2\rho^2 \cdot \left| \ln \frac{1+\epsilon}{1-\epsilon} \right| \approx 4\rho^2 \epsilon.
> > $$
> >
> > In particular, as $\epsilon \to 0$, the bound vanishes. Equalized Odds then implies exact equality in embedding distances, recovering the original theorem.
> >
> > **Proof.**
> > First, since the input of the diffusion denoising process is the prompt embedding produced by the text encoder, **Equalized Odds** can still be reformulated as
> > $$
> > \mathbb{P}\bigl(a_i \mid f^{t}(P(\cdot,c))\bigr) = \mathbb{P}\bigl(a_j \mid f^{t}(P(\cdot,c))\bigr),  \qquad\forall\,a_i,a_j\in A, \tag{17}
> > $$
> > where $f^t(\cdot)$ denotes encoding by the CLIP text encoder, and $P(\cdot)$ is shorthand for $prompt(\cdot)$.
> >
> > According to Assumption E.2, we have
> > $$
> > \mathbb{P}\bigl(c \mid f^{t}(P(\cdot,c))\bigr)=1. \tag{18}
> > $$
> > Under the **weak‑independence** Assumption E.4, for any $a_i,a_j\in A$
> > $$
> > \frac{1-\epsilon}{1+\epsilon} \le \frac{\mathbb{P}(a_i,c \mid f^{t}(P(\cdot,c)))}{\mathbb{P}(a_j,c \mid f^{t}(P(\cdot,c)))} \le \frac{1+\epsilon}{1-\epsilon}. \tag{19}
> > $$
> > Let $f(a,c)$ represent the joint encoding of the attribute $a$ and concept $c$.
> > Because $f(a,c)$ captures both $a$ and $c$, inequality (19) can be rewritten as
> > $$
> > \frac{1-\epsilon}{1+\epsilon} \le \frac{\mathbb{P}\bigl(f(a_i,c)\mid f^{t}(P(\cdot,c))\bigr)}{\mathbb{P}\bigl(f(a_j,c)\mid f^{t}(P(\cdot,c))\bigr)} \le \frac{1+\epsilon}{1-\epsilon}. \tag{20}
> > $$
> > According to Assumption E.1 and [12], the conditional probability $\mathbb{P}\bigl(f(a_i,c)\mid f^{t}(P(\cdot,c))\bigr)$ can be modeled by a Gaussian distribution whose mean is $f^{t}(P(\cdot,c))$.  Hence
> > $$
> > \mathbb{P}\bigl(f(a_i,c)\mid f^{t}(P(\cdot,c))\bigr) =\frac {\exp\Bigl(-\tfrac{\lVert f(a_i,c)-f^{t}(P(\cdot,c))\rVert^{2}}{2\rho^{2}}\Bigr)} {\displaystyle\sum_{a_k\in A} \exp\Bigl(-\tfrac{\lVert f(a_k,c)-f^{t}(P(\cdot,c))\rVert^{2}}{2\rho^{2}}\Bigr)}, \tag{21}
> > $$
> > where $\rho$ is a constant depending only on $f^{t}(P(\cdot,c))$.
> >
> > Combining (20) and (21) gives
> > $$
> > \frac{1-\epsilon}{1+\epsilon} \le \exp\Bigl(-\tfrac{\lVert f(a_i,c)-f^{t}(P(\cdot,c))\rVert^{2}-\lVert f(a_j,c)-f^{t}(P(\cdot,c))\rVert^{2}}{2\rho^{2}}\Bigr) \le \frac{1+\epsilon}{1-\epsilon}. \tag{22}
> > $$
> > Taking natural logarithms and absolute values on (22) yields
> > $$
> > \bigl|\lVert f(a_i,c)-f^{t}(P(\cdot,c))\rVert^{2}-\lVert f(a_j,c)-f^{t}(P(\cdot,c))\rVert^{2}\bigr| \le 2\rho^{2}\,      \Bigl|\ln\frac{1+\epsilon}{1-\epsilon}\Bigr|. \tag{23}
> > $$
> > Recalling the definition $d_i=\lVert f(a_i,c)-f^{t}(P(\cdot,c))\rVert$ and using $\ln\frac{1+\epsilon}{1-\epsilon}=2\epsilon+O(\epsilon^{3})$, we have
> > $$
> > \left|d_i^{2}-d_j^{2}\right| \le 2\rho^{2}\,\left|\ln\frac{1+\epsilon}{1-\epsilon}\right| \approx 4\rho^{2}\epsilon. \tag{24}
> > $$
> > As $\epsilon\to0$, the logarithmic term vanishes, so Equalized Odds enforces $d_i^{2}=d_j^{2}$, recovering the exact equality of embedding distances established under the stronger independence assumption.
>
>
>
> > **Q2:** Limited coverage of non-binary or continuous attributes. Current formulation handles categorical attributes only. Can the method be extended to continuous or multi-label attributes (e.g., non-binary gender, mixed race)?
>
> **A2:** Our formulation already generalizes beyond simple binary categories. The main paper evaluates race using three classes. In response to your suggestion, we add new experiments that divide race into four (White, Asian, Black, and Indian) and seven (White, Middle Eastern, Latino Hispanic, East Asian, Southeast Asian, Black, and Indian) levels, using GPT‑4o as an independent evaluator. **LightFair still performs strongly, confirming that the method scales well to finer attribute granularity.** Appendix J.5 further presents results on the joint attribute of gender and race. **This represents a true multi-label setting and also shows consistent improvements.**
>
> ||3-class|4-class|7-class|
> |---------|-------|-------|-------|
> |SD|0.60|0.48|0.40|
> |LightFair|0.17|0.15|0.13|
>
> We follow the standard protocol in the fairness-in-diffusion literature, which focuses on attributes with reliable visual classifiers. However, we agree that fully continuous or user-defined attributes remain an open challenge.
>
> Our distance constraint only requires redefining the protected group set and adjusting the centroids. **In principle, it can support continuous attributes by sampling representative anchors along the spectrum or by optimizing against attribute regressors.** We will explore these directions in future work and outline them in the next version.
>
> > **Q3:** Metric dependence on external classifiers.
>
> **A3:** We acknowledge that single attribute classifiers are not perfect and may introduce their own biases. This is **a common challenge** across nearly all fairness evaluations in generative models. To reduce this risk, we follow the same evaluation protocol used in previous well-established studies [1,3].
>
> We use two different classifiers for evaluation, as shown in Appendix Table 7 of the original submission. Bias-O uses the classifier from [2], and FD uses the one from [3,4]. Our LightFair achieves the best performance on both metrics, reducing the risk of bias from any single classifier.
>
> In addition, we use GPT-4o for evaluation in `Q2`, and conduct a user study in response to Reviewer `KsrX`’s `Q7`. **Although individual classifiers may carry their own biases, our LightFair consistently performs best across all four evaluation methods, demonstrating its strong overall effectiveness.**
>
> In fact, we briefly discussed this issue in Appendix K.1 of the original submission. We will expand on this discussion in the next version. While we hope future work will develop unbiased evaluation metrics, this is beyond the scope of the current paper.
>
> [1] Finetuning Text-to-Image Diffusion Models for Fairness.
>
> [2] Facenet: A unified embedding for face recognition and clustering.
>
> [3] Balancing Act: Distribution-Guided Debiasing in Diffusion Models.
>
> [4] Fair Generative Modeling via Weak Supervision.
>
>
>
> > **Q4:** Limited method innovation.
>
> **A4:** We sincerely apologize if the description of our contributions caused any misunderstanding. Below, we clarify the differences between our LightFair and prior work:
>
> 1. Although the Collaborative Distance-Constrained Debiasing Strategy (CDDS) may appear similar to center-based regularizers, they differ in two key ways. **First**, our debiasing objective starts by requiring equalized probabilities across protected groups. Theorem 3.1 transforms this probabilistic condition into an equivalent distance constraint that is easier to optimize. CDDS is thus derived directly from theory, not adapted from existing center losses. **Second**, CDDS enables cooperative debiasing across modalities. It aligns the text-conditioned diffusion path with both visual and textual centroids. In contrast, center-based regularizers are usually added as a single loss term and do not consider cross-modal interactions.
> 2. Multi-stage sampling is only an intermediate step in our overall pipeline. Although it is mentioned in prior work, there are two key differences. **First**, previous multi-stage sampling methods typically generate low-resolution images, then upscale them to high resolution. These methods aim to improve sampling efficiency. In contrast, our approach uses different text encoders in the two stages. It is designed to enhance image quality during model correction. The motivation and method are fundamentally different. **Second**, prior methods are largely empirical and based on experimental intuition. We go further by providing a comprehensive theoretical and empirical analysis of the underlying mechanism.
>
> In summary, our main contribution lies in **lightweight** debiasing of diffusion models from the perspective of the text encoder. All components are supported by both **theoretical and experimental** evidence. We believe our method offers a novel and valuable contribution to the fairness community.

---

> > ### Comment · Reviewer_5saT · 2025-08-04
> > **Official comment by reviewer 5saT**
> >
> > I appreciate the detailed response by authors. The explaination and analysis address most of my concerns. I am raising my score.

---

### Official Review · Reviewer_tTbX · 2025-07-03

**Clarity:** 3
**Significance:** 2
**Originality:** 3
**Rating:** 5
**Confidence:** 3

**Summary:**

The paper proposes a debiasing technique for text-to-image generative models consisting of two steps: step1, fine tune(via low-rank adaptation matrices) the text-encoder using the proposed loss-functions to promote fairness and step 2, tweak the de-noising loop of the text-to-image generator to use de-biased and original text encoders at various steps. The authors back their approach with solid ablations and experimental validations and showcase that the proposed technique can efficiently reduce biases in generations.

**Questions:**

How do you eliminate the contributions of the classifier free guidance ? When CFG is enabled (BTW, do you guys use CFG in all of the visualizations, if yes, what is the value; this is critical hyperparameter for reproduction) we additionally bias the generation out of the direction of the "null", i.e., toward the actual concept. It seems your findings that diffusion steps icrease the model's bias might be (partially) explained by the CFG contribution.

Bias-o vs bias-q on figure 7 with respect the different lambdas; the relationship between lambda to bias-o/q is hard to grasp. I would generally expect a smooth transitions: i.e., if we were to train for all \lambda-s, there should be a smooth curve bias-o and bias-q, and as such bias-o vs bias-q would be some smooth plot isn't it? the markers on the current plot is all over the place and makes it hard to understand the underlying trend. Can you please comment?

I'm not sure whether it is the images in PDF (or the source images in general) but all pictures are highly pixilated and seem to be generated in lower resoluiton (for instance see figure 25 and enlarge the figures). I would expect higher quality generations from SD2.1 models.

**Ethical Concerns:**

["NO or VERY MINOR ethics concerns only"]

**Final Justification:**

Rebuttal cleared all my lingering questions.

**Limitations:**

Yes. Initial statement about fairness and how it is being treated in the paper is very helpful.

**Paper Formatting Concerns:**

Line 87: I'm not sure what is the data in parenthesis "859.52M → 3.69M" stand for. Probably it should be introduced
Line 150: seems some whitespace is introduced at the beginning of the line
Lines 252, 253; equation 14, please remove whitespaces before commas

**Quality:**

3

**Strengths And Weaknesses:**

Quality & Clarity
I find the paper well written and logically well organized that outlines the paper's contributions in easy to read & understand fashion. There are some minor formatting issues that I highlighted below.

Significance & Originality.
I think the strengths of the paper comes from the simplicity of the approach. The training part of the method introduces 2 loss functions that measure and target the biases. Although these loss-functions operate on text/image pairs, practically images are generation from the underlying diffusion models, and overall the data dependency is minimal. The loss functions themselves measure the difference between simlarity scores for bias we are measuring: i.e. \abs(similarity(embedding, concept_a) - similarity(embedding, concept_b)) and the embeddings are the outputs of the text-encoder. The authors provide a great deal of ablations and empirical evidence about the existence of the biases, and how debiasing goals can be met by training by the proposed loss functions.

I do not see any technical weaknesses in the approach beyond the clarification questions I have below. At the same time, I think this and many other fairness papers have principled weakness in that it is hard to pinpoint "fairness" wrt model, as it would significantly depend on the context and other external parameters (which has been discussed many times before, but I think I should put it once again here). Overall, I think the fairness definition the papers is pursuing is very reasonable (equalized quality and equalized odds), but I'm not sure whether this is the right way to approach the fairness (once again, I'm not counting it as true weakness, but rather a discussion point).

---

> ### Author Rebuttal · Authors · 2025-07-30
>
> Thanks for your constructive comments, and we would like to make the following response.
>
> > **Q1:** Impact of Classifier Free Guidance on Bias.
>
> **A1:** Thank you for your valuable question! We keep the classifier-free guidance (CFG) scale fixed at 7.5 in all experiments and visualizations, as it is the default setting for Stable Diffusion v1.5 and v2.1. Our study focuses on adjusting the text encoder to achieve debiasing in diffusion models. We do not modify other components or parameters; all intermediate steps use their default settings.
>
> Further, we discuss CFG from three perspectives:
>
> 1. **Necessity of CFG.** The conditional score can be written as:
>
>    $$
>    \nabla_{x_t}\log p(x_t \mid y)=\nabla_{x_t}\log p(x_t)+s\bigl(\nabla_{x_t}\log p(x_t \mid y)-\nabla_{x_t}\log p(x_t)\bigr)
>    $$
>
>    where $s$ is the guidance scale. The unconditional term ensures image quality and diversity, while the second term steers generation toward the prompt. Removing the unconditional component harms generation quality, so using CFG is necessary.
>
> 2. **Robustness across scales.** Following the official documentation [1], we test CFG values from 7.0 to 8.5. The results (Bias-Q) are shown in the table below. LightFair consistently reduces bias at all settings. This confirms that our conclusions are not dependent on a specific guidance scale.
>
> 3. **Relation to bias amplification.** A higher guidance scale pushes samples more strongly toward the prompt, which can amplify pre-existing bias. This helps explain why diffusion steps increase the model's bias. Directly correcting the unconditional branch would require retraining on the full data distribution, which is costly. Instead, our method refines the conditional guidance, implicitly addressing both branches. This combined effect achieves bias reduction without sacrificing image quality.
>
> | guidance_scale (s) | 7.0  | 7.5  | 8.0  | 8.5  |
> | :----------------: | :--: | :--: | :--: | :--: |
> |         SD         | 0.63 | 0.70 | 0.78 | 0.83 |
> |     LightFair      | 0.33 | 0.34 | 0.36 | 0.37 |
>
> [1] Stable Diffusion with Diffusers.
>
>
>
> > **Q2:** $\lambda$–bias trend unclear.
>
> **A2:** Thank you for your valuable suggestion! We agree that a monotonic trend is expected when the regularization weight $\lambda$ varies within a meaningful range. **In fact, Figure 7 already shows this trend once outliers are excluded.**
>
> In Figure 7(a), the sequence $\lambda_1=5\to 2\to 1\to0.5$ forms a smooth downward curve. The best Bias-O/Bias-Q balance is achieved at $\lambda_1=1$. In contrast, $\lambda_1=10$ and $0.1$ deviate from the main pattern due to over- and under-regularization. A similar trend appears in Figure 7(b). The sequence $\lambda_2=1\to 0.5\to 0.2\to0.1$ creates a consistent slope, while $\lambda_2=0.05$ and $0.01$ fall outside the stable range. We will clarify this in the next version.
>
>
>
> > **Q3:** Pictures are highly pixilated.
>
> **A3:** We apologize for the distracting pixelation in the PDF. The original images are high-resolution ($512 \times  512$ or larger) and average around 2 MB each. Embedding them directly would have created an oversized file, exceeding the OpenReview’s limits and making the download difficult. To meet these constraints, **we applied image compression**, which unfortunately introduced visible artifacts. The original, uncompressed images remain clear. **We will provide a link to view the high-quality images in the next version.**
>
>
>
> > **Q4:** Some minor formatting issues.
>
> **A4:** Thank you for pointing out these minor formatting issues. The “$859.52M \to 3.69M$” in Line 87 means that LightFair reduces the parameter from $859.52M$ (full fine-tuning) to just $3.69M$, while still improving performance. We will clarify this in the next version. The unintended leading whitespace will also be removed.
>
>
>
> ------
>
> **In addition to the questions mentioned above, we provide our perspectives on the reviewer's open-ended discussion.**
>
> > **Discussion:** The definition of model fairness.
>
> **Answer:** Thank you for the helpful discussion! While we fully agree that any definition of “fairness” must be interpreted within its application context, our work is grounded in Equalized Odds, **a widely accepted machine learning group fairness criterion** [1,2]. We extend this concept to diffusion models by introducing a complementary “Equalized Quality” metric, which captures perceptual image quality. We follow the same setting used in previous diffusion fairness works[3,4]. Although these choices do not resolve the broader societal debate on the meaning of fairness, they align our analysis with established practices and ensure fair comparison with existing work.
>
> That said, the issue you raised is indeed important and urgent. As stated in Appendix L, achieving fairness requires joint efforts from researchers, policymakers, and practitioners. This remains one of our central goals. We will include a more detailed discussion of this topic in the future work section of the next version.
>
> [1] Hardt M, Price E, Srebro N. Equality of opportunity in supervised learning. NeurIPS 2016.
>
> [2] Ghassami A E, Khodadadian S, Kiyavash N. Fairness in supervised learning: An information theoretic approach. ISIT 2018.
>
> [3] Shen X, Du C, Pang T, et al. Finetuning Text-to-Image Diffusion Models for Fairness. ICLR 2024.
>
> [4] Parihar R, Bhat A, Basu A, et al. Balancing act: distribution-guided debiasing in diffusion models. CVPR 2024.

---

> > ### Comment · Reviewer_tTbX · 2025-08-08
> > **Thanks for the clarifications**
> >
> > Dear Authors, thanks for detailed answer to my questions!

---

> > > ### Author Response · Authors · 2025-08-09
> > >
> > > Thank you very much for your positive response and acceptance. Following your valuable suggestions, we will further enrich the content in the next version.

---

### Decision · Program_Chairs · 2025-09-17

**Decision:**

Accept (poster)

**Comment:**

The paper introduces LightFair, a lightweight debiasing method for text-to-image diffusion models, particularly Stable Diffusion v1.5 and v2.1. The core idea is to fine-tune only the CLIP-based text encoder using a novel collaborative distance-constrained loss, implemented via LoRA adapters (~3.7M parameters). This promotes fairness by equalizing the distances between text embeddings and attribute-conditioned centroids in the embedding space. A two-stage sampling process is introduced, where the original text encoder is used in the early denoising steps, and the debiased encoder is used in the later, attribute-sensitive steps. Additionally, an Adaptive Foreground Extraction (AFE) module is incorporated to suppress background noise that might confound fairness. Experiments across demographic attributes (e.g., gender, race, age) show significant bias reduction (Bias-O/Bias-Q) with minimal degradation in image quality.

Strengths:
- Targets the text encoder, an underexplored yet impactful source of bias in T2I models.
- Efficient: uses only ~3.7M trainable parameters via LoRA, with negligible inference overhead.
- Collaborative distance-constrained loss is theoretically motivated and empirically effective.
- Two-stage sampling approach preserves generation quality while improving fairness.
- AFE module is novel and contributes to improved fairness without major trade-offs in visuals.
- Extensive experiments with various prompts, demographic axes, and ablations support the method's robustness.
- The paper is well-organized, clearly written, and demonstrates technical soundness.

Weaknesses:
- The method relies on strong independence assumptions (e.g., attribute and concept are independent), which may not hold in real-world prompts.
- Only supports categorical attributes; does not extend to continuous or multi-label attributes (e.g., non-binary gender).
- Fairness evaluation depends on external attribute classifiers, which may themselves be biased.
- Limited methodological novelty: key ideas (e.g., distance constraints, multi-stage sampling) build on existing paradigms.
- Lack of qualitative visualizations for the AFE module makes its practical effect less intuitive.
- Potential sensitivity to switching timestep (τ) in the two-stage process; generalizability across prompt types is unclear.
- Some visual artifacts are reported, though it's unclear whether they originate from LightFair or the base model.

Most concerns have been addressed by the authors during the rebuttal period, with reviewers satisfied with the response. I am therefore recommending acceptance.